# INSTANCE - the Italian seismic dataset for machine learning

Alberto Michelini[1], Spina Cianetti[2], Sonja Gaviano[4,2], Carlo Giunchi[2], Dario Jozinović[1,3], and Valentino Lauciani[1]

[1]Istituto Nazionale di Geofisica e Vulcanologia, via di Vigna Murata, 605, 00143 Rome, Italy
[2]Istituto Nazionale di Geofisica e Vulcanologia, via Cesare Battisti, 53, Pisa Italy
[3]Unversità degli Studi Roma Tre, Largo San Leonardo Murialdo 1, Rome, Italy
[4]Unversità degli Studi di Firenze, Via La Pira 4, Firenze, Italy

*Correspondence to:* Alberto Michelini (alberto.michelini@ingv.it)

**Abstract.** The Italian earthquake waveform data are here collected in a dataset suited for machine learning analysis (ML) applications. The dataset consists of nearly 1.2 million three-component (3C) waveform traces from about 50,000 earthquakes and more than 130,000 noise 3C waveform traces, for a total of about 43,000 hours of data and an average of 21 3C traces are provided per event. The earthquake list is based on the Italian seismic bulletin (http://terremoti.ingv.it/bsi) of the "Istituto Nazionale di Geofisica e Vulcanologia" between January 2005 and January 2020 and it includes events in the magnitude range between 0.0 and 6.5. The waveform data have been recorded primarily by the Italian National Seismic Network (network code IV) and include both weak (HH, EH channels) and strong motion recordings (HN channels). All the waveform traces have a length of 120 s, are sampled at 100 Hz, and are provided both in counts and ground motion physical units after deconvolution of the instrument transfer functions. The waveform dataset is accompanied by metadata consisting of more than 100 parameters providing comprehensive information on the earthquake source, the recording stations, the trace features, and other derived quantities. This rich set of metadata allows the users to target the data selection for their own purposes. Many of these metadata can be used as labels in ML analysis or for other studies. The dataset, assembled in HDF5 format, is available at http://doi.org/10.13127/instance (Michelini et al., 2021).

## 1 Introduction

Important breakthroughs in the understanding of earthquake phenomena can be achieved through the analysis of the very large number of continuous waveform recordings stored in the existing seismic archives. To this end it can be important to make available well organized representative subsets of the archives together with their associated metadata information.

The recent developments of machine learning (ML) software platforms like TensorFlow (https://www.tensorflow.org), PyTorch (https://pytorch.org), Keras (https://keras.io), Caffe (https://caffe.berkeleyvision.org) (see Abadi et al., 2016; Paszke et al., 2019; Chollet and others, 2015; Jia et al., 2014, respectively), the availability of high performance computing hardware

(i.e., GPUs) and the access to thoroughly selected benchmark datasets (e.g., STEAD https://github.com/smousavi05/STEAD, LEN-DB https://doi.org/10.5281/zenodo.3648231) offer new opportunities to apply ML methodologies to seismological and earthquake engineering problems. In particular, the use of sophisticated and optimized ML algorithms for the analysis of large amounts of seismic data can lead to remarkable improvements for automated tasks like seismic waveform onset picking, ground motion prediction, earthquake early warning or for the detection of hidden signals currently recognized as noise or for novel modeling and inversion strategies (see Kong et al., 2018; Bergen et al., 2019; Dramsch, 2020, for recent reviews). Specifically, the advent of ML in the field of seismology has highlighted the importance of reference datasets for benchmarking the developed methodologies and it has fostered more thorough and statistically sound schemes for analyzing the data, like splitting all the available data into training, validation and test sets. Moreover, the introduction of competitions like those for predicting laboratory earthquakes launched on the Kaggle platform (https://www.kaggle.com/c/LANL-Earthquake-Prediction/data) or the SeismOlympics (Fang et al., 2017), that attracted several thousand teams, evidences even more the great potential of benchmark datasets (Johnson et al., 2021) and the general interest to tackle seismology problems with ML.

The application of ML techniques to seismological waveform data can be quite straightforward. Indeed, large amounts of labelled data are already available thanks to the analyses carried out since many decades by expert analysts that have compiled and reviewed earthquake catalogs (that include phase onset readings, earthquake location and size estimates) or that have assembled ground motion parameters in special flat files and maps of strong ground motion amongst the most common tasks. Their work provides effectively metadata that can be associated with the recorded waveforms and that can be used as labels when performing ML analysis. A main bottleneck in wide-scale implementation of ML is, however, the fast access to the waveforms and to the associated metadata. Open access waveforms archives available to the seismological community (e.g., EIDA, Strollo et al. (2021) http://eida.ingv.it/, or IRIS, Ingate (2008)) were mainly designed for preserving the continuous data and making them available to the scientific community. In practice, one of the main goals of seismological data centers has been the seamless acquisition of continuous data from the networks and the preservation, curation and archiving of the entire record of continuous waveforms. In this context, the users have complete flexibility in the selection of the data to download but accessing large data volumes can be very time consuming. Thus, despite the achievements attained in the last decades with the implementation of well tested and efficient web services (e.g., FDSN `dataselect`), the accessibility of remote servers still remains cumbersome (Quinteros et al., 2021). It follows that in order to attract a broader audience of users and developers there is a strong need to assemble and publish benchmark datasets that can be readily used with the existing software platforms (Mousavi et al., 2019). In practical terms, the matter consists of assembling quality checked data and metadata according to volume and formats ready to be used in ML applications.

Recently, effort has been made to assemble and make publicly available datasets consisting of waveforms and associated metadata. In detail, the dataset used in the works by Ross et al. (2018a, b); Meier et al. (2019) is downloadable from the Southern California Earthquake Data Center at the web portal https://scedc.caltech.edu/data/deeplearning.html. This dataset includes 4.8 million time series recorded by nearly 700 receivers from more than 270,000 earthquakes in southern California. The STEAD dataset assembled by Mousavi et al. (2019) includes 1.2 million of 3C traces comprising 450,000 local earthquakes and 100,000 noise windows recorded by more than 2600 stations at global scale. The LEN-DB dataset (Magrini et al., 2020)

is also a global dataset of local earthquakes and includes 1.2 million 3C waveform traces, half belonging to earthquakes and half to noise. The NEIC dataset (Yeck and Patton, 2020) includes global data and has been used by Yeck et al. (2020) to train the 1.3 million seismic-phase arrivals using three separate convolutional neural network models to predict arrival time onset, phase type, and distance.

Results attained by Ross et al. (2018b), Zhu et al. (2019a), Zhu et al. (2019b), Mousavi et al. (2020), and Mousavi and Beroza (2020) are excellent examples of successful applications of ML which can improve substantially the earthquake detection level with respect to most traditional methods, leading to the location of tiny and previously undetected earthquakes improving our knowledge on the heterogeneity of stress release on known and unknown faults. This enhanced information is crucial to make more thorough assessments of the ongoing seismotectonics and seismic hazard. The ML methods are likely to become an irreplaceable tool in seismology to extract as much information as possible from the large amount of data already stored in the archives. Among the indirect advantages, the enhanced detection can, to some extent, also govern network densification with sensible reductions in equipment investments and maintenance costs.

In general, the impressive performances of ML applications have been strongly related to the availability of large amounts of data with associated properly labeled metadata. Large amounts of data are critical to perform proper training and avoid data overfitting. However, the preparation of a ML dataset is also tedious and very time consuming. These are the main reasons that motivated the work presented in this article. Our goal is to provide an open access dataset consisting of raw and instrument removed waveform data and associated metadata to study earthquake occurrence in Italy. The data collection, named INSTANCE, gathers seismic waveform data from weak and strong motion stations that have been extracted from the Italian EIDA node (Danecek et al., 2021, see section 7 for a full list of the FDSN networks included in the dataset). The metadata associated to the waveforms are extracted from the INGV earthquake catalogue and from the waveform traces themselves. We expect this reference dataset to be used for several different purposes spanning from improvements of the existing configurations of seismic monitoring in Italy to the development and testing of new techniques for earthquake detection and ground motion estimation.

## 2 Earthquakes

### 2.1 Data preparation

The data collection was assembled following the main stages listed below:

1. earthquake selection;

2. station selection;

3. waveform data selection and download;

4. cross-validation between phase-based station selection and downloaded waveform data;

5. processing of the data counts waveforms;

6. application of the instrument transfer function to the waveforms.

### 2.1.1 Earthquake selection

To compile the waveform dataset, we started from the Italian Seismic Bulletin (http://terremoti.ingv.it/en/bsi, INGV bulletin hereinafter) and seismic stations archives (http://terremoti.ingv.it/iside). These data are public and can be queried using the
5 `fdsnws-event` (https://www.fdsn.org/webservices/fdsnws-event-1.2.pdf) and the `fdsnws-station` web services provided by INGV. The event data belong to the INGV bulletin which has been adopting the same velocity model and earthquake location software in the time period included in this study (see Appendix B for detail).

The first step consisted of retrieving all the earthquakes with $M \geq 0$ from 1 January 2005 to 31 January 2020 in an enlarged area within the latitude and longitude corners (35.0,5.0) and (49.0,19.0). A total of 315,225 earthquakes were found. The
10 beginning of the query corresponds approximately with the update, renovation and increase in the number of stations of the national seismic network (Michelini et al., 2016; Danecek et al., 2021; Margheriti et al., 2021). Around 2005, the INGV network (FDSN code IV) underwent a major upgrade with the existing, predominantly analog, instruments being replaced by high quality digital seismic data loggers and new, mostly broadband (and some extended short period), three-component (3C) sensors. Selected stations were also complemented with additional 3C strong motion sensors. The upgrade resulted in
more than a two fold increase of the number of stations of IV network. Also and since 2005, there have been many temporary deployments of seismic stations coinciding with earthquake sequences and specific experiments which data are also available through the EIDA INGV node (Danecek et al., 2021). The total number of stations also increased thanks to the contribution of the networks belonging to other Italian institutions (e.g., the University of Genoa, National Institute of Oceanography and Experimental Geophysics-OGS, and the University of Naples amongst others). This increment resulted in a significant
improvement of the detection of low magnitude earthquakes. At the regional scale of Italy, the magnitude of completeness of the INGV bulletin is around $\sim M1.7 - M1.8$ although significant differences occur depending on the area. To this regard, the preferred INGV catalogue magnitude is the local magnitude, Ml, (Richter, 1935) but sometimes also Mw and Md (see below for additional detail).

A relevant aspect when compiling a large dataset to be used for ML purposes consists of gathering a balanced distribution
of data. In seismology, when using earthquake magnitude for classification, balanced representation is impossible to achieve because small size earthquakes, following the Gutenberg-Richter magnitude versus the number of earthquakes power-law (Gutenberg and Richter, 1944), outnumber larger earthquakes. To address this issue (or at least to mitigate its influence), we choose to select in our target area:

- the great majority of the earthquakes with $M \geq 4.0$. The earthquakes that have been discarded (30) occurred all but five
outside the Italian country borders and mainly in the Balkan area. (The earthquakes in Italy, all with $M < 5$, will be included in a future update of the dataset.)

- earthquakes with origin times differing by more than 120 s in the range $2.0 \leq M < 4.0$;

- additional 20,000 earthquakes, randomly selected, with origin times differing by more than 120 s for $M < 2.0$.

The resulting distribution of the earthquakes according to their magnitude is detailed in Table 1 and they are mapped in Fig. 1a.

**Table (1).** Final data selection. "All" indicates the total number of earthquakes in the INGV bulletin in the time period between 01/01/2005 and 01/31/2020, "Selected" and "Percent kept" refer to the earthquakes, and "Nb. 3C records" to the waveform traces included in the dataset.

| $\geq M_{min}$ | $< M_{max}$ | All | Selected | Percent kept | Nb. 3C records |
|---|---|---|---|---|---|
| 0 | 1 | 57746 | 4462 | 7.73 | 39794 |
| 1 | 2 | 209652 | 15249 | 7.27 | 202572 |
| 2 | 3 | 43109 | 30845 | 71.55 | 757129 |
| 3 | 4 | 4342 | 3106 | 71.53 | 139338 |
| 4 | 5 | 342 | 315 | 92.11 | 18659 |
| 5 | 6 | 31 | 28 | 90.32 | 1593 |
| 6 | 7 | 3 | 3 | 100.0 | 164 |
| 0 | 7 | 315225 | 54008 | 17.13 | 1159249 |

### 2.1.2 Station selection

In order to gather high quality earthquake signals, we based our choice on the most accurately picked P- and S-wave onset phases published in the INGV bulletin. To this regard, the manual picking of the arrival phases is routinely performed by a group of about 20 INGV highly trained staff personnel who also review the hypocenter locations and magnitude determination before bulletin publication. These manually reviewed locations are indicated as *preferred* solutions in the INGV bulletin. In practice, we have selected only those stations that had P- and, if available, S-wave onset picks associated to the *preferred* location of the INGV bulletin. We note that the strong motion data provided by the national strong motion network ("Rete Accelerometrica Nazionale") operated by the Italian Department of Civil Protection do not enter in the earthquake picking and location performed by the INGV staff and the same data are not available through EIDA. They may be included, however, in future releases of the dataset.

In summary, we have adopted the following criteria to identify the waveform records to be included in the dataset after the earthquake selection above was applied:

- all stations that feature P-wave (and S-wave when available) onset phases used for the *preferred* earthquake location (no distinction is made between Pg and Pn and no secondary phases like PmP are picked);

- all stations with waveform data available through the INGV EIDA node (see the dataset contributing networks in the pie diagram of Fig. 5b);

- P- and S-wave location residual times less than 1.0 s;

- P- and S-wave phases that contributed to the location with a weight larger than 10 %.

This selection procedure reduced the number of P- and S-wave phases from $\sim 1.9$ to $\sim 1.2$ and from $\sim 1.1$ to $\sim 0.7$ millions, respectively.

### 2.1.3 Waveform data selection and download

The selection procedure described in section (2.1.2) resulted in the compilation of a list of waveform data time windows to be downloaded from the EIDA continuous waveform archive. We choose a time window of 120 s in order to include both P- and S-waves from stations whose distance is up to ~600 km from the hypocenter. Indeed, in these cases, the $S - P$ time differences are approximately 75-80 s. Adding about 20 s of signal before the P-wave time and about 20 s after the S-wave, we end up with a 120 s window choice providing the most significant earthquake signals for either the most distant stations, in case of crustal depth earthquakes, or closer stations, in case of deep earthquakes of the Calabrian arc subduction.

More technically, the time windows set for data download were defined by inserting a randomly selected buffer time ranging between 15 and 20 s before the P-wave onset arrival phase and enlarging the time window to 125 s. The adoption of 125 s long windows at the data download stage is arbitrary since after data processing the time windows have been all set to 120 s. This criterion ensured that the great majority of the waveform traces downloaded featured a pre-P-wave onset buffer time between 15 and 20 s. However, we found that, when dealing with such a large number of waveforms acquired by diversified instruments configured differently, some discrepancies may occur. In practice, since the data are archived in miniSEED compressed format that feature different sizes of the logical records, and since the web service extracts the full logical record containing the pre-defined trace start time, it can occur that the start time of the trace is earlier than the predefined minimum time of 20 s (i.e., in this case, there is a longer time interval between the P-arrival and the actual trace start time). In contrast, when data are missing before the P-wave onset time (i.e., in the 15–20 s pre-P-onset buffer time), start time of the extracted window can be delayed and a shorter time interval will separate the trace window start time from the P-wave arrival time (i.e., $< 15$ s). See Fig. D1 in the appendix for the distribution of the P- and S-wave phase arrival time samples. The data (miniSEED format) were downloaded using the FDSN `dataselect` web services provided by INGV (http://terremoti.ingv.it/en/webservices_and_software). Using a set of 14 container based querying procedures running in parallel, this stage required about 7 days to complete the ~4 million waveform traces (i.e., ~1.3 million of 3C traces) download for a storage requirement of ~80 GB (miniSEED STEIM1 compression).

### 2.1.4 Cross-validation between phases-based metadata and downloaded waveform data

After the massive data download was concluded, a list of all the downloaded files was generated. This list was *intersected* with the originally selected metadata (section 2.1.2) to have a *one-to-one* correspondence between the miniSEED data and the metadata (i.e., each 3C waveform record — three miniSEED files —must correspond to a row of the metadata file).

### 2.1.5 Preparation of processed waveforms in digital units

This part of our data assembling procedure targets the preparation of the digital counts waveform traces. It includes the following steps.

- – removal of traces containing data gaps (i.e., missing data);

- – trimming the waveform trace to the nearest sample to the start time;

- – 120 s trace windowing;

- – removal of mean and linear trends from the data;

- – re-sampling at 100 Hz;

- – calculation of the signal-to-noise ratio;

- – extraction of the data quality metrics.

No rotation of the horizontal component along the N-S and E-W directions was required since all sensors used are oriented accordingly. For each waveform trace (i.e., each component), the maximum value of signal-to-noise ratio ($SNR$) was extracted and kept as metadata. The $SNR$ was calculated as

$$SNR = 20 \, log_{10} \frac{|S_{95}|}{|N_{95}|} \tag{1}$$

where $|S_{95}|$ and $|N_{95}|$ are the 95 % percentile of the data absolute values in a 5 s window immediately after the S-wave onset and right before the P-wave arrival time. If the S-wave onset were not available, the S-wave window was determined after calculation of the predicted S-wave arrival using an average velocity of 3.0 km s$^{-1}$ and the hypocentral distance.

During this stage of the data preparation, we have also calculated some quality parameters extracted from the waveform traces to the purpose of a later inclusion in the metadata information. These additional parameters, providing the distribution
of the trace values, have been computed using the `MSEEDMetadata` class of the `obspy` python software (Beyreuther et al., 2010; Megies et al., 2011; Krischer et al., 2015). To the same purpose, we have determined the number of spikes using a Hampel filter on a 161 samples sliding window to find outliers in the traces.

The final dataset consists of a total of 1,159,249 3C waveform data records from 54,008 earthquakes in counts units assembled within an `hdf5` format file. Table 1 provides the number of traces within each magnitude interval of the final assembled
dataset.

### 2.1.6 Application of the instrument transfer function to the waveforms

To make the dataset of more general use, we have also generated a dataset in units of physical ground motion after deconvolving the instrument response. To this end, we have downloaded the station response files for all the stations used and applied the

transfer functions to the individual traces with frequency filtering corners 0.01, 0.04, 25, 40 Hz using a cosine flank frequency domain taper (see `cosine_sac_taper` in Obspy), and applying a 5 % cosine tapering at both ends of the trace signal. After removing the instrument response, we extracted the intensity measures (IMs, i.e., peak ground acceleration, PGA, peak ground velocity, PGV, and the spectral accelerations at 0.3, 1.0 and 3.0 seconds period) on each component so that they could be included amongst the metadata parameters. Peak ground displacements are not included since they be from single or double integration of velocity and acceleration records, respectively, and their determination can result inaccurate when performed automatically.

## 2.2 Metadata description

The 115 metadata associated to each 3C waveform trace of our collection are listed in Table 2. They provide different kind of information that can be subdivided into four main types — *source*, *station*, *trace* and *path* metadata. The unit of each metadata is provided in its denomination.

The *source* metadata provide information on the earthquake with description of the source origin time, location, size and, when available, the focal mechanism, the moment tensor, and the finite fault.

The *station* metadata provide information on the characteristics of the recording station which include the station, channel, network and location (SCNL) (cf. http://www.fdsn.org/seed_manual/SEEDManual_V2.4.pdf), the geographical coordinates and the average shear-wave velocity of the top 30 m of the Earth, $V_{S,30}$, which is an important parameter for classifying sites in seismic engineering applications (e.g., Boore, 2004) and is extracted from the map used in the INGV implementation of the USGS-ShakeMap software in Italy (Michelini et al., 2019).

The *trace* metadata consists of parameters that are extracted from the waveform traces like maximum and minimum amplitudes, root mean squared values of the traces and, after application of the transfer function, intensity measures (IMs) of the ground motion. In this class of metadata, we include the P- (and S-wave) provided by the INGV bulletin and, in addition, the number of P and S picks obtained by processing the waveforms with two deep learning phase picking and event detection algorithms (GPD and EQTransformer; Ross et al., 2018a; Mousavi et al., 2020) to make the user aware that the waveform trace being used may include more than a single earthquake (see discussion further below).

The *path* metadata follow from the calculation of parameters that link the types of metadata above (e.g., traveltimes, hypocentral and epicentral distances).

The rationale of our metadata selection reflects our intention of providing the users with comprehensive information about the data. This appears an important issue since the data, being recorded automatically, can suffer of many diverse problems deriving from malfunctioning of the data loggers, of the sensors or from poor data transmission. Since we seek to assemble a data set that can be used also for analysing real time data streams using ML, we note that the automatic processing summarized above does not differ significantly from that routinely applied to the streamed data.

One alternative to our metadata "comprehensive" approach would have consisted of "cleaning" the dataset by removing the faulty traces from the dataset altogether. We do not think this approach appropriate since in this case the dataset would not be representative of the "true" data that are collected in real-time by the monitoring networks. Thus, the basic idea behind our

criterion is that we would like to enable the users to make their own choices using opportune filters to exploit the data for their own purposes. For example, if a user looks for the cleanest data, this can be achieved by filtering the metadata accordingly (e.g., saturated velocimetric data acquired by broadband sensors equipped with 24 bit data loggers could be removed in a conservative fashion just by selecting only those traces with counts within $\pm 0.8 * 2^{23}$). In contrast, the user could also opt to leave the ML model to learn the "data problems" so that they can be detected when using real data. An approach of this kind has been used by (Jozinovic et al., 2020) for missing data. In Jozinovic et al. (2020), the dataset used for ML consists of a fixed number of stations and when data from one or more stations are missing (either the whole trace or parts of it), the signal trace is set to be an array of zeros. The ML model used there was found to detect and learn the problematic values, and compensate for it, having a similar prediction accuracy on those stations as the accuracy on the stations which had the input data available . In practice, the provision of a rich set of waveform descriptive metadata is important not only to make use of an enlarged suite of labels that can be used for diverse purposes but also to identify problems with the waveform data and include or filter them out.

Our metadata includes P- and S-wave onsets manually picked by INGV analysts as provided in the INGV bulletin. Recall that the traces were selected to include just one P-wave arrival time and possibly one S-wave arrival time since we sought to assemble one earthquake per window trace. This criterion was chosen to the purpose of facilitating the training of ML models using traces containing just one earthquake (e.g., for phase picking, peak ground motions, ...). However, even though we have made considerable efforts to isolate only one earthquake per time window, more than one can be present effectively within the same time window (e.g., the analyst did not see or just disregarded other events with smaller amplitudes). Because the presence of additional, unidentified earthquakes adds complexities to the ML training phase, we followed the same approach taken by Mousavi et al. (2019) to run automatic picking algorithms upon the waveform dataset and include as metadata also the number of P- and S-wave phases picked automatically by the generalized phase detection, GPD, technique proposed by Ross et al. (2018a) and the EQTransformer technique by Mousavi et al. (2020). In the analysis we have used as detection threshold: 0.99 for P- and S-phase detection for GPD, and 0.2, 0.1 and 0.1 for earthquakes, P- and S-phase detection, respectively, for EQTransformer. Both GPD and EQTransformer have been run only on the high gain channels (i.e., HH, EH).

As presented above, metadata are important constituents of data collections. They can be used for identifying the data to be analyzed and they can be used as labels in ML applications. In addition to the fact that not all the metadata information in INSTANCE is always available (e.g., moment tensors are generally available only for events with magnitudes $\sim M \geq 3.5$ or the S-wave onset pick retrieved from the INGV bulletin may not be present), we have found that the automatically processed ground motion trace data may suffer from errors because the original traces contained already undetected malfunctioning problems (e.g., spikes, anomalous trends) which, after application of the instrument transfer function, are mapped into erroneous ground motion traces and IM values. Similarly, it may have also occurred that in isolated cases the coefficients of the instrument transfer functions were incorrect producing also in this case incorrect traces and IM values. To address these problems, we have operated in two ways. First, we have chosen to detect the traces' maximum and minimum values lying outside the acceptable physical range and to replace them with *numpy* `nan` in the metadata file. This acceptable range was based on the IMs reported in the "flat" file of the ESM DB (https://esm-db.eu/, Lanzano et al., 2018) which includes all the IMs (obtained

from analyst processing) of all the recordings available of earthquakes with $M \geq 4.0$ in Europe. Secondly, we have verified our instrument transfer function processing procedure by cross-validating all our IM values with those reported in the ESM DB "flat" file. To this regard, we found a very good correspondence between the IMs obtained using the two methodologies giving us confidence in the quality of the applied data processing and of the IM metadata being provided.

**Table (2).** List of the metadata for the events and noise waveform traces. The units are given in parenthesis in the "Description" column. Only a subset of metadata can be associated to the noise traces (star in the "Noise" column).

| Metadata parameter-name | Noise | Description |
|---|---|---|
| source_id | ⋆ | Earthquake and noise ID (INGV and UTC time, respectively) |
| source_origin_time | | Location preferred origin time (YYYY-MM-DDTHH:MM:SS.SSZ) |
| source_latitude_deg | | Location preferred latitude (°) |
| source_longitude_deg | | Location preferred longitude (°) |
| source_depth_km | | Location preferred depth (km) |
| source_origin_uncertainty_s | | Location preferred origin time uncertainty (s) |
| source_latitude_uncertainty_deg | | Location preferred latitude uncertainty (°) |
| source_longitude_uncertainty_deg | | Location preferred longitude uncertainty (°) |
| source_depth_uncertainty_km | | Location preferred depth uncertainty (km) |
| source_stderror_s | | Preferred earthquake location standard deviation (s) |
| source_gap_deg | | Location preferred location gap (°) |
| source_horizontal_uncertainty_km | | Location preferred horizontal uncertainty (km) |
| source_magnitude | | Preferred magnitude |
| source_magnitude_type | | Preferred magnitude type |
| source_mt_eval_mode | | Moment tensor evaluation mode (e.g., manual) |
| source_mt_status | | Status of the evaluation ('reviewed' or 'final') |
| source_mt_scalar_moment | | Scalar moment (N m) |
| source_mechanism_strike_dip_rake | | Strike, dip, rake of the two planes (2 tuples) |
| source_mechanism_moment_tensor | | 6 components of the moment tensor (m_rr, m_tt, m_pp, m_rt, m_rp, m_tp) |
| source_type | | earthquake or other sources (e.g., quarry_blast, controlled explosion, experimental explosion, ...) |
| station_network_code | ⋆ | Two characters FDSN network code (e.g., IV) |
| station_code | ⋆ | Station name (International Registry of Seismograph Stations (IR)) |
| station_location_code | ⋆ | Location name identifier (Buland, 2006) |
| station_channels | ⋆ | Two characters identifying the sampling and the intrument gain (e.g., HN, HH, EH, ...) |
| station_latitude_deg | ⋆ | Station latitude (°) |
| station_longitude_deg | ⋆ | Station longitude (°) |
| station_elevation_m | ⋆ | Station elevation (m) |
| station_vs30_mps | ⋆ | $V_{S,30}$ (m s$^{-1}$) |
| station_vs30_detail | ⋆ | $V_{S,30}$ information |
| path_ep_distance_km | | Epicentral distance |
| path_hyp_distance_km | | Hypocentral distance |
| path_azimuth_deg | | Direction from event location to station (°) |
| path_backazimuth_deg | | Direction from station location to event epicenter (°) |
| path_residual_[P,S]_s | | P- or S-arrival time residual between picked arrival time and traveltime using preferred location (s) |
| path_weight_phase_location_[P,S] | | P- or S-phase location weight resulting from preferred location (range 0-100) |
| path_travel_time_[P,S]_s | | P- or S-wave traveltime (s) |

**Table (2).** List of the metadata, continued.

(a) The horizontal double line separates the additional metadata obtained after application of the instrument response transfer function.

| Metadata parameter-name | Noise | Description |
|---|---|---|
| trace_name | ⋆ | Waveform name within the hdf5 file |
| trace_start_time | ⋆ | Waveform trace UTC start time (YYYY-MM-DDTHH:MM:SS.SSZ) |
| trace_dt_s | ⋆ | Sampling interval (s) |
| trace_npts | ⋆ | Number of samples in waveform trace ($integer$) |
| trace_[P,S]_uncertainty_s | | Assigned P- or S-onset arrival time uncertainty (s) |
| trace_eval_[P,S] | | P- or S-type of picking (currently only 'manual') |
| trace_[P,S]_arrival_time | | P- or S-arrival UTC start time (YYYY-MM-DDTHH:MM:SS.SSZ) |
| trace_polarity | | P onset polarity ('negative','positive','undecidable') |
| trace_[P,S]_arrival_sample | | P-, S-onset sample number on waveform trace ($integer$) |
| trace_[E,N,Z]_median_counts | ⋆ | E-, N- or Z-component sample median ($counts$, $integer$) |
| trace_[E,N,Z]_mean_counts | ⋆ | E-, N- or Z-component sample mean ($counts$, $integer$) |
| trace_[E,N,Z]_min_counts | ⋆ | E-, N- or Z-component sample minumum ($counts$, $integer$) |
| trace_[E,N,Z]_max_counts | ⋆ | E-, N- or Z-component sample maximum ($counts$, $integer$) |
| trace_[E,N,Z]_rms_counts | ⋆ | E-, N- or Z-component sample root mean squared |
| trace_[E,N,Z]_lower_quartile_counts | ⋆ | E-, N- or Z-component sample lower quartile ($counts$, $integer$) |
| trace_[E,N,Z]_upper_quartile_counts | ⋆ | E-, N- or Z-component sample upper quartile ($counts$, $integer$) |
| trace_[E,N,Z]_snr_db | | E-, N- or Z-component signal to noise ratio |
| trace_[E,N,Z]_spikes | ⋆ | E-, N- or Z-component number of spikes ($integer$) |
| trace_GPD_[P,S]_number | ⋆ | P, S number of picks retrieved with GPD |
| trace_EQT_[P,S]_number | ⋆ | P, S number of picks retrieved with EQT |
| trace_EQT_number_detections | ⋆ | Number of detections retrieved with EQT |
| trace_[E,N,Z]_pga_cmps2 | | E-, N- or Z-component PGA (cm s$^{-2}$) |
| trace_[E,N,Z]_pgv_cmps | | E-, N- or Z-component PGV (cm s$^{-1}$) |
| trace_[E,N,Z]_pga_perc | | E-, N- or Z-component PGA (% g) |
| trace_[E,N,Z]_pga_time | | E-, N- or Z-component PGA UTC time (YYYY-MM-DDTHH:MM:SS.SSZ) |
| trace_[E,N,Z]_pgv_time | | E-, N- or Z-component PGV UTC time (YYYY-MM-DDTHH:MM:SS.SSZ) |
| trace_[E,N,Z]_sa03_cmps2 | | E-, N- or Z-component spectral acceleration at t=0.3 (cm s$^{-2}$) |
| trace_[E,N,Z]_sa10_cmps2 | | E-, N- or Z-component spectral acceleration at t=1.0 (cm s$^{-2}$) |
| trace_[E,N,Z]_sa30_cmps2 | | E-, N- or Z-component spectral acceleration at t=3.0 (cm s$^{-2}$) |
| trace_pga_cmps2 | | Max.horizontal components PGA value (cm s$^{-2}$) |
| trace_pgv_cmps | | Max.horizontal components PGV value (cm s$^{-2}$) |
| trace_pga_perc | | Max.horizontal components PGA value (% g) |
| trace_sa03_cmps2 | | Max.horizontal components spectral acceleration (t=0.3) (cm s$^{-2}$) |
| trace_sa10_cmps2 | | Max.horizontal components spectral acceleration (t=1.0) (cm s$^{-2}$) |
| trace_sa30_cmps2 | | Max.horizontal components spectral acceleration (t=3.0) (cm s$^{-2}$) |
| trace_deconvolved_units | | ground motion units of the traces in the HDF5 volume (e.g., mps and mps2 for m s$^{-1}$ and m s$^{-2}$, respectively) |

## 2.3 Dataset description

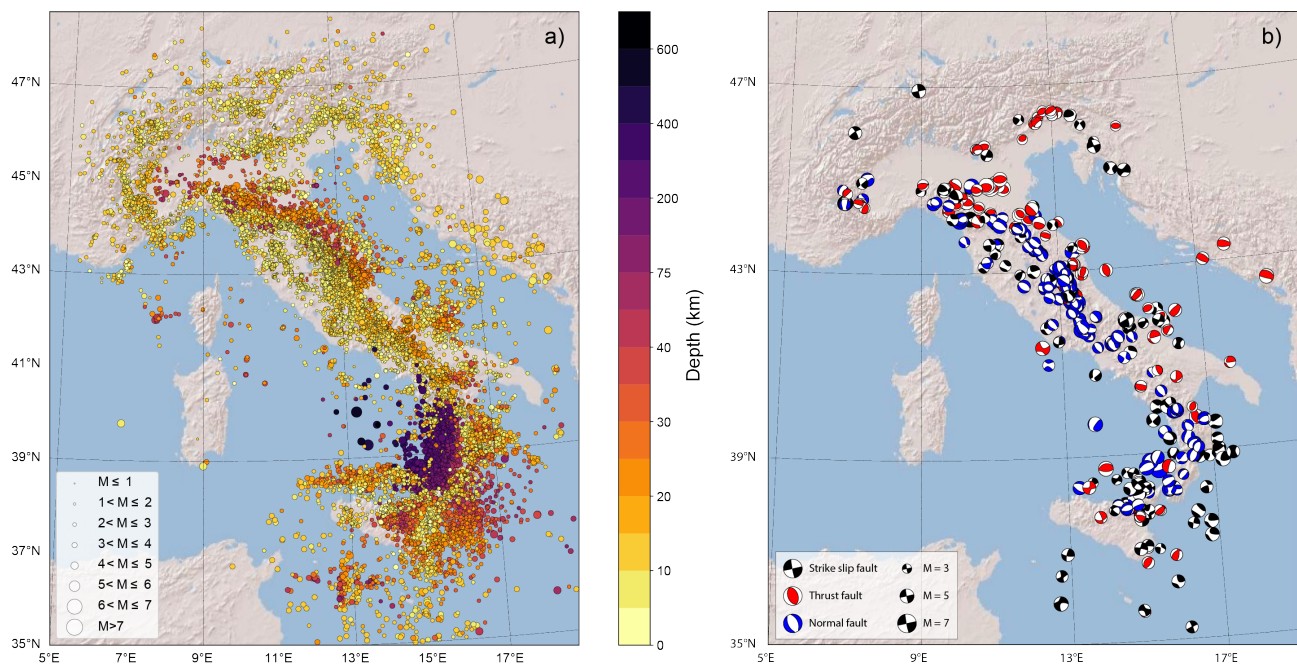

**Figure (1).** Map of the earthquakes included in the dataset shown as solid circles with colors selected according to depth (a), and map of the available moment tensors with colors assigned depending on the focal mechanism (b). Symbol size, in both maps, is proportional to earthquake magnitude.

Figure 1a shows the earthquakes included in the dataset. The symbol size is proportional to the earthquake magnitude. We observe that the 54,008 selected earthquakes composing the dataset can be considered a representative subset of the entire seismicity in Italy and, for the larger events, also for those earthquakes occurring in the near vicinity to the Italian national borders. During the time span of our data selection three important sequences have occurred in Italy after the main shocks of the 2009 L'Aquila M 6.0, the 2012 Emilia M 5.9 and the 2016 central Italy extended sequence which featured three main earthquakes with magnitudes M 6.0, M 5.9, and M 6.5.

In Fig. 1b we plot the 527 moment tensors included in the metadata. The size of the moment tensors symbol is proportional to `source_magnitude` while the colors are defined according to the prevalent strain regime: black, blue and red for strike slip, normal and thrust faults, respectively. The prevalent strain regime is determined according to the fault's rake as derived from `source_mechanisms_strike_dip_rake`: strike slip for $-45° <$ rake $< 45°$ and $135° <$ rake $< 225°$; normal for $225° \leq$ rake $\leq 315°$; thrust for $45° \leq$ rake $\leq 135°$.

In Fig. 2 we show the maps of the stations included in the events and noise datasets, respectively. The symbol size in panel a) is proportional to the number of reported phase arrivals by each station, while in panel b) is proportional to the number of waveforms included in the dataset for each station. Figure 2a demonstrates that quite a different number of phases have been

reported by the stations included in the event dataset. These differences depend on several factors like whether the stations are permanent or temporary, the time length of the acquisition, the noise level and the level of seismicity of the area where the stations have been deployed. For example, it is evident that many stations in central Italy display many phases (and associated trace recordings) mainly because the area was struck by the 2009 and 2016 earthquake sequences. In contrast, stations that

5    are located in the Po Plain generally feature small number of phases mainly because the noise level is high making the phase picking difficult. The same diversification in the number of available traces is not observable for the noise dataset shown in Fig. 2b. This occurs because it was an intentional choice to select a more or less even number of traces for all the station channels.

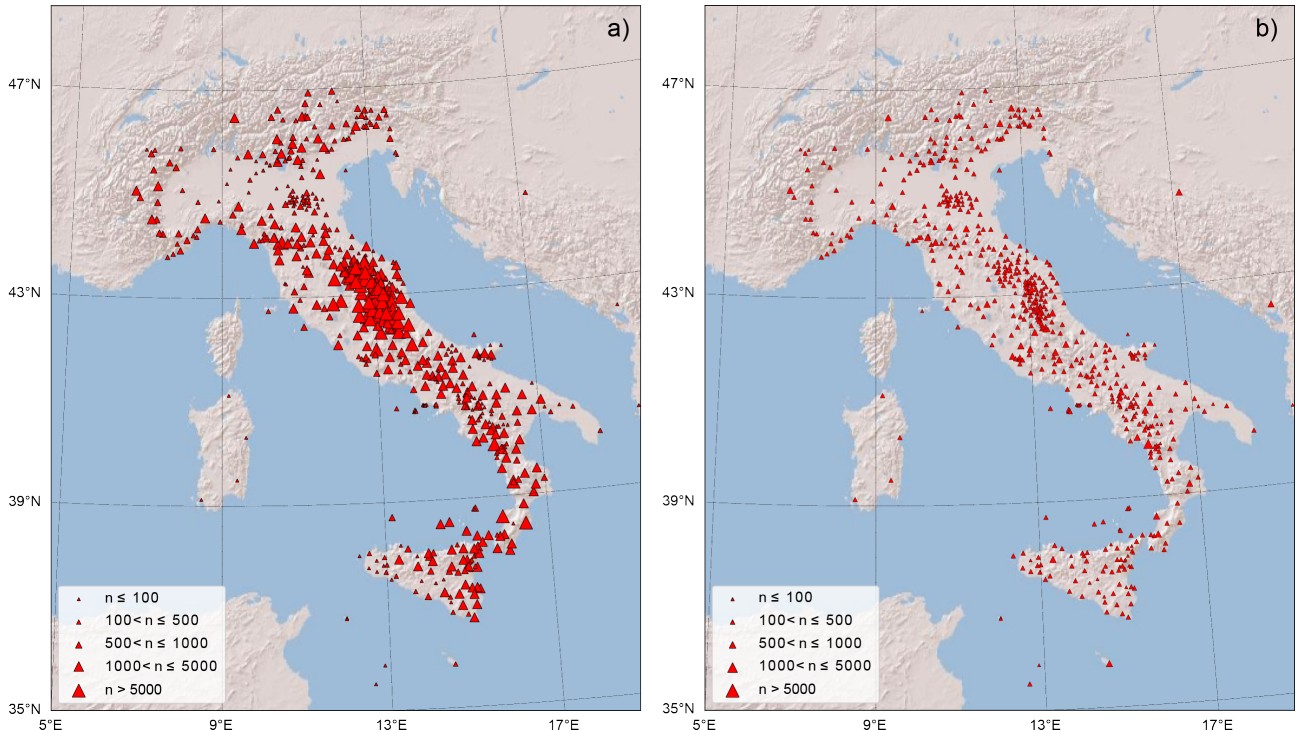

**Figure (2).** Map of the stations used to assemble the events (a) and noise (b) datasets. The symbol size in (a) is proportional to the number of P-phases and corresponding waveform traces available for each station. In (b) the symbol size is proportional to the number of traces. A total of 620 stations are included.

In Fig. 3, we show the distribution according to magnitude, earthquake to station epicentral distance, earthquake depth and

10    backazimuth of the 3C record traces composing the dataset. The panels show the histograms using the $log_{10}$ scale to provide a complete representation of the distribution of the dataset. We adopt the linear scale, however, to emphasize the distribution of the backazimuth in Fig. 3d. Despite the attempt to balance the distribution of earthquakes according to magnitude (Sect. 2.1.1), Fig. 3a shows that our selection still reflects (inevitably) the Gutenberg-Richter increase of the number of earthquakes at smaller magnitudes. The largest amount of trace records in the dataset belongs to earthquakes in the magnitude range $2 \leq M < 3$. The

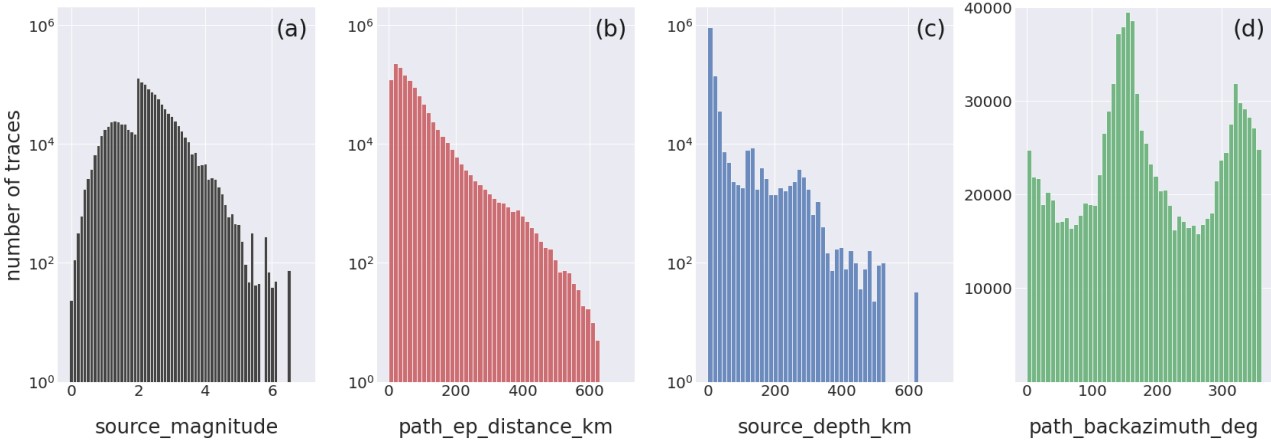

**Figure (3).** Histograms of the distribution of the trace records composing the dataset according to magnitude (a), epicentral distances (b), earthquake depth (c) and backazimuth (d). The labels of the horizontal axis are assigned using the metadata names listed in Table 2.

significant decrease of the number of traces for $M < 2$ follows from our choice to balance the dataset at small magnitudes by taking only about 7 % of the whole dataset. For what concerns the epicentral distances of the stations (Fig. 3b), the great majority of the traces have been recorded within 200 km. A better appreciation of the selected traces can be obtained from the observation of Fig. 4 where we show the magnitude versus hypocentral distance distribution of the dataset traces represented
as density plots using hexagon binning (hexbin, Hunter, 2007). The earthquake depth distribution (Fig. 3c) shows that the great majority of the traces belong to shallow crustal earthquakes although a few thousand occur in the depth range 100 to 300 km. At greater depths, the number of traces decreases sharply and only a few hundred or less recordings are included in the depth range 400 to 550 km. Figure 3d shows that the great majority of the P- and S-wave onsets belong to paths more frequent along the NW-SE direction in agreement with the geographical trend of the Apennines and of peninsular Italy overall.

Figure 5a shows the distribution of the trace channels of the dataset (`station_channels`). The weak motion, high gain channels represent more than 70 % of the total number of traces. These are subdivided into HH channels associated to the broadband high gain velocimeters (51 %) of the total whereas the extended short period channels (EH) traces account for 20 %. The low gain accelerometric channels form the remaining part of the dataset. In Fig. 5b, we show the distribution of the records subdivided according to the different networks (`station_network_code`) operating in Italy and in neighboring
countries that have been included in the dataset. The dominant portion of the data (~96 %) have been acquired by the Italian National Seismic Network (IV code) and by the MedNet (MN code) both operated by INGV (Michelini et al., 2016; Danecek et al., 2021). The full list of the contributing networks is provided in the caption.

     The polarities associated to the P-wave onsets (`trace_polarity`) are shown in Fig. 5c and have been reported in only 20 % of the total number of traces. Although this represents only a fraction of the dataset, we are confident that its number
(~235,000 ) is likely large enough to be used in a ML dedicated model (e.g., Ross et al., 2018b) for training and testing, and

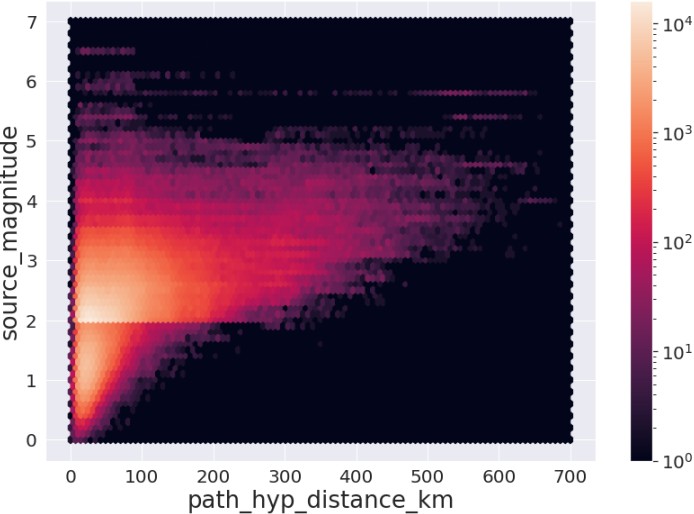

**Figure (4).** Diagram of the earthquake magnitude distribution of the dataset versus receiver distance represented as hexbin plot. The labels are assigned using the metadata names listed in Table 2.

then used to recover the polarities of the remaining batch. To this regard, it is noteworthy to observe that the positive and negative polarities have a ratio nearly 2:1. In the Appendix C we have examined the possible origin for this asymmetry. In addition, we would like to point out that, although the `source_type` is provided amongst the metadata, there are inherent difficulties to identify man made sources by the staff analysts.

The magnitude type distribution (`source_magnitude_type`) is shown in Fig. 5d. The Wood-Anderson local magnitude $M_L$ (Richter, 1935) is calculated predominantly (~96 %). The moment magnitude $M_w$ is determined for earthquakes with ~$M_L \geq 3.5$ and when enough good quality station data are available (Scognamiglio et al., 2009). The $M_d$ magnitude is used only when it is impossible to determine the $M_L$ and it is provided mainly in the first years of the dataset when the IV network still included a considerable number of analog stations.

In Fig. 6a,c we present the histograms of the P- and S-wave residual times included in the dataset. Fig. 6b,d shows the phase arrival weights resulting from the earthquake locations for P- and S-phases, respectively.

To provide a broader perspective of the dataset and with the intent of showing the wide range of waveform paths that have been included we present, in Fig. 7, the hexbin plots of the traveltime for both P- and S-wave arrival times used in the locations. These panels have been arranged using four different maximum distances and are useful for visualizing the dominant structure

of the data selection given the large number of data. More specifically, it can be observed that the hexagon binning panels for the larger distance ranges (700 and 200 km max distance) and for both P- and S-wave traveltimes (Fig. 7a,b,e,f) highlight well both the direct and Moho refracted travel times. At smaller distance ranges (100 and 40 km, Fig. 7c,d,g,h), it is evident that our dataset includes waveforms that propagated across crustal structures with different velocities. This is very evident, for example, for both P- and S-wave in the hexbin plots where at small distance are observable very low Vp and Vs velocities.

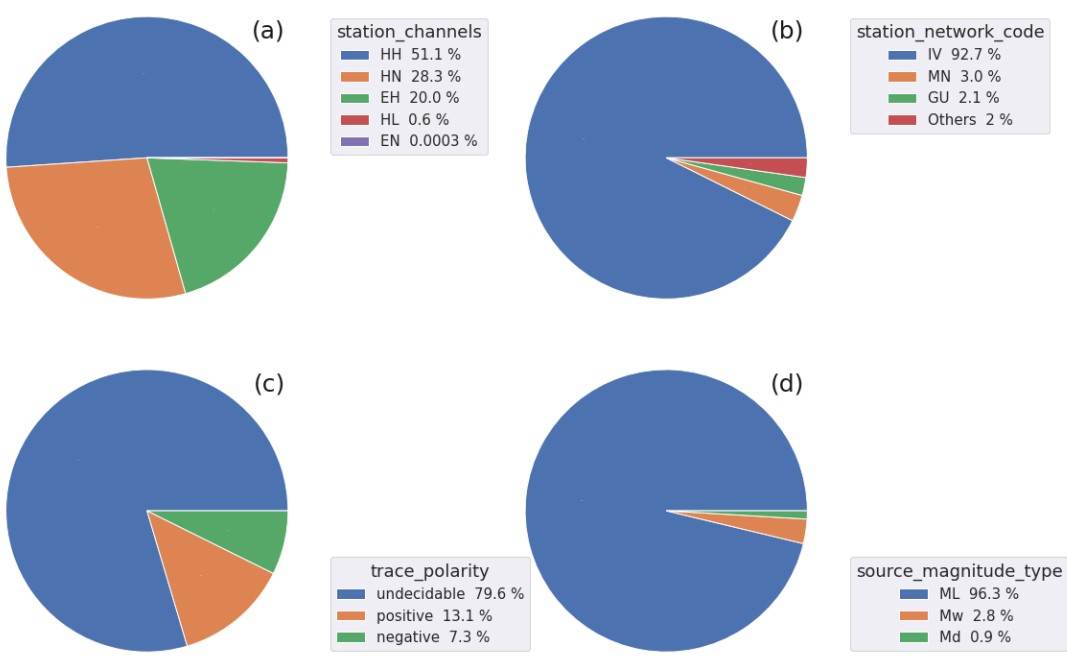

**Figure (5).** Pie diagrams of the earthquake dataset summarizing the distribution of the channels (a), the data contributing networks (b), the P-wave polarities (c) and the magnitude types (d) of the dataset. The full list of `station_network_code` with % < 1 collected in Others in decreasing order is OX, ST, SI, XO, NI, IX, OT, RF, YD, TV, B1, AC, HL, ZM, 3A. See the metadata names listed in Table 2 for the specific metadata being represented.

In the following, we will focus on the *trace* amplitude metadata. These parameters are important for refined selection of the traces and are extracted from both the raw waveforms expressed in "counts" and from the traces in physical units after application of the instrument transfer function. Some of these parameters can be obtained without any knowledge on the earthquake source parameters whereas others, like the SNR, require knowledge on the arrival times of P- and S-wave onset times.

The panels of Fig. 8 display the median (`trace_[E,N,Z]_median_counts`) and the mean values (`trace_[E,N,Z]_mean_counts`) of the dataset traces. To evidence the whole range of values attained by these two metadata, we adopt the base-10 log scale. The histograms show, for all the three components, remarkable differences of the distributions. The mean values, which have been removed in the pre-processing preparation stage (cf Sect. 2.1.5), are (obviously) centered about the zero value whereas the median histograms, while being similarly centered about the zero value, do

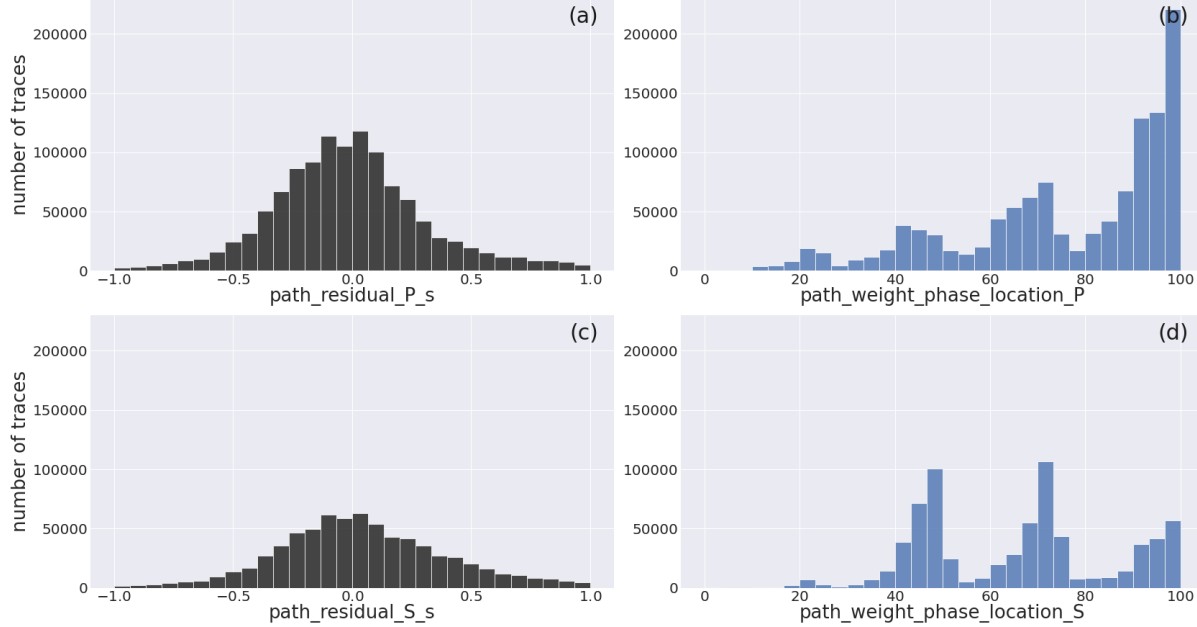

**Figure (6).** Histogram of the P- and S-wave residuals (a,c) and of the pre-assigned phase arrival weights, expressed as percent, resulting from the location (b,d). The metadata names listed in Table 2 are used as labels for the specific metadata being represented.

display a broader distribution of values around zero. This last behaviour occurs whenever the waveform trace values are unevenly distributed about the mean and it derives from the pre-processing of faulty traces that, for example, results into defective removal of the linear trend. The same figure for the full range of the parameters, is available in Fig. D2.

In Fig. 9, we present the histograms of the distribution of the trace quality control parameters obtained from the application

of the `MSEEDMetadata` module of the `Obspy` seismological software suite. The figure shows the distribution of the quality control parameters in a closer view (see the full range of values in Fig. D3). The histograms show that the largest majority of the traces feature root mean square values less than $2.5 \times 10^4$ with a minor contribution from traces featuring higher values. The minimum and maximum values follow a similar trend for negative and positive values, respectively. The lower and upper quartile values of the traces show that the peak of the distributions are at $\mp 2 \times 10^3$, respectively.

The SNR distributions are shown in Fig. 10 as histograms and versus distance and magnitude ($M \geq 2$) as hexbin plots in Fig. 11. The histograms show that the peak values for the whole dataset are at ~10 db for the two horizontal components and slightly less for the vertical (~6 db). This is expected because the S-wave motion in the shallow, near surface low velocity layers is polarized on a plane perpendicular to the nearly vertical propagation direction of the wavefront, implying that the ground motion occurs mainly along the horizontal components. In any event, the distribution of the SNR values of our dataset can be

considered sensible given that values larger than 2 already provide distinct earthquake signals. In contrast and at the lower end of the SNR distribution, we find that 10% (see Table 3) of the trace data of the HH channels have SNR values less than 2.3

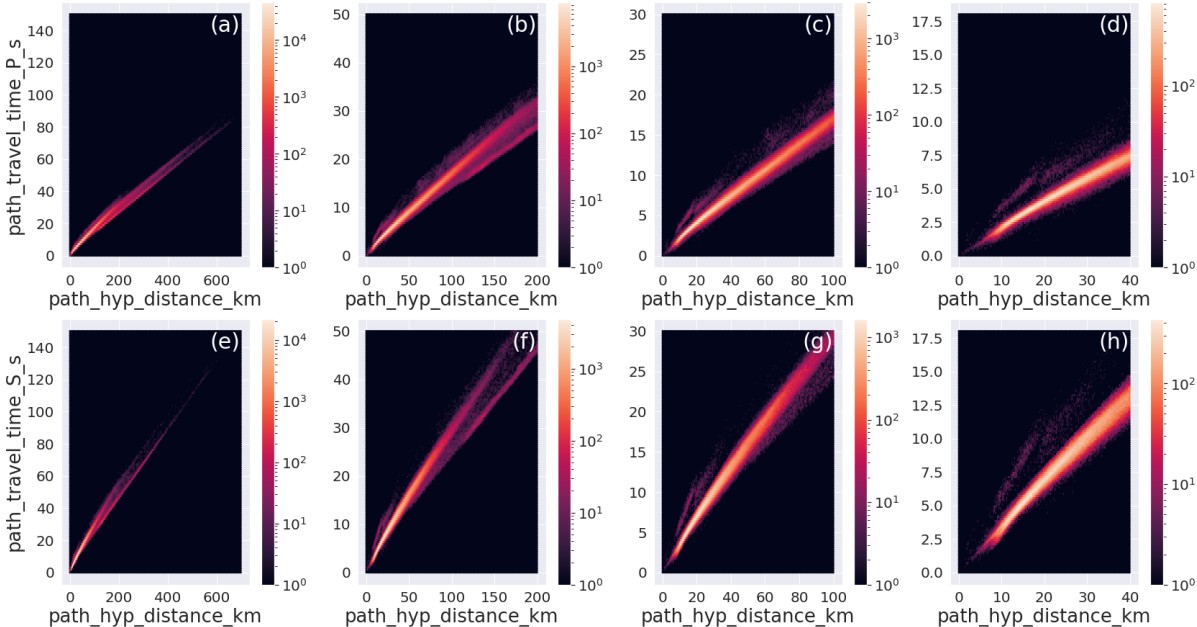

**Figure (7).** Hexbin plot of the traveltimes for different hypocentral distance ranges for P- (top panels) and S-waves (bottom panels).(a,e) 0-700 km; (b,f) 0-200 km; (c,g) 0-100 km; (d,h) 0-40 km. The metadata names listed in Table 2 are used as labels.

(1.2 for the vertical component) that corresponds to roughly to 59,000 waveform traces out of the 592,000 traces of the HH channels included in the dataset. This number of low SNR traces could still be used, for example, to train machine learning models aimed to the detection of very small magnitude earthquakes little above the background noise level.

The hexbin plots of Fig. 11 provide a snapshot of the dominant levels of SNR with distance and magnitude. It is observed that higher values occur for nearby earthquakes and that the SNR progressively decreases at farther distances. Conversely and as expected, the SNR generally increases with larger magnitude earthquakes.

The hexbin plots of the distribution of the IMs with distance for earthquakes with $M \geq 2$ are shown in Fig. 12 whereas their associated distributions are shown in Fig. D4. The panels evidence a broad concentration of ground motion values deriving from the inclusion of earthquake recordings from different distances and magnitudes. The panels also evidence some horizontal stripes at higher and lower values of ground motion resulting presumably from the acquisition and processing problems mentioned in Sect. (2.2).

To show the distance dependence of the IMs for a given magnitude, in Fig. 13 we plot the values for $M = 3$ earthquakes (i.e., IMs in the range $2.9 \leq M \leq 3.1$). The maximum concentration of IMs represent an average ground motion model for for $M = 3$ earthquakes in Italy.

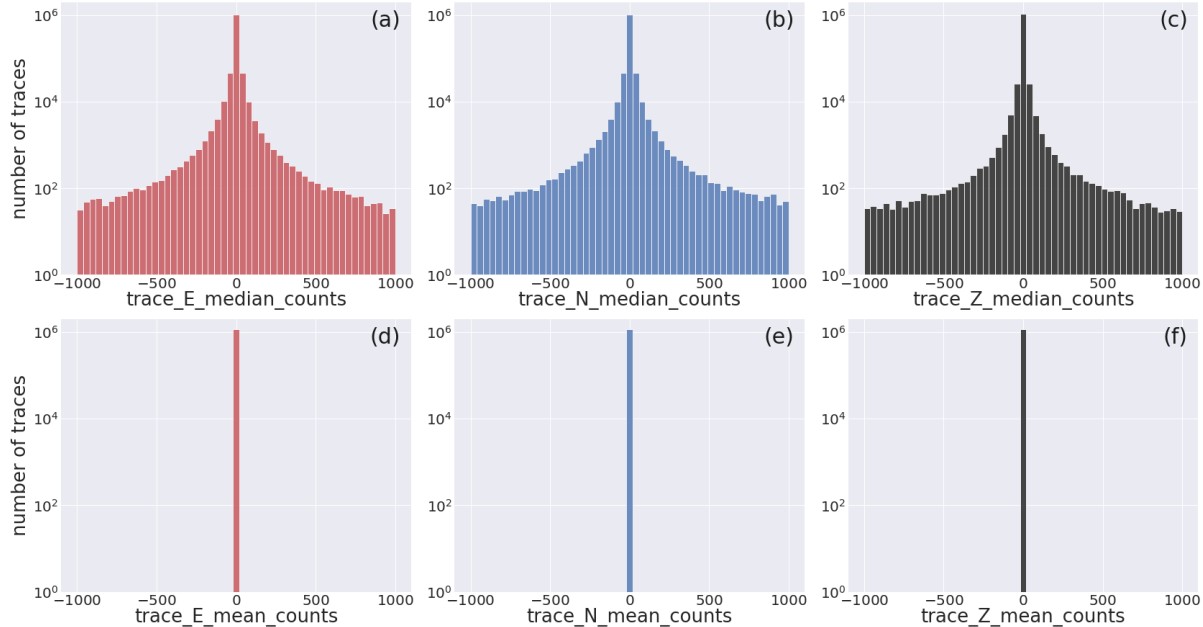

**Figure (8).** Close view of the histogram of the distribution of the median and mean values of the E, N and Z component earthquake waveform traces. The full distribution is shown in Fig. D2. Note that the mean values (bottom row) are shown to the sole scope of reference. The metadata names listed in Table 2 are used as labels for the specific metadata being represented.

## 2.4 Examples of event data traces

Some examples of the data traces are shown in Fig. 14, 15 and 16. The traces have been selected randomly according to certainly non-exhaustive criteria described in the figure caption using, as a guideline, the metadata distribution illustrated in Sect. 2.3.

In Fig. 14 we show the traces in counts of events recorded by the broadband instruments (HH channels). Specifically, the first three rows (Fig. 14a-i) show traces for different ranges of magnitude which, taken together, represent more than 80 % of the total HH traces. In the following two panel rows (Fig. 14j-o) we show examples of traces selected according to SNR and distance criteria that evidence that more than 65 % of the traces feature relatively high SNR (i.e., $\geq 10$) within the whole distance range covered by our data collection. The seismograms shown in the last panel row (Fig. 14p-r) provide some samples of recordings of the largest events ($M \geq 4$) where we found that $\sim 87\,\%$ feature SNR $\geq 10$.

To show how metadata can be used to isolate end members of the dataset, we focus next on examples of problematic traces. Although different criteria could have been used given the comprehensive set of metadata available, here for simplicity, we base our identification on: i.) the number of picks and detections resulting from application of the GPD and EQTransformer algorithms to isolate those traces likely containing more than a single event; ii.) the value of the SNR to identify poor quality noisy traces; iii.) the values of the trace median values which are expected to diverge from zero whenever the trace values

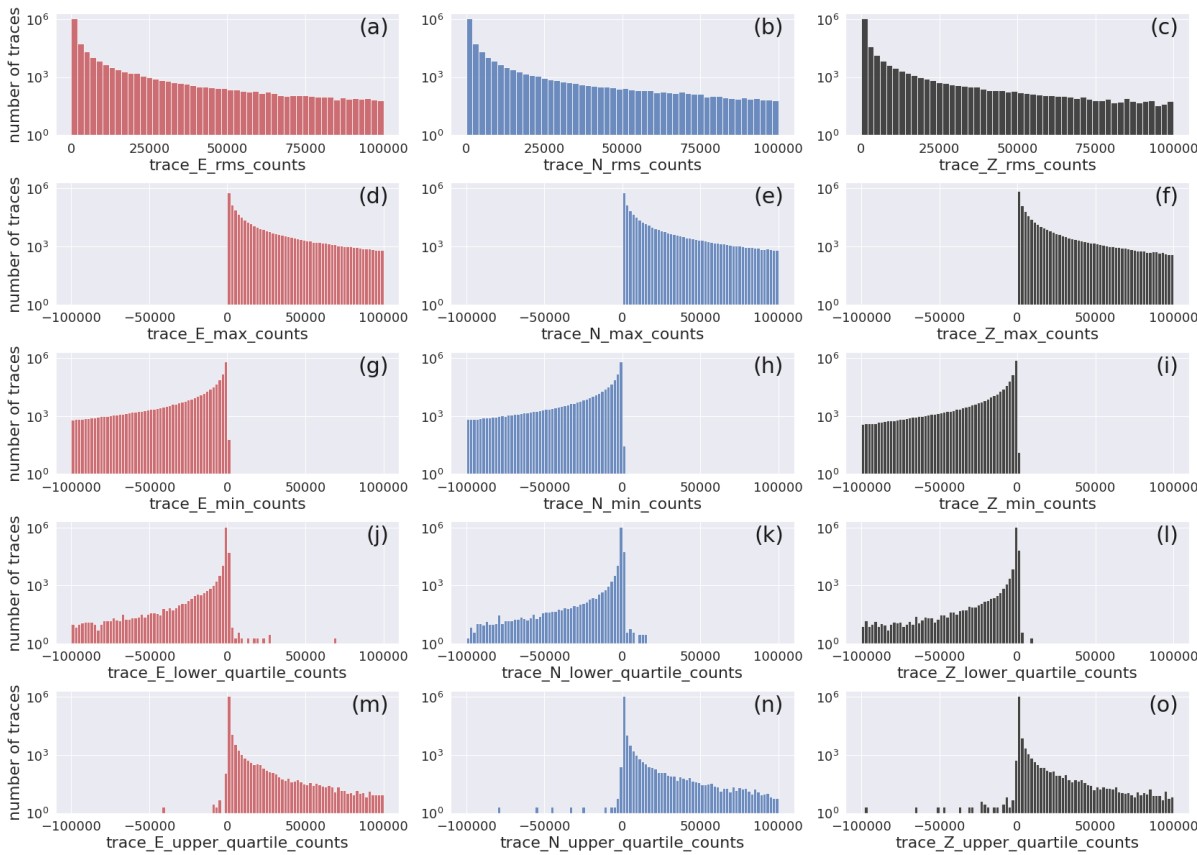

**Figure (9).** Histogram of the distribution of quality control metadata of the earthquake E, N, Z component waveform traces: rms, min, max, first and third quartile. The width of the bins is $2 \times 10^3$. The full distribution of values is provided in Fig. D3. The horizontal axis labels correspond to the metadata being represented which are listed in Table 2.

are unevenly distributed about the mean value as result of acquisition or processing problems; and iv.) the values of peak acceleration and velocity ground motion parameters. The user, depending on desiderata, can customize the selection criteria. In Table 3, we provide a basic quantification of the distribution of the relevant metadata shown in Fig. 15 and, in Table 4, we present the distribution of the values of the maximum horizontal ground acceleration and velocity expressed as % g and cm s$^{-1}$, respectively. Some of the values reported in these two tables are used for our trace selections.

In Fig. 15 (a-c) and Fig. 15 (d-f) we show some traces that have been selected from the HH channels according to the number of P- and S-wave onset picks greater than 3 detected through the application of the GPD and EQTransformer techniques, respectively. Based on the values reported in Table 3, the presence of these multiple event traces in the dataset is less than the 10 %. In Fig. 15 (g-i), we focus attention on the traces that feature SNR values on at least one component within the lowest 10 % of the dataset. These traces are good examples of noisy traces and low amplitude event signals. In Fig. 15 (j-l), we plot three traces for which the median values of all the three components fall within the two 10 % extremes. They represent about

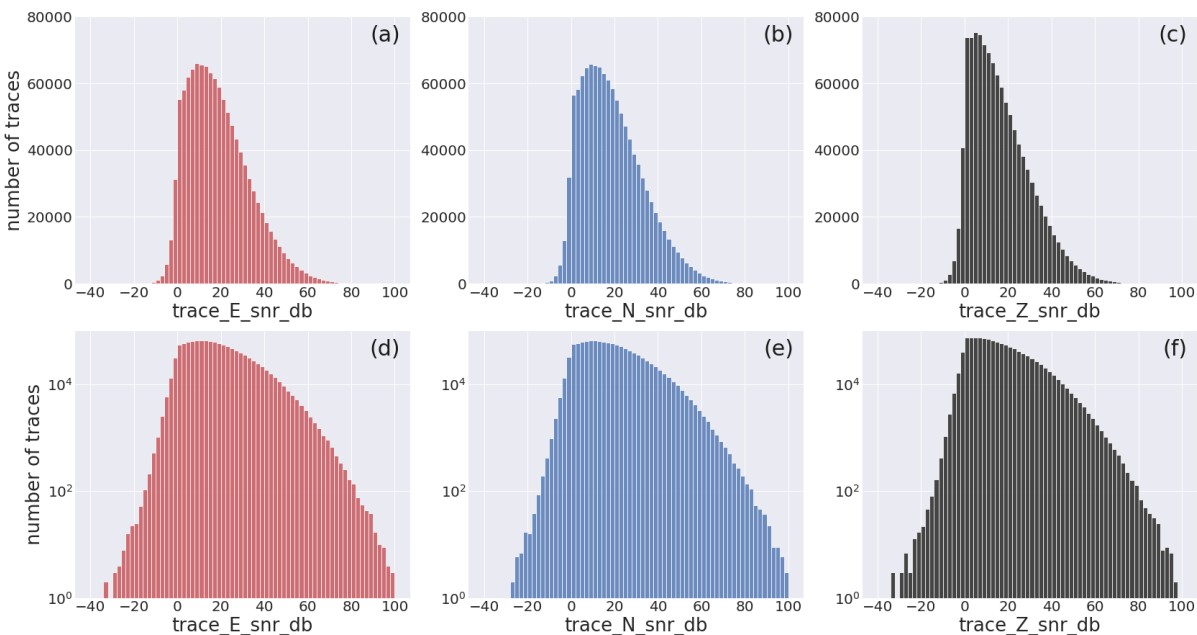

**Figure (10).** Distribution of the signal-to-noise ratio of the earthquake E, N, Z component waveform traces. The panels on the top row have linear y-axes, whereas those on the bottom are in logarithmic scale. The horizontal axis labels correspond to the metadata being represented which are listed in Table 2.

**Table (3).** Distribution according to different quantiles of selected metadata (cf. Table 2) for the HH channels of the event dataset.

| Metadata parameter-name | min | 10 % | 25 % | 50 % | 75 % | 90 % | max |
|---|---|---|---|---|---|---|---|
| trace_E_median_counts | -6.57e+06 | -22 | -6 | 0 | 6 | 22 | 3.03e+06 |
| trace_N_median_counts | -6.51e+05 | -22 | -6 | 0 | 6 | 22.5 | 2.5e+06 |
| trace_Z_median_counts | -7.63e+05 | -11.5 | -3 | 0 | 3 | 12 | 9.92e+05 |
| trace_E_snr_db | -25.5 | 2.31 | 7.29 | 15 | 25 | 35.4 | 95.4 |
| trace_N_snr_db | -26.9 | 2.3 | 7.27 | 15.1 | 25.1 | 35.5 | 95.8 |
| trace_Z_snr_db | -23.3 | 1.21 | 5.51 | 12.5 | 22.2 | 32.5 | 95.4 |
| trace_EQT_number_det. | 0 | 1 | 1 | 1 | 1 | 1 | 7 |
| trace_GPD_P_number | 0 | 0 | 1 | 1 | 1 | 2 | 13 |
| trace_GPD_S_number | 0 | 0 | 1 | 1 | 2 | 3 | 22 |

6 % of the entire HH channels dataset. In the bottom row of Fig. 15 (m-o), we show that excluding the very low 25 % of SNR values from the previous selection (Fig. 15 (j-l)) it is possible to select traces that do not suffer of particular problems.

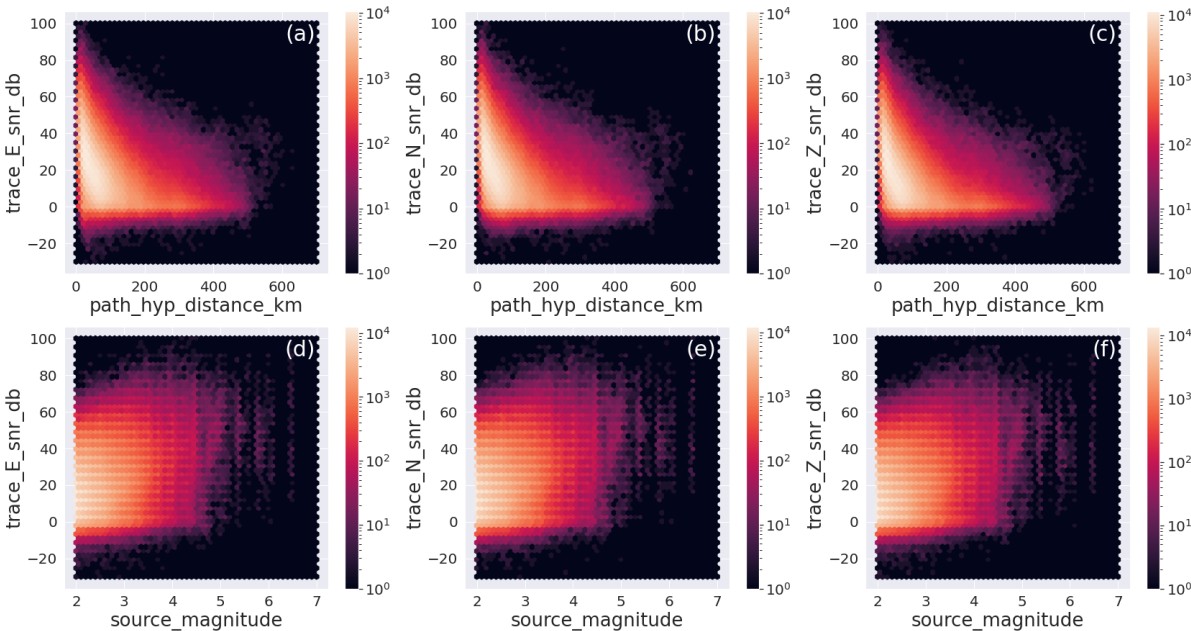

**Figure (11).** Hexbin representation of the distribution of the signal-to-noise ratio for the E, N and Z components of the earthquake dataset as function of hypocentral distance distance and magnitude. The metadata names listed in Table 2 are used as labels for the specific metadata being represented.

In particular, we see that just by selecting a higher threshold of SNR values, about 85 % of the first and last 10 % of the distribution of median values, the traces appear acceptable.

**Table (4).** Distribution according to different quantiles of IMs selected metadata for the HH, EH and HN channels

| Metadata parameter-name | 10 % | 25 % | 50 % | 75 % | 90 % | max |
|---|---|---|---|---|---|---|
| `trace_pga_perc` (HH) | 0.000219 | 0.000506 | 0.00148 | 0.00532 | 0.0201 | 57.6 |
| `trace_pgv_cmps` (HH) | 7.81e-05 | 0.000158 | 0.000413 | 0.00139 | 0.00499 | 58.6 |
| `trace_pga_perc` (EH) | 0.000341 | 0.000927 | 0.00328 | 0.0139 | 0.0559 | 71 |
| `trace_pgv_cmps` (EH) | 0.000174 | 0.00035 | 0.000882 | 0.00302 | 0.0114 | 55.6 |
| `trace_pga_perc` (HN) | 0.000329 | 0.000875 | 0.00328 | 0.0141 | 0.0509 | 77.8 |
| `trace_pgv_cmps` (HN) | 0.000392 | 0.000582 | 0.00134 | 0.00501 | 0.0153 | 59.1 |

In Fig. 16, we show the instrument corrected traces randomly chosen in groups of 6 for each channel. The traces drawn from the entire dataset belong to the 75 % with the largest values of the maximum horizontal acceleration (i.e., second, third and fourth quartile of the value distribution, cf. Table 4). The total of traces satisfying this criterion amounts to more than 860,000

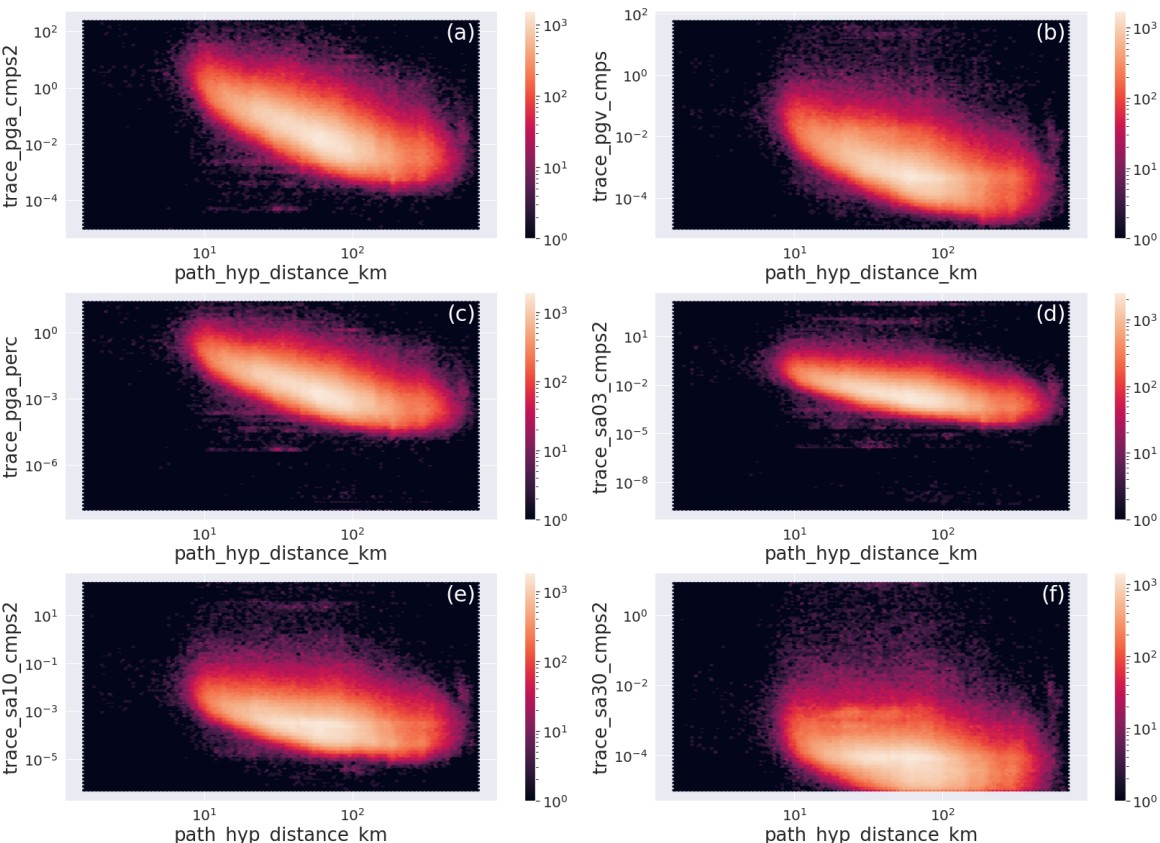

**Figure (12).** Hexbin plot of the distribution of the intensity measures (IMs) with hypocentral distance of the earthquake dataset for the $M \geq 2$ earthquakes. The units are $km$ along the horizontal axis in all panels, and, along the vertical axis, $cm\ s^{-2}$ in panels (a,d-f), $cm\ s^{-1}$ in panel (b), and $\%$ g in panel (c). The metadata names listed in Table 2 are used as labels for the specific metadata being represented.

3C traces. Application of the instrument transfer function appears to be generally successful without introduction of particular side effects with the exception of some amplification of the very low frequencies for some very low amplitude traces of the EH channels (e.g., panels (h,k) in Fig. 16). This effect results from our choice to bandpass filter all the traces channels in the same frequency range: this has the negative effect of boosting the low frequencies of the narrower band EH channels although it can be promptly removed by high-pass filtering. Overall, the quality of the ground motion units dataset can be considered of satisfactory quality to perform analysis of ground motions.

## 3  Noise

Noise is generated by many different sources such as ocean waves, wind, traffic, instrumental noise, electrical noise, etc. and its suppression in earthquake recordings represent a long standing objective (Zhu et al., 2019b, and references therein). The in-

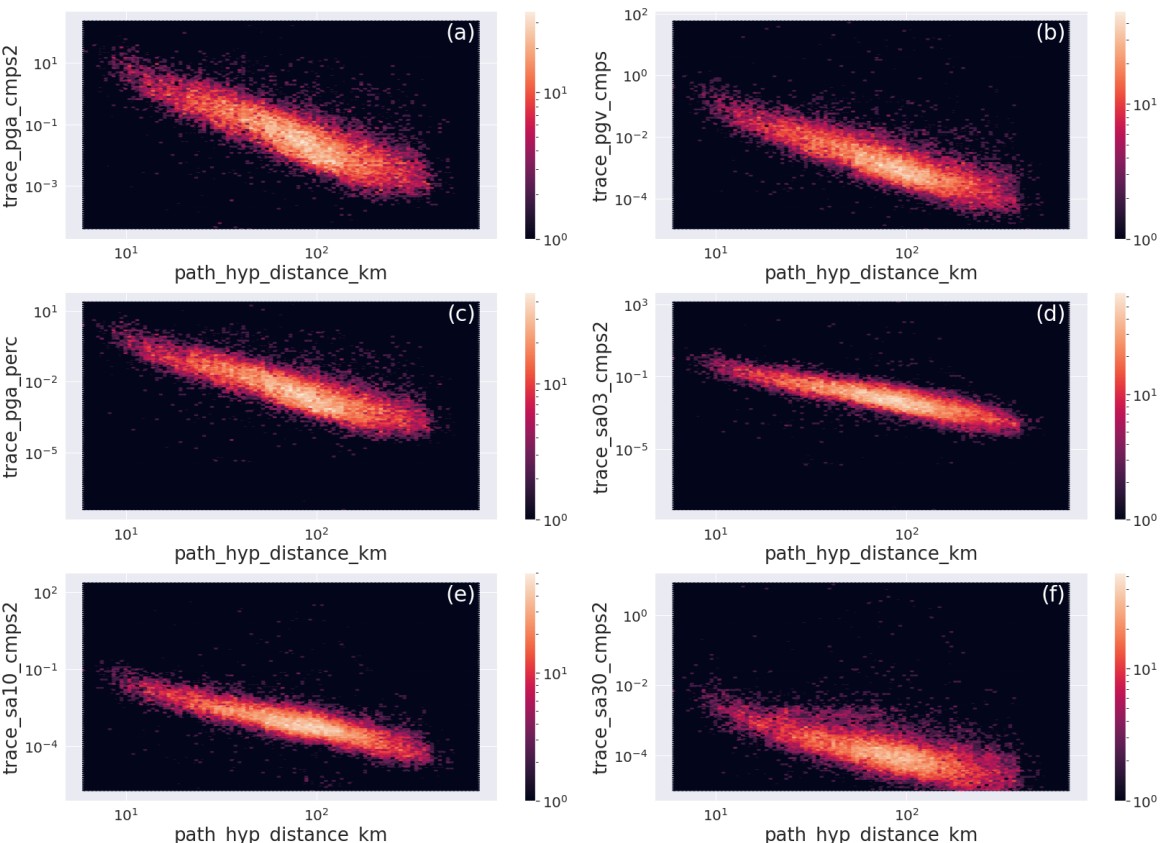

**Figure (13).** Hexbin plot of the distribution of the intensity measures (IMs) with hypocentral distance for $M = 3$ earthquakes. The units are $km$ along the horizontal axis in all panels, and, along the vertical axis, $cm\ s^{-2}$ in panels (a,d-f), $cm\ s^{-1}$ in panel (b), and % g in panel (c). The metadata names listed in Table 2 are used as labels for the specific metadata being represented.

clusion of noise data in a dataset like INSTANCE is thus important because it provides information on the noise characteristics of the individual stations in the absence of earthquake generated signal. ML models can reveal to be effective for noise removal or, in a classification analysis, for improving the detection of earthquakes. The noise data have been assembled starting from the stations gathered in the event selection stage described above.

5   **3.1   Data Preparation**

Starting from the entire catalogue consisting of more than 300,000 events (Table 1), we first identified 600 s long time windows free of any earthquake. Secondly, we obtained the operational times of acquisition of each station. The third step consisted of identifying the 120 s time windows to be included in the dataset for each station and channel. This was achieved by intersecting the time window series obtained in the previous two stages (i.e., the event free windows and the periods of station acquisition).

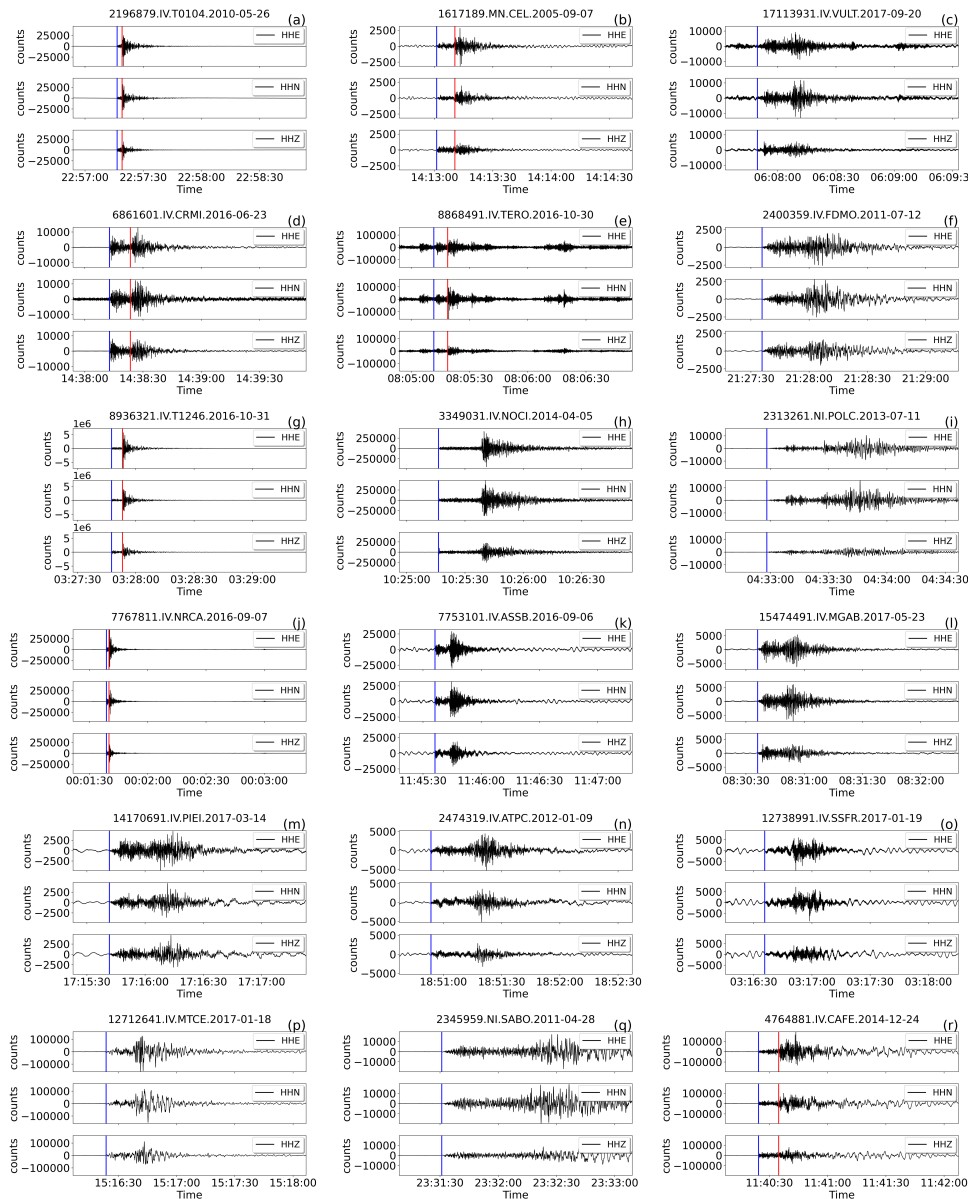

**Figure (14).** Example of earthquake waveforms of the broadband HH channels contained in INSTANCE randomly drawn according to different criteria based on the metadata provided in Table 2. Each row contains three, randomly selected, 3C traces based on the following criteria: (a-c) earthquakes $2 \leq M < 3$ (66.8 % of the total of the HH channels); (d-f) earthquakes $3 \leq M < 4$ (13.5 %); (g-i) earthquakes $M \geq 4$ (2.0 %); (j-l) earthquakes `trace_E_snr_db` $\geq 10$ and `path_ep_distance` $< 100$ km (55.0 %); (m-o) earthquakes `trace_E_snr_db` $\geq 10$ and `path_ep_distance` $\geq 100$ km (10.8 %); (p-r) earthquakes $M \geq 4$ and `trace_E_snr_db` $\geq 10$ (1.7 %). The arrival times of P- and S-wave onsets (i.e., `trace_[P,S]_arrival_time`) are shown by blue and red vertical lines, respectively.

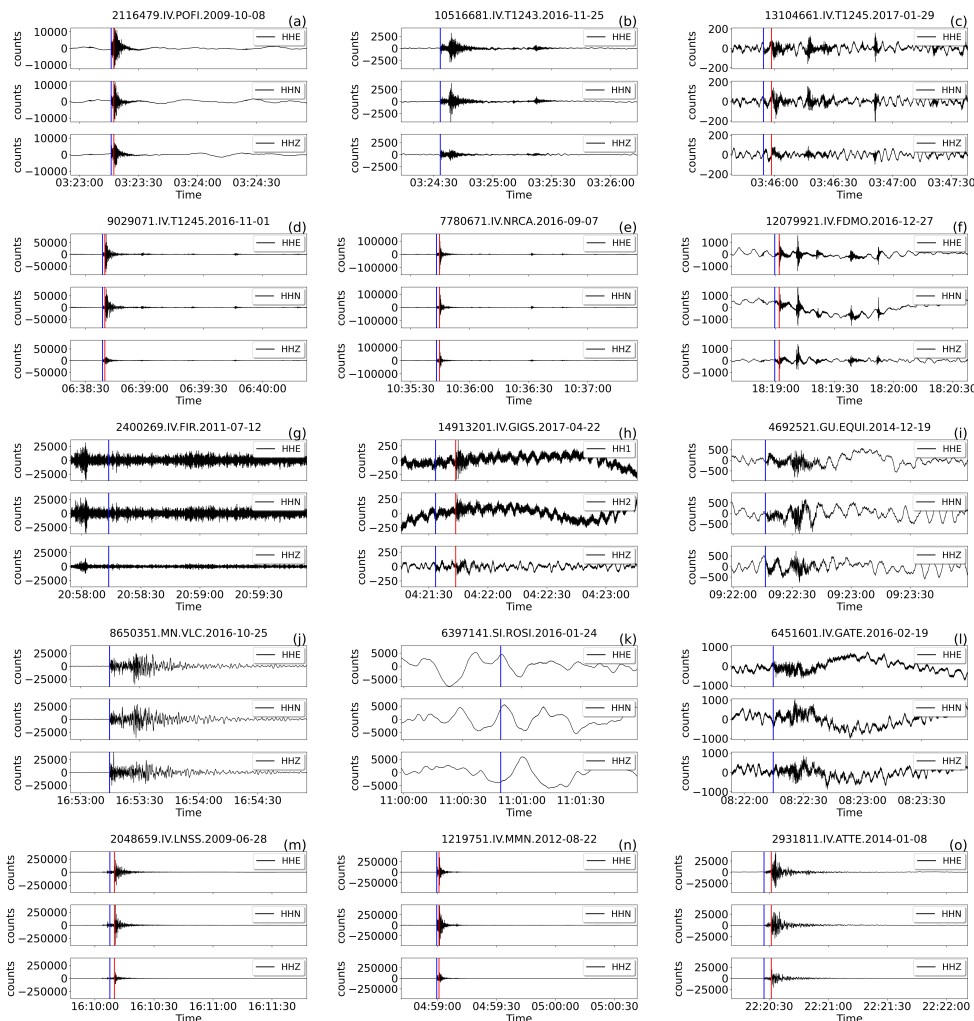

**Figure (15).** Example of randomly selected "problematic" earthquake waveforms of the broadband HH channels. Each row contains three, randomly selected, 3C traces drawn according to the following criteria based on the metadata listed in Table 2: (a-c) traces with `trace_GPD_[P,S]_number` > 3 (7.96 % of the total of the HH channels); (d-f) traces with `trace_EQT_number_detections` > 3 (0.38 % of the total of the HH channels); (g-i) traces `trace_[ENZ]_snr_db` with at least one component in the 10 % quantile (18.10 % of the total of the HH channels); (j-l) traces with all `trace_[ENZ]_median_counts` either in the first 10 % or the last 10 % quantiles (5.90 % of the total of the HH channels); (m-o) traces with `trace_[ENZ]_median_counts` either in the first 10 % or the last 10 % quantiles and corresponding `trace_[ENZ]_snr_db` excluded from the first quartile (5.06 % HH dataset). The arrival times of P- and S-wave onsets (i.e., `trace_[P,S]_arrival_time` ) are shown by blue and red vertical lines, respectively.

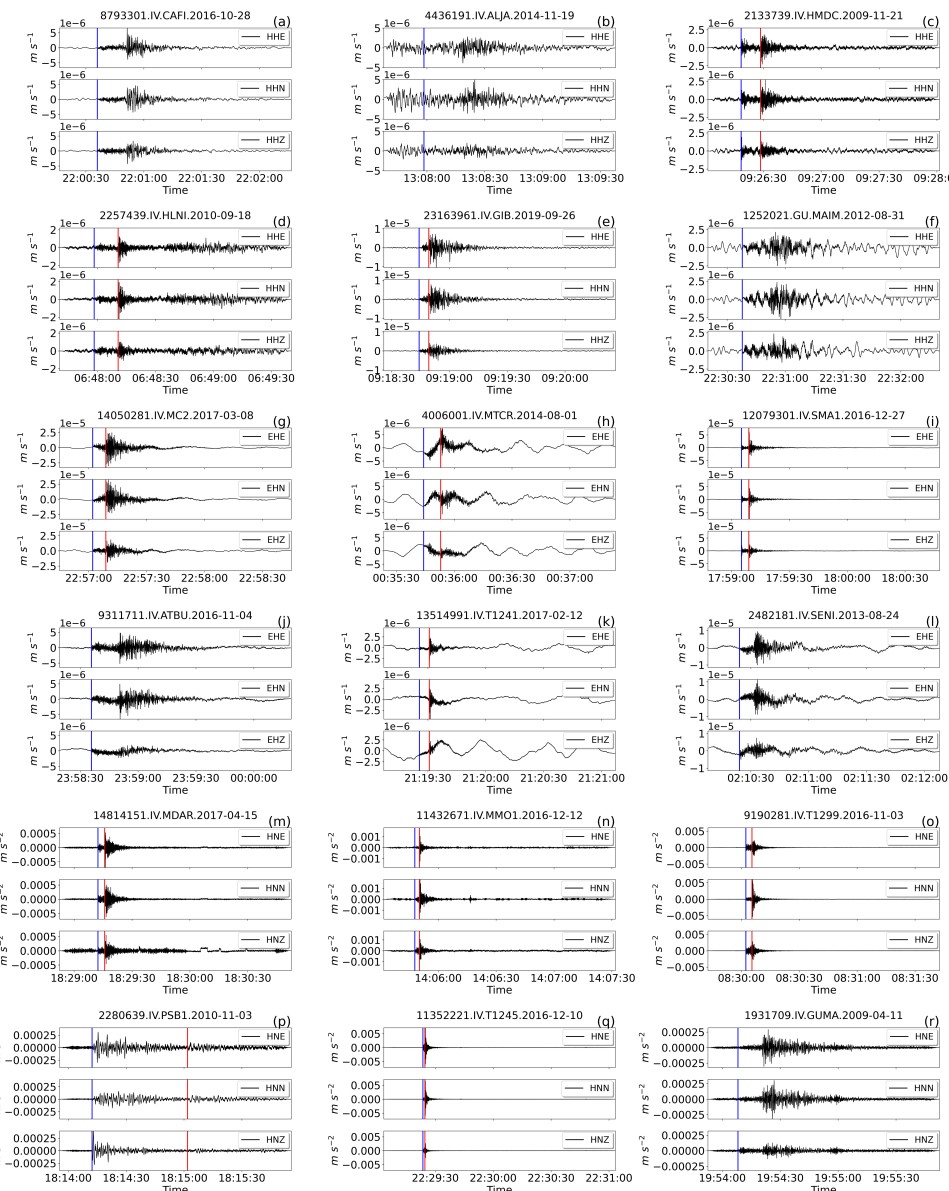

**Figure (16).** Example of randomly selected event waveforms in ground motion physical units of the HH, EH and HN channels in INSTANCE. The traces are representative of 75 % of the data and belong to the second, third and fourth quartiles of each channel. Each row contains three, randomly selected, 3C traces drawn according to the following criteria based on the metadata listed in Table 2 and the quantile values provided in Table 4: (a-f) HH traces with `trace_pga_perc` > 5.1e-4 % g; (g-l) EH traces with `trace_pga_perc` > 9.3e-4 % g; (m-r) HN traces with `trace_pga_perc` > 8.7e-4 % g; The arrival times of P- and S-wave onsets (i.e., `trace_[P,S]_arrival_time` ) are shown by blue and red vertical lines, respectively.

It follows that the adopted procedure does not entail the selection of the same time window for multiple stations. For stations acquiring more than one channel type (e.g., HH and HN), noise windows for all the channels were identified and downloaded. The resulting total number of noise trace windows is 132,288 corresponding to about 10 % of the total number of traces of INSTANCE. We note also that this procedure does not preclude the presence of noise traces that include energy from regional and teleseismic events.

## 3.2 Metadata description

The 46 metadata elements (Table 2) used for the noise data selection include for each 3C waveform trace an identifier based on the start time, the *station* parameters, the *trace* quality control that include the automatic picks and event detection obtained using the GPD and EQTransformer procedures. These picks provide potential insights on whether any earthquake not catalogued in the INGV bulletin might be present in the selected time windows.

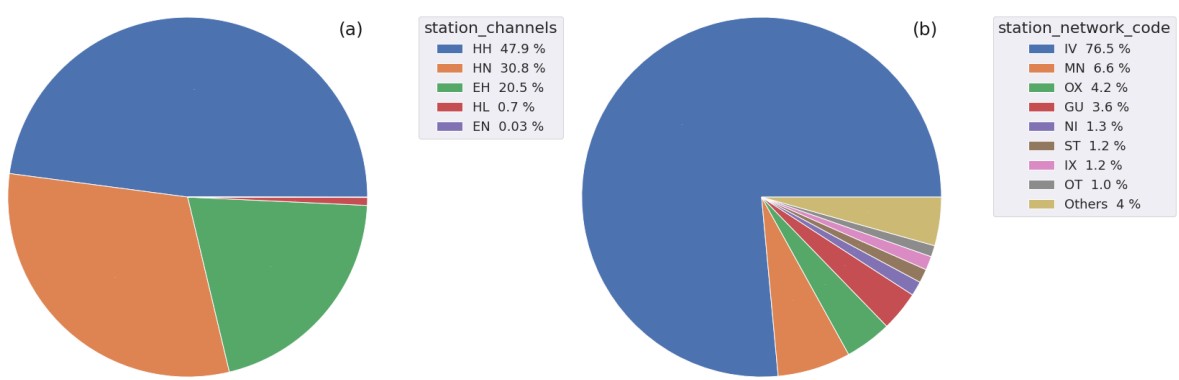

**Figure (17).** Pie diagrams summarizing the distribution of the channels (a) and the data contributing networks (b) of the noise dataset. The full list of station_network_code with % < 1 collected in Others in decreasing order is SI, YD, 3A, XO, ZM, BA, AC, HL, TV, RF.

In Fig.17 we show the channel subdivision of the downloaded noise together with the networks the stations belong to. In Fig. 18 and 19, in analogy with what presented for the event data, we present the *trace* characteristics of the metadata. The trace_[E,N,Z]_mean_counts and trace_[E,N,Z]_median_count provide an outlook on the distribution of the mean and median values and, likewise the same parameters extracted from the event traces, could be used to identify high quality data. The histograms of the trace_[E,N,Z]_rms_counts noise values fall mainly in the range of values from 0 to 2000 counts with similar peak values for either trace_[E,N,Z]_max_counts or trace_[E,N,Z]_min_counts. This all would suggest that the gathered noise traces are of fairly good quality responding to the expectation of traces characterized by amplitudes with small number of counts.

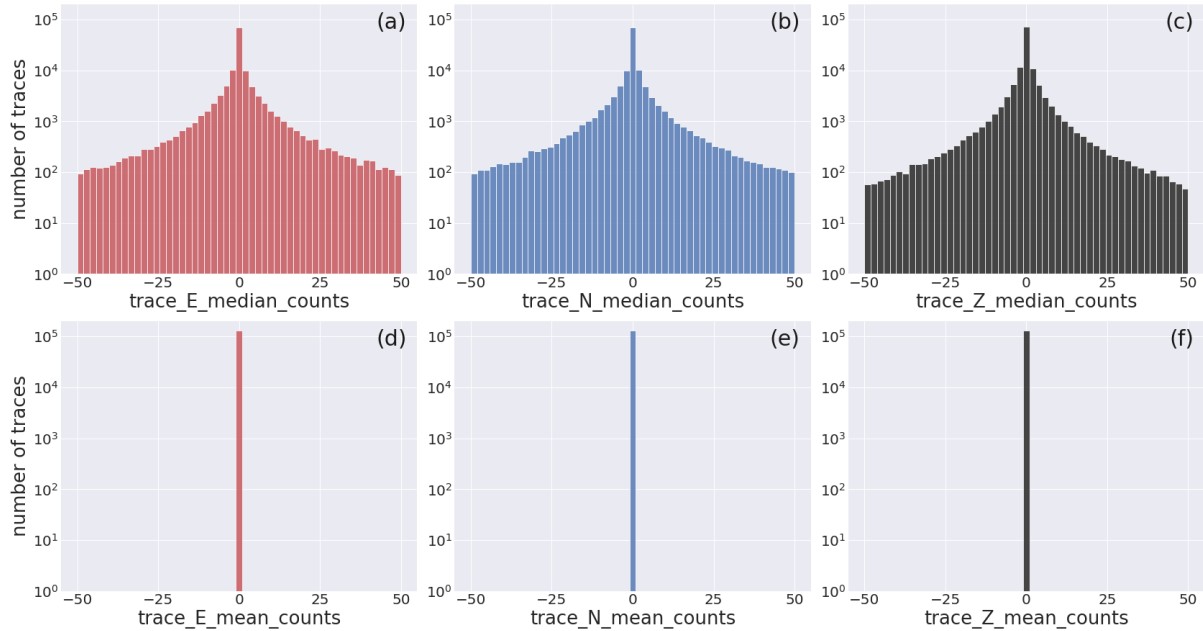

**Figure (18).** Close view of the histogram of the distribution of the median and mean values of the E, N and Z component noise waveform traces. The full distribution is shown in Fig. D5. Note that the mean values (bottom row) are shown to the sole scope of reference. The metadata names listed in Table 2 are used as labels for the specific metadata being represented.

### 3.3 Examples of noise data traces

Examples of the noise traces are shown in Figure 20. To perform the selection, we have used the distribution of the trace *rms* values that is provided in Table 5.

In the top two rows of Fig. 20, there are shown some examples of events detected using the GPD and EQTransformer algorithms on the EH channels. As it was the case with the event dataset, the noise traces also contain undetected events although their number according to our analysis seems rather small especially for the earthquakes detected by EQTransformer. This result gives us good confidence that the noise traces are for the great majority free of earthquake events. The following rows of Fig. 20 provide waveform samples drawn from the 90 % of the dataset (panels g-i and m-o) for the HH and EH channels, respectively. Both sets of panels exemplify some of the features of the great majority of the noise data. In contrast, the panels (j-l) and (p-r) have been chosen to show what could be considered traces exceeding noise values or that contain finite duration events of uncertain origin.

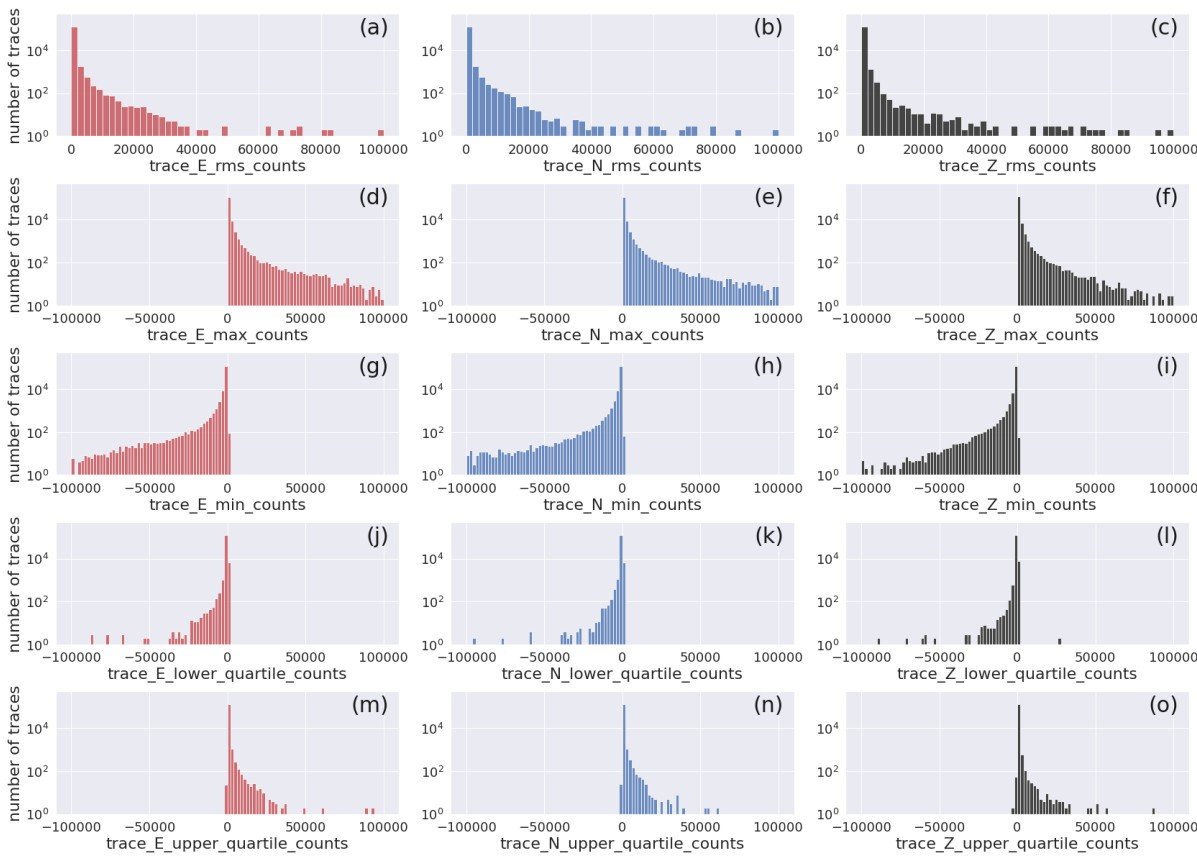

**Figure (19).** Histogram of the distribution of the quality control metadata of the noise E, N, Z component waveform traces: rms, min, max, first and third quartile. The width of the bins is $2 \times 10^3$. The full distribution of values is provided in Fig. D6. The horizontal axis labels correspond to the metadata being represented which are listed in Table 2.

## 4 Discussion

The primary objective of this work has been to assemble a benchmark dataset consisting of seismic waveforms and associated metadata. It has been designed to be used for the analysis of earthquakes in Italy (and neighboring areas) using ML techniques and it could prove useful for ML analysis also elsewhere in other active tectonic regions by adopting transfer learning methodologies (Jozinović et al., 2021). The dataset consists of three HDF5 volumes — raw and instrument removed event traces, and raw noise traces — and of the associated metadata.

The selection of the waveform traces to be included was based on the availability of low ($\leq 1$ s) P- and S-phase location residual times and large location weights taken from the *preferred* solutions listed in the INGV bulletin. To counteract the Gutenberg-Richter power law which affects the compilation of seismological datasets targeting ML analysis applications, attention was paid towards assembling a dataset that was not completely skewed by a large number of small magnitude earth-

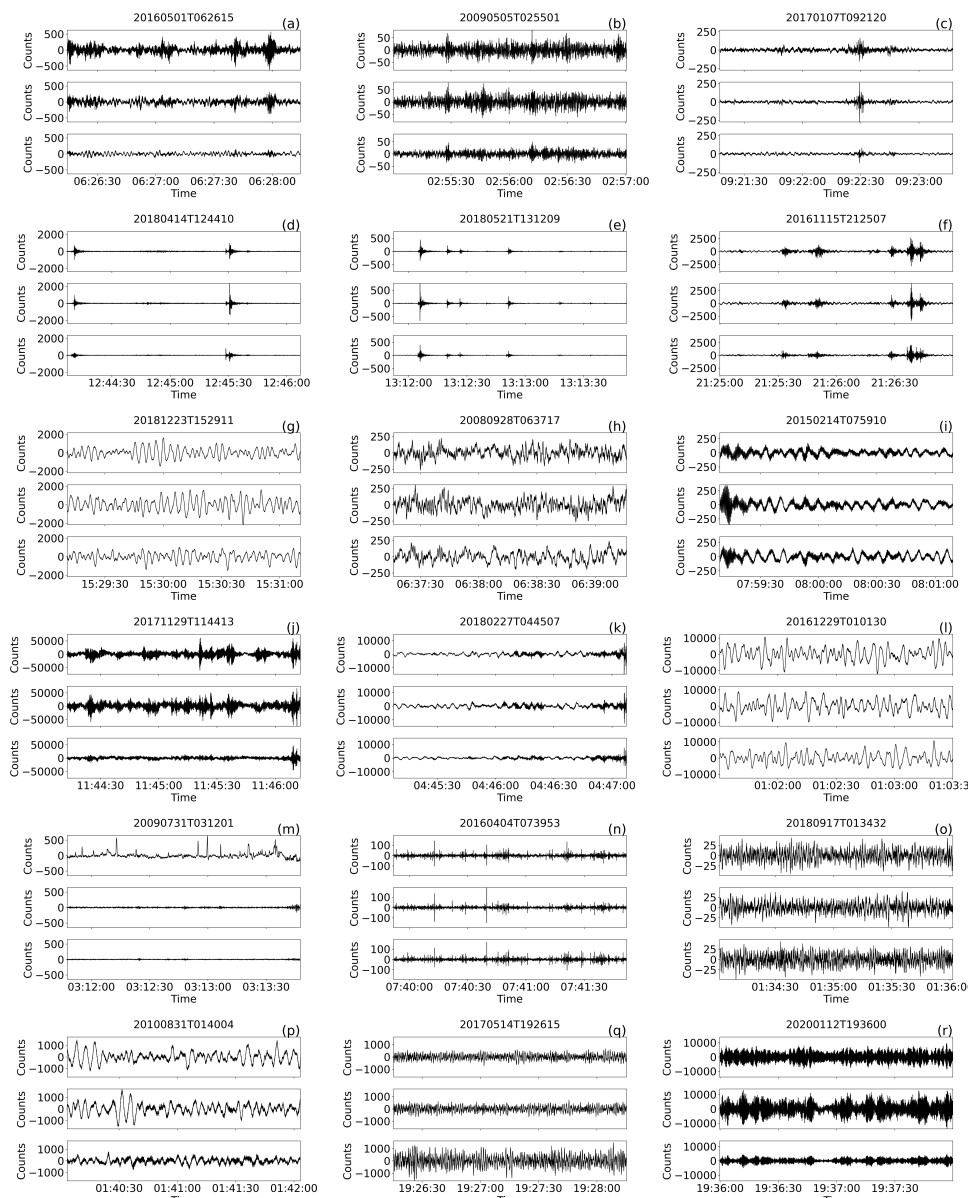

**Figure (20).** Example of randomly selected noise waveforms of the HH and EH channels contained in INSTANCE. The traces are drawn randomly according to different criteria based on the metadata provided in Table 2 and on the quantile values listed in Table 5. Each row contains three, randomly selected, 3C traces drawn according to the following criteria: (a-c) `trace_GPD_[P,S]_number` > 3 (11.6 % of the total of the EH channels); (d-f) `trace_EQT_number_detections` > 3 (0.13 % of the total of the EH channels); (g-i) all the `trace_[E,N,Z]_rms_counts` < [1013,1071,793] (86.31 % of the total of the HH channels); (j-l) any of the `trace_[E,N,Z]_rms_counts` > [1013,1071,793] (13.69 % of the total of the HH channels); (m-o) all the `trace_[E,N,Z]_rms_counts` < [327.1,332,307] (86.36 % of the total of the EH channels); (p-r) any of the `trace_[E,N,Z]_rms_counts` > [327.1,332,307] (13.64 % of the total of the EH channels);

**Table (5).** Distribution according to different quantiles of selected noise metadata (cf. Table 2) for the HH and EH channels.

| Metadata parameter-name | 10 % | 25 % | 50 % | 75 % | 90 % | max |
|---|---|---|---|---|---|---|
| `trace_E_rms_counts` (HH) | 52.79 | 101.6 | 205 | 447.9 | 1013 | 1.919e+07 |
| `trace_N_rms_counts` (HH) | 53.47 | 102 | 207.3 | 465.8 | 1071 | 1.902e+07 |
| `trace_Z_rms_counts` (HH) | 44.68 | 85.42 | 166.3 | 364 | 793.1 | 9.986e+05 |
| `trace_EQT_number_det.` (HH) | 0 | 0 | 0 | 0 | 0 | 5 |
| `trace_GPD_P_number` (HH) | 0 | 0 | 0 | 0 | 1 | 31 |
| `trace_GPD_S_number`(HH) | 0 | 0 | 0 | 1 | 2 | 24 |
| `trace_E_rms_counts` (EH) | 7.53 | 22.92 | 58.29 | 141.8 | 327.1 | 7.54e+05 |
| `trace_N_rms_counts` (EH) | 7.864 | 22.88 | 57.65 | 140.9 | 332.6 | 2.913e+05 |
| `trace_Z_rms_counts` (EH) | 5.639 | 18.44 | 50.09 | 119.8 | 307.1 | 6.236e+05 |
| `trace_EQT_number_det.` (EH) | 0 | 0 | 0 | 0 | 0 | 5 |
| `trace_GPD_P_number` (EH) | 0 | 0 | 0 | 1 | 2 | 23 |
| `trace_GPD_S_number` (EH) | 0 | 0 | 0 | 2 | 4 | 26 |

quakes. To this end, we included all the traces available of the larger size earthquakes and then we decreased progressively the number of smaller size earthquakes and associated traces. Our effort, however, trades-off with the need of assembling a dataset that is sufficiently large for ML purposes. The distribution of the selected traces shown in Fig. 3 and Fig. 4 according to magnitude, distance and focal depth allows the users to make the appropriate choices for their purposes even though we recognize that the achievement of the sought balanced distribution remains difficult. Other data selection criteria could have been used (e.g., select all the data acquired within distance ranges depending on earthquake magnitude) but the (un)balanced magnitude and distance distribution would have persisted. Thus, given the criteria adopted it is pleonastic to remark that this dataset is not designed for studies addressing the earthquake magnitude power-law distribution (e.g., the b-value). Similarly, although the dataset contains an average of 21 traces per earthquake, it may not be optimal for dedicated earthquake relocation studies.

Our criterion, based on the available high-quality P- and S-phases with low location residual times, is expected to provide a large number of traces with distinct earthquake signal and high SNR ratios. The distribution of SNR values shown in Fig. 10 and 11 and in Table 3, and the example seismograms shown in Fig. 14(j-r) and Fig. 15(m-o), appear to confirm our choices.

The selection of 120 s trace length time window is longer than those made by other authors for analogous benchmark datasets (e.g., Mousavi et al., 2019; Magrini et al., 2020). This relatively long time window was required, however, because we sought to include the entire seismicity occurring in Italy that spans from very shallow to very deep (Fig. 3). Unfortunately, this long window trades-off with a higher probability of including earthquakes close in time that had not been reported in the INGV catalogue. For this reason, we carried out also a (preliminary) automatic picking and earthquake detection analysis using two well established recent ML techniques (GPD and EQTransformer; Ross et al., 2018a; Mousavi et al., 2020) to possibly isolate

those traces that include multiple events. The results of this analysis summarized in Table 3 would indicate that about 90 % of the event data contain only one earthquake according to the EQTransformer analysis whereas the GPD analysis returned some slightly higher numbers of P- and S-phase detections.

Our metadata for the earthquake part of the dataset consist of 115 parameters. They are subdivided into three main classes plus one additional class derived from the previous ones. This is a rather rich set of parameters that can be used either to select subsets of the dataset, or as additional features to rely on when developing ML models, or as labels in supervised ML analysis, or for unsupervised ML applications. In addition, the metadata could be used by themselves for specific studies (e.g., seismic velocity model regionalization, travel time tomography, ground motion prediction models, local site corrections, ...).

Earthquake data gathered by seismic instruments and streamed in realtime to earthquake monitoring centers or preserved within archives can suffer from problems of different nature (e.g., sensor, data logger, equipment installation, data transmission, and processing among the most common). Thus, the compiled dataset could be useful for the development of robust techniques of analysis and this is one main reason for including several *trace* quality parameters as metadata since they can help the user to identify the possibly "faulty" records which can be then either removed or included to train the ML model just to "learn" them. This approach may seem to contradict one of the main purposes of compiling a high-quality dataset and it may also be an obstacle when attempting to reveal deep information therein but the expectation is that, by including all the data together, the rich set of metadata leaves the users enough freedom to identify the "good" data for their purposes. It is worthwhile to mention that Yeck et al. (2020) have found that inclusion of only "good", high SNR trace data during training of various body waves resulted in lower performance when applied to real-time pick data.

The INSTANCE dataset includes intensity measures (i.e., PGA, PGV, spectral accelerations) obtained after deconvolution of the instrument response performed automatically and possibly affected by digital signal processing problems induced, for example, by the presence of abnormal drifts and spikes. Given the difficulty to verify the quality of all the individual processed traces, the availability of a rich set of trace metadata can be useful (again) to detect the faulty traces.

The example traces drawn randomly from the dataset that we have presented in Figs. 14, 15, 16 and 20 provide some evidence of the characteristics of the traces contained in the dataset and how they can be promptly selected through the provided metadata. Although the great majority of the data appear to be of very good quality, we are also aware that low quality data are almost inevitable to occur. Inspection of the waveform traces by using other selection criteria than those shown here, and of the IM metadata (cf. Fig. 12) give us, however, good confidence on an overall good quality of the dataset.

The INSTANCE data collection assembles for the first time a very large amount of earthquake and noise data throughout Italy. If on one side this might seem a limitation when compared to other recent data collections like STEAD and LEN-DB that have gathered data globally, on the other hand, the dataset can be considered a representative subset of the seismicity in Italy and neighbouring areas. The dataset equals to more than 43,000 hours of continuous event and noise data and associated metadata with an average of 21 3C traces per earthquake. To the purpose of comparison, in Table 6, we summarize the main features of the currently available seismological datasets assembled for ML analysis. As noted above, the main features that distinguish INSTANCE from the other datasets are the number of metadata for both earthquakes and noise traces and the average number of traces per event. In addition, the dataset provides a generally large number of traces for each recording site

making the dataset suitable for quite diversified target studies. The dataset is also unique since it is the only one (yet) to provide the waveform traces in both digital counts and physical units. In this context, the set of parameters provided by INSTANCE spans both specific seismological parameters like P and S arrival times, fault plane and moment tensor solutions, and also peak ground motion parameters in physical units (e.g., PGA, PGV) which can be used for studies that target the estimation of the ground shaking (e.g., shakemaps).

**Table (6).** Comparison between INSTANCE and other published seismic waveform datasets. It was not possible to retrieve some attributes of the original SCEDC dataset since it is available as different subsets extracted for specific application (list available at https://scedc.caltech.edu/data/deeplearing.html). [1] INSTANCE, doi:10.13127/instance. [2] STEAD, doi:10.1109/ACCESS.2019.2947848. [3] SCEDC, https://scedc.caltech.edu/data/deeplearning.html. [4] LenDB, doi:10.5281/zenodo.3648232. [5] ConvNetQuake_INGV (CNQ_INGV), doi:10.5281/zenodo.5040865. [6] NEIC, doi:10.5066/P9OHF4WL. [7] D: digital; P: physical. [8] L: local; R: regional; G: global. [9] BB: broadband; SM: strong motion; SP: short period.

| | INSTANCE[1] | STEAD[2] | SCEDC[3] | LEN-DB[4] | CNQ_INGV[5] | NEIC[6] |
|---|---|---|---|---|---|---|
| Metadata (events) | 115 | 35 | – | 14 | 6 | 5 |
| Metadata (noise) | 46 | 8 | – | 7 | 2 | – |
| Trace length (s) | 120 | 60 | 4,6 | 27 | 50 | 60 |
| Units[7] | D, P | D | D | P | P | D |
| Events | 54,008 | $\sim 450,000$ | 273,882 | 304,874 | 6,213 | 136,716 |
| Traces (events) | 1,159,249 | 1,050,000 | – | 629,095 | 22,046 | – |
| Traces (noise) | 132,288 | $\sim 100,000$ | – | 615,847 | 12,543 | – |
| Receivers | 620 | 2,613 | – | 1,487 | 26 | 2361 |
| Average receivers per event | 21 | 2 | – | 2 | 4 | – |
| Duration in hours (events) | 38,641 | $\sim 17,500$ | – | 4,718 | 306 | – |
| Duration in hours (noise) | 4,409 | $\sim 1,700$ | – | 4,618 | 174 | – |
| Epicentral distance range (km) | $< 620$ | $< 350$ | $< 360$ | $< 189$ | $< 19,310$ | $< 10,000$ |
| Magnitude range | $0 - 6.5$ | $0 - 7.9$ | $-0.81 - 7.3$ | $0.4 - 7.1$ | $3 - 9.1$ | $1 - 8.3$ |
| Sampling rate (Hz) | 100 | 100 | 100 | 20 | 20 | 40 |
| Storage size (GB) | 331.2 | 91.4 | – | 18.4 | 0.9 | $\sim 51$ |
| Focal mechanism | 527 | 6,200 | – | – | – | – |
| Event type[8] | L, R | L | L, G | L | L, R, G | L, R, G |
| Data type[9] | BB, SM, SP | BB, SM, SP | BB, SM | – | BB | BB, SP? |

In summary, the dataset features strengths such as the prompt availability of a large number of records assembled within a ready-to-use data volume that can be certainly considered representative of the whole waveform data archive of the INGV ORFEUS-EIDA node and that can be used for many diverse studies. In our opinion, the strengths of providing a diversified

set of data outnumber the weaknesses and the latter ones can be isolated, and their negative contribution reduced through the exploitation of the very rich set of metadata.

# 5   Applications

To the purpose of describing the range of possible applications of INSTANCE we follow the basic exposition schema adopted by Mousavi et al. (2019) for the STEAD benchmark dataset. These authors addressed four main areas in which benchmark datasets can prove very effective for improving seismological knowledge and seismic monitoring operational activities: earthquake trace denoising, earthquake detection and onset picking, classification/discrimination and direct earthquake characterization.

The seismic noise level at a station is frequency dependent and derives from many factors such as types of equipment, installation, meteorological conditions, anthropic generated noise, geography, season, and time of day (McNamara and Buland, 2004). Seismic trace denoising enhances the SNR that is crucial to lowering the magnitude detection level of earthquake catalogs and, by so doing, increase the number events detected. Analogously, denoising can be relevant to pre-process seismic traces when performing ambient noise cross-correlation analysis (e.g. Baig et al., 2009), or for detecting speed-of-light changes of the gravitational field (Vallée et al., 2017), or for the analysis of seismic data acquired in urban areas (e.g. Parolai, 2009) just to mention a few among many applications. ML techniques seem very promising to address the reduction of noise in seismic data. For example, Zhu et al. (2019b) (and references therein for a list of applications in applied geophysics and seismology) have proposed a denoising/decomposition method, DeepDenoiser, based on a deep neural network which is based on the adoption of signal and noise masks which are then used to effectively decompose the input data into a signal of interest and noise. The technique has been tested against a dataset composed of broadband recordings of the North California Seismic Network which is similar to the data of INSTANCE. The adoption of unsupervised machine-learning method has been instead advocated by Chen et al. (2019) who have proposed it in combination with an autoencoder algorithm that adaptively learns the features from the raw noisy seismological datasets and use the sparse constraint to suppress the learned trivial features that may be associated with partial noise component. They apply the technique to the waveform stacked data used in Shearer (1991) and similar stacks can be promptly prepared using INSTANCE at the local/regional scale and apply the denoising technique accordingly.

Earthquake detection (including phase picking), discrimination and rapid characterization represent main pillars of seismic monitoring and surveillance. During their lifetime, operational seismic centers alternate calm periods characterized by low levels of seismicity, in which it can become of relevance the detection of even the smallest possible events to delineate the activation of often hidden tectonic structures, to paroxysmal periods starting with significant earthquakes and followed by hundreds or thousands of aftershocks felt by people. To ameliorate the response of the centers in both these extreme cases, we find that the INSTANCE dataset can be of importance to calibrate and benchmark methodologies for i.) phase onset picking and earthquake detection methods to lower the magnitude detection level (e.g., Ross et al., 2018a; Zhu et al., 2019a; Walter et al., 2020; Mousavi et al., 2020, amongst others); ii.) discriminate between volcanic and tectonic earthquakes (e.g., Esposito

et al., 2006) and, in the future, after updating INSTANCE with new data, discriminate earthquakes and other sources of seismic energy (e.g., sonic booms, quarry blasts, underwater explosions) often felt by the population (e.g., Del Pezzo et al., 2003; Linville et al., 2019); iii.) the rapid and accurate characterization of the earthquake source, distance, depth (e.g., Perol et al., 2018; Trugman and Shearer, 2018; Kriegerowski et al., 2018; Zhang et al., 2020; Lomax et al., 2019; Mousavi and Beroza, 2020; Münchmeyer et al., 2021) and of the ground shaking (e.g., Alavi, 2011; Derras et al., 2012; Derras, 2014; Jozinovic et al., 2020; Münchmeyer et al., 2020).

Indeed, the field of application of the dataset is quite extensive and it can be used to address many diverse topics depending on how the data are grouped and it can also be useful for applications not relying on ML techniques. For example, the dataset features some stations with several thousand of traces recording earthquakes from different azimuths and distances that can be used to construct common-station gathers of seismograms for swaths of sources in almost any desired geometry (e.g., Korneev et al., 2003), or to study in detail the local site response. Analogously, the metadata alone provide a rich set of arrival times (cf. Fig. 7) that could be used as is for traveltime tomography at regional scale in Italy and, in addition, the availability of the associated waveforms makes it possible the application of methodologies that resolve the velocity structure jointly using arrival times and waveform data (e.g., Zhang* and Chen, 2014). For what concerns the ground motion amplitude data, the availability of these metadata can be of relevance in combination with the shakemaps (http://shakemap.ingv.it) to develop new tools for rapid earthquake ground motion estimation. Other applications of the data collection include the adoption of unsupervised ML algorithms to group the waveforms independently of the earthquake location and just on the waveform themselves (e.g., Seydoux et al., 2020). INSTANCE can also be used, as a dataset with a large number of data, for creating pre-trained models when using transfer learning techniques either for seismological or other applications which use time-series data (Otović et al., 2021).

Overall, we believe that the dataset will be useful for stepping up towards a new generation of earthquake monitoring tools that will profit of the ongoing very fast developments in machine learning. What is certain is that seismology is in great need of benchmark datasets (Mousavi et al., 2019) upon which test new and existing techniques. To this end, standardization of the input data and metadata formats is of great relevance and in constructing this dataset we have adopted the schema proposed by the `SeisBench` initiative (Wollam et al., 2021). Needless to emphasize that widespread adoption of the same metadata schema and data volume formats can foster the compilation of similar datasets also for other regions with the possibility to merge them all together giving the opportunity to perform ML analysis exploiting the potentials of the resulting huge large datasets. Perhaps more importantly, standardization of data and metadata formats will make it easier to test different datasets using the same ML model or, alternatively, benchmarking different models on the same dataset and in both cases the benefits appear clear.

## 6  Data availability

The data used in this work were downloaded using the webservices provided by INGV (http://terremoti.ingv.it/en/webservices_ and_software). The networks used for the ISTANCE dataset are organized in Table 7.

**Table (7).** Seismic networks used in the compilation of INSTANCE dataset

| Code | Name | Identifier | Citation |
|---|---|---|---|
| 3A | Seismic Microzonation Network 2016 Central Italy (2016-2016) | https://doi.org/10.13127/SD/ku7Xm12Yy9 | Istituto Nazionale di Geofisica e Vulcanologia (INGV) et al. (2018) |
| AC | Albanian Seismological Network | https://doi.org/10.7914/SN/AC | Institute of Geosciences, Energy, Water and Environment (2002) |
| BA | UniBAS | https://www.fdsn.org/networks/detail/BA/ | Universita della Basilicata (2005) |
| GU | Regional Seismic Network of North Western Italy | https://doi.org/10.7914/SN/GU | University of Genoa (1967) |
| HL | National Observatory of Athens Seismic Network | https://doi.org/10.7914/SN/HL | National Observatory of Athens, Institute of Geodynamics, Athens (1997) |
| IV | Italian National Seismic Network | https://doi.org/10.13127/SD/X0FXNH7QFY | INGV Seismological Data Centre (1997) |
| IX | Irpinia Seismic Network | http://isnet.unina.it/ | Universita Federico II (2005) |
| MN | MedNet | https://doi.org/10.13127/SD/FBBBTDTD6Q | MedNet Project Partner Institutions (1988) |
| NI | North-East Italy Broadband Network | https://doi.org/10.7914/SN/NI | OGS (Istituto Nazionale di Oceanografia e di Geofisica Sperimentale) and University of Trieste (2002) |
| OT | OTRIONS | https://doi.org/10.7914/SN/OT | University of Bari "Aldo Moro" (2013) |
| OX | North-East Italy Seismic Network | https://doi.org/10.7914/SN/OX | Istituto Nazionale di Oceanografia e di Geofisica Sperimentale - OGS (2016) |
| RF | Friuli Venezia Giulia Accelerometric Network | https://doi.org/10.7914/SN/RF | University of Trieste (1993) |
| SI | Province Sudtirol | https://www.fdsn.org/networks/detail/SI/ | ZAMG - Zentralanstalt fur Meterologie und Geodynamik (2006) |
| ST | Trentino Seismic Network | https://doi.org/10.7914/SN/ST | Geological Survey-Provincia Autonoma di Trento (1981) |
| TV | INGV experiments network | https://www.fdsn.org/networks/detail/TV/ | Istituto Nazionale di Geofisica e Vulcanologia (INGV) (2008) |
| XO | EMERSITO Working Group. (2018) | https://doi.org/10.13127/SD/7TXeGdo5X8 | EMERSITO Working Group (2018) |
| YD | INGV SISMIKO Emergency Seismic Network for Molise-Italy (2018-2018) | https://doi.org/10.13127/SD/FIR72CHYWU | Moretti et al. (2018) |
| ZM | Seismic Emergency for Ischia by Sismiko (2017-2021) | https://www.fdsn.org/networks/detail/ZM_2017/ | Istituto Nazionale di Geofisica e Vulcanologia (INGV) (2017) |

## 7   Code and data availability

Routines and notebooks for analysis and display of the dataset (and the sample dataset) are available on https://github.com/ingv/instance. The processing was performed using *ObsPy* (Beyreuther et al., 2010; Megies et al., 2011; Krischer et al., 2015), *NumPy* (Harris et al., 2020), *SciPy* (Virtanen et al., 2020) and *Pandas* (McKinney, 2010; pandas development team, 2020)
5   python modules and the graphics was prepared using the *Matplotlib* library (Hunter, 2007) and *seaborn* (Waskom, 2021).

The dataset can be downloaded from http://doi.org/10.13127/instance (Michelini et al., 2021). A versioning schema has been also included since the dataset is expected to undergo modifications or expansions in the future. For instance, it is possible that some earthquakes or noise traces have been misclassified, or future significant seismic events be included. A sample dataset is also provided on the same landing page.

## 10   8   Conclusions

INSTANCE is the first dataset designed for the application of ML methodologies compiled using the seismic data archived on the ORFEUS-EIDA node of INGV for Italy (Danecek et al., 2021). One of the main scopes of the dataset is to provide a benchmark for developing improved techniques for earthquake and ground motion characterization. The dataset consists of about 1.3 M, 120 s long each, 100 Hz sampling, 3C traces subdivided into about 1.2 M containing seismic events and more
15   than 100,000 that include noise. The traces are assembled within HDF5 formatted volumes to facilitate access and analysis. More than 100 metadata grouped according to *source*, *station*, *trace* (and derived quantities) are associated to each trace to give the user much flexibility and control for the selection of the most appropriate data for her/his scientific targets.

The event data include recordings of earthquakes in the magnitude range $0 \leq M \leq 6.5$ and in the distance range between 0 and more than 600 km although the great majority of the traces belong to earthquakes in the magnitude range $2 \leq M \leq 3$ and within 250 km. The depth of the earthquakes varies between very shallow crustal earthquakes and about 600 km depth in the Calabrian subduction slab. The data have been recorded by more than 600 stations operating in Italy in the time span January 2005 — January 2020. The dataset equals to more than 43,000 hours of continuous event and noise data and associated metadata. An average of 21 3C traces are provided per earthquake.

## Appendix A:  Preparation of the HDF5 data containers

The waveform trace data are provided in binary HDF5 files volumes. The volumes have been prepared for event data in counts and ground motion units, and for noise data in counts. The HDF5 format allows for rapid and easy access to the individual traces without the need of loading the whole dataset into memory. The waveform trace datasets have been created using the HDF5-Group "data" structure https://portal.hdfgroup.org/display/HDF5/HDF5 which contains as many HDF5-Datasets as 3C waveforms. Every 3-component waveform is a separate HDF5-Dataset and is accessed by its trace name (`trace_name`) found in the metadata file.

## Appendix B:  Velocity model used by the Italian Seismic Bulletin for the earthquake locations

The earthquake locations provided in the Italian Seismic Bulletin (http://terremoti.ingv.it/en/help#BSI) are fully described by Mele et al. (2010). The model consists of two layers over a half space assuming a ratio $V_P/V_S = 1.732$.

**Table (B1).** P and S velocity model used in the location procedures for the Italian Seismic Bulletin. Two crustal layers are superimposed to and half-space.

|  | Thickness (km) | $V_P$ (km/s) | $V_S$ (km/s) |
|---|---|---|---|
| Upper Crust | 11.1 | 5.0 | 2.89 |
| Lower Crust | 26.9 | 6.5 | 3.75 |
| Mantle | half-space | 8.05 | 4.65 |

The software IPOP developed by Alberto Basili is used for the earthquake locations (Bono, 2008).

## Appendix C:  Positive and negative polarities

In this appendix, we examine the origin for the observed asymmetry (almost 2:1 ratio) in the number of reported positive (up) and negative (down) polarities of the INSTANCE dataset. We evaluate whether *i)* the inclusion of anthropic, unidentified

sources like quarry blasts mistaken for earthquakes can affect the reported asymmetry and *ii)* the tectonics of the region can condition the number of positive and negative polarities in INSTANCE.

For the first investigation, we have followed Mele et al. (2010) (see also the recent work by Gulia and Gasperini, 2021) who found that in the 2008 bulletin the 99.6 % of the blasts have local magnitude $M_L \leq 2.2$ (Fig. 23 of their study). We have progressively increased the lower magnitude threshold to verify whether the nearly 2:1 ratio between positive and negative polarities persists as the magnitude is increased. The expectation is that, as the magnitude increases, the ratio progressively levels out since the blasts (or other artificial sources) do not produce magnitudes greater than $M=3$ in Europe (cf. Giardini et al., 2004). To address the variation of the proportion between positive and negative polarities with magnitude, we have selected from the INSTANCE dataset earthquakes with progressively larger minimum threshold magnitudes and found that the fraction (percent values) of negative polarities increases progressively from 36 % to  41 % when including earthquakes with $M > 0.25$ and $M > 3$, respectively. For larger minimum magnitudes, the percentage stabilizes around 42-43 %. This indicates that inclusion of the polarities of e.g., unrecognized blasts (i.e., with $M < 3$) has a moderate but still significant impact on the observed asymmetry between the reported positive and negative polarities. This asymmetry, although somewhat surprising, seems to occur also elsewhere. For example, Ross et al. (2018b) report, in their analysis of the southern California earthquake dataset (before data augmentation), a content of 67 % and 33 % for up and down polarities, respectively. We also note that the regional tectonic setting in Southern California is quite different from that in Italy.

Secondly, we have subdivided the Italian area into two zones selecting earthquakes with magnitude $M > 2.5$ occurring in the Apennines (defined as a rectangular area from 41°N to 44°N latitude and 9°E to 15°E longitude), and elsewhere outside this area. This data selection is aimed to verify if the observed asymmetry of positive and negative polarities can result from the dominant extensional stress field characterizing the Apennines when compared to the other areas in Italy. In the Apennines target area, the largest majority of the earthquakes are characterized by normal faulting mechanisms: in this case the lobes of the seismic radiation pattern show negative polarities at short epicentral distances.

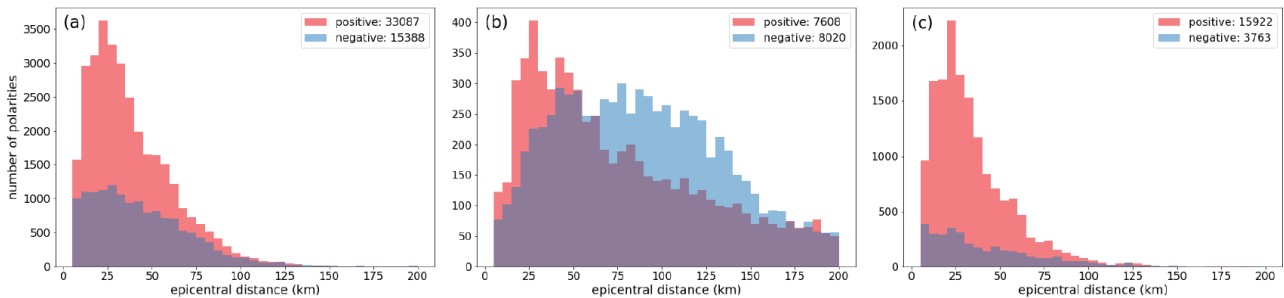

**Figure (C1).** Distribution of the positive and negative P-wave polarities for earthquakes with $M > 2.5$ in the Apennines region (41°N – 44°N and 9°E – 15°E) (a); (b) as in (a) but for earthquakes outside the Apennines; (c) as in (a) but filtered by backazimuth along the NE-SW directions, corresponding to the intervals 15° – 105° and 195° – 285°.

Given these conditions, the observed asymmetry could result from the complex interplay between the source receiver geometry, the 200-300 km width of peninsular Italy and the dominant NW-SE extensional faulting characterizing the tectonic regime in the Apennines. In this setting, the radiation pattern predicts negative polarities in the near field and positive polarities farther away from the source. Also the negative lobes are characterized by a smaller extension with respect to the positive lobes. Since the seismic receivers are more or less evenly distributed, the chance to record a negative pulse is smaller with respect to the positive one.

The Fig. C1 shows the histograms of the distribution of the positive and negative polarities with distance. The Fig. C1(a) shows the distribution of the polarities for the chosen target area in the Apennines, while Fig. C1(c) exhibits the polarities in the same area but only along backazimuth approximately NE-SW (i.e., the ranges 15°–105° and 195°–285° degrees). Finally, Fig. C1(b), displays the polarities from earthquakes outside the target area. We note that within the target area (Fig. C1(a)) the polarities are overwhelmingly positive in gross agreement with the explanation above. For further confirmation, we note in Fig. C1(c) that, if we restrict to the NE-SW propagation direction perpendicular to the Italian peninsula, the ratio between positive and negative polarities (pos %, neg %) increases further from (68 %, 32 %) to (81 %, 19 %), respectively. Conversely, the number of polarities for the earthquakes outside the target area are pretty much well balanced (49 %, 51 %) as shown in Fig. C1(b) .

In conclusion, *i)* the INSTANCE dataset does contain positive polarities resulting from the inclusion of quarry blasts misidentified as earthquakes for magnitudes $M \lesssim 2.5 - 3.0$. This follows from what reported by Mele et al. (2010) (see also the recent work by Gulia and Gasperini, 2021) and the change in positive and negative polarities proportions in INSTANCE at varying minimum magnitude thresholds appears to confirm it; *ii)* the current modalities of earthquake revision at INGV do not allow for the identification of all the anthropic sources so that the (source_type) parameter of the metadata can be misleading; *iii)* the target area in the selected Apennines region includes  76 % of the total number of polarities of the dataset and features a dominance of positive polarities: a possible explanation is the dominant normal type of earthquake faulting in the selected Apennines region; *iv)* the asymmetry observed in the target Apennines region disappears for $M > 2.5$ elsewhere in Italy.

**Appendix D:  Additional Quality Control figures**

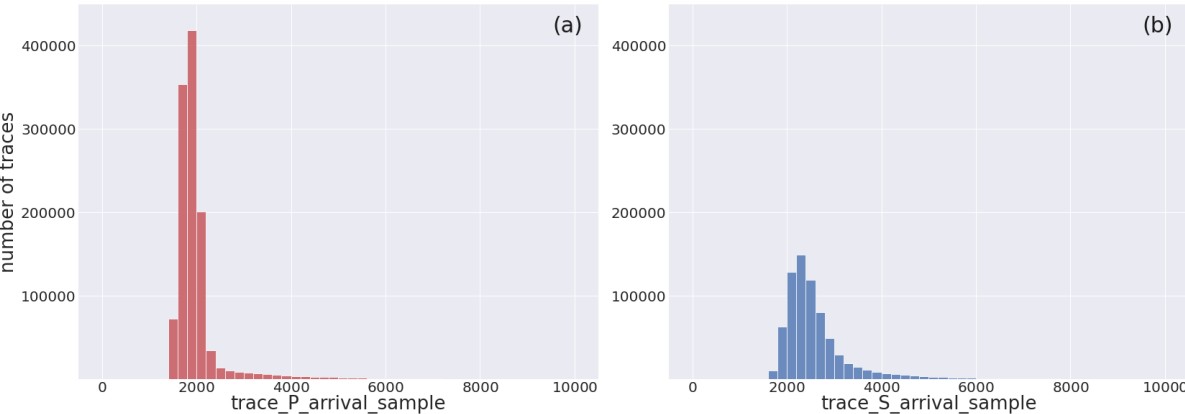

**Figure (D1).** Distribution of P- (a) and S-arrival (b) samples of the extracted waveform traces belonging to the earthquake dataset.

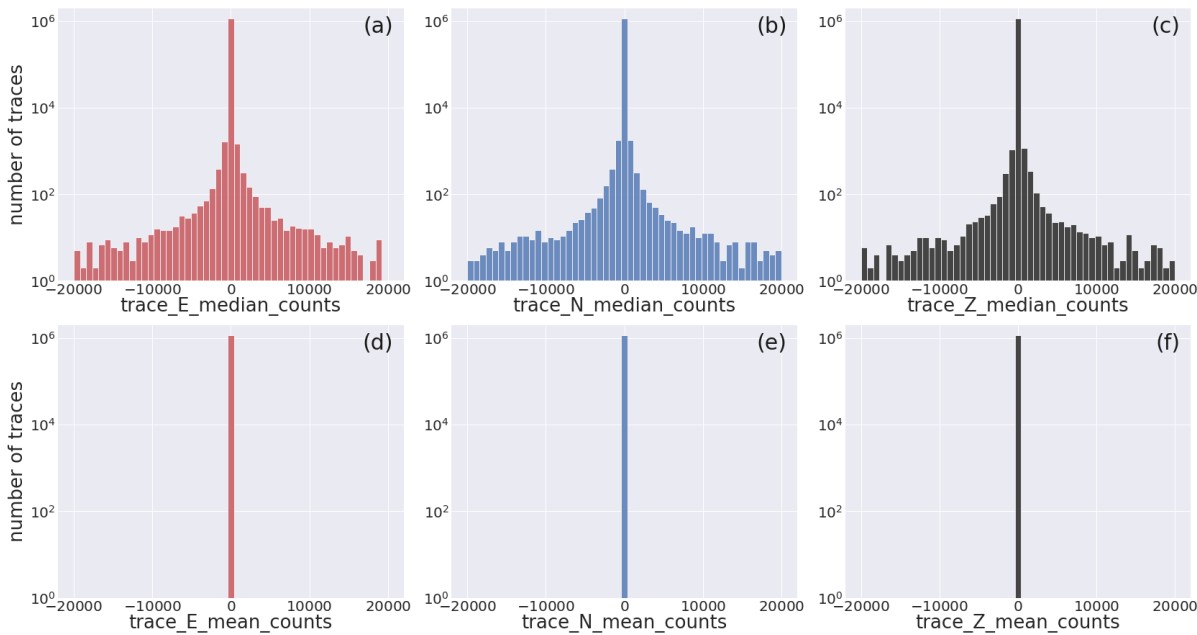

**Figure (D2).** Histogram of the distribution of the quality control metadata of the full earthquake dataset with the horizontal axis inclusive of the complete range of values: median (a-c) and mean (d-f). The width of the bins is $2 \times 10^3$; axes are labelled according to the metadata listed in Table 2.

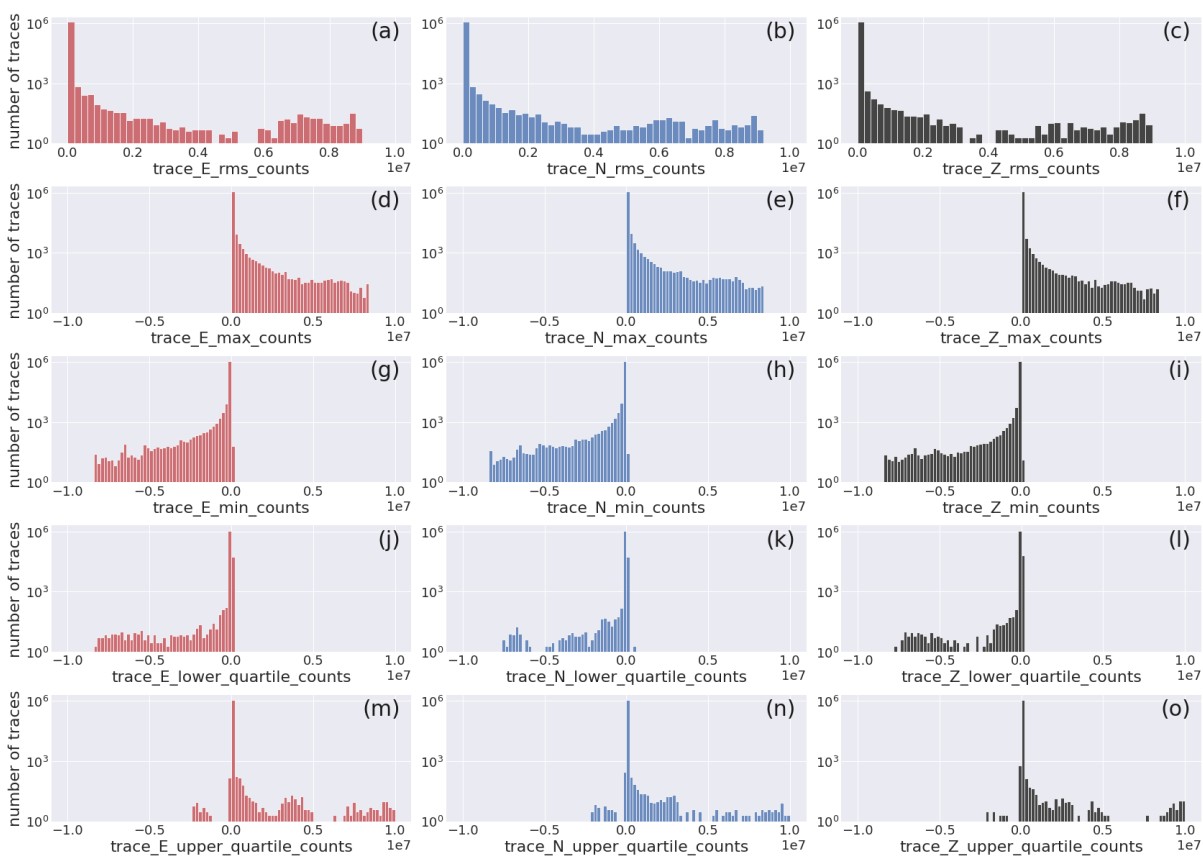

**Figure (D3).** Histogram of the distribution of quality control metadata of the full earthquake dataset with the horizontal axis inclusive of the complete range of values: rms, min, max, first and third quartile. The width of the bins is $2 \times 10^5$; axes are labelled according to the metadata listed in Table 2.

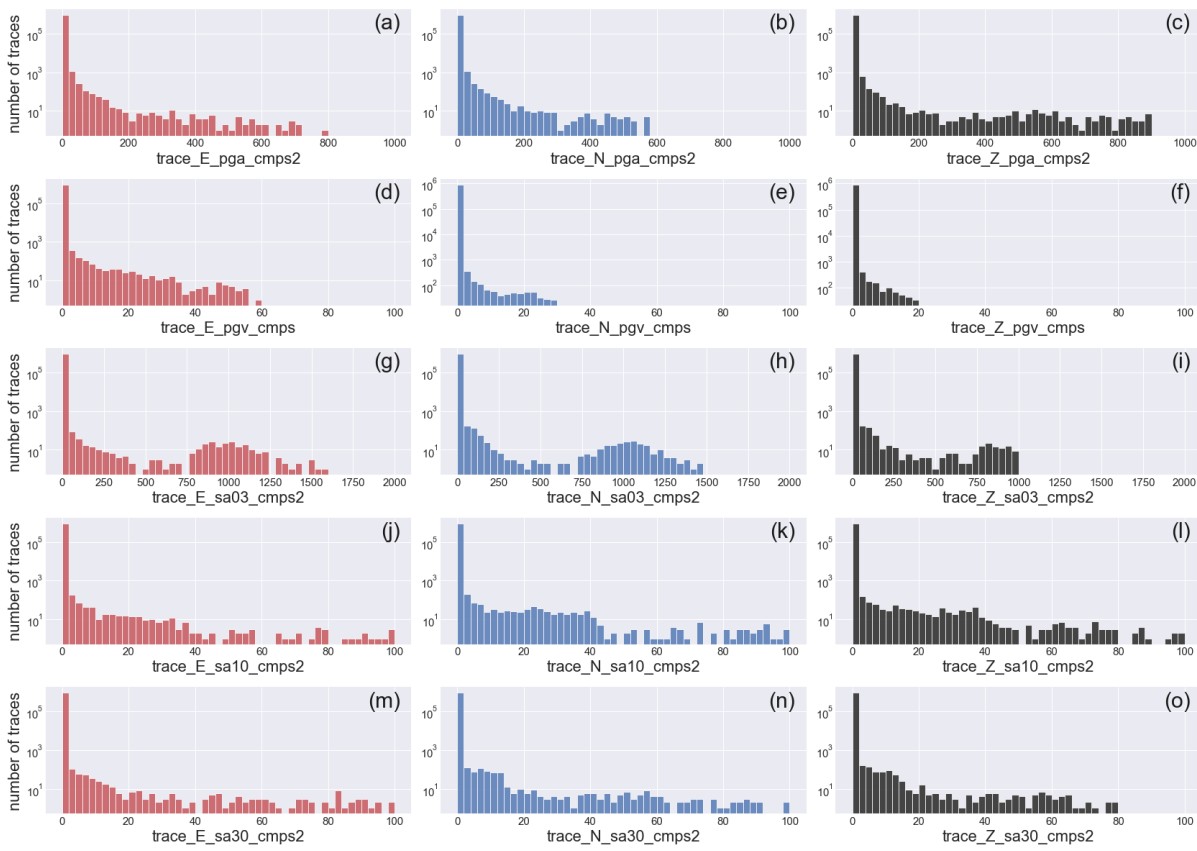

**Figure (D4).** Histograms of the distribution of the intensity measures (IMs) of the earthquake dataset for $M \geq 2$ earthquakes with the horizontal axis inclusive of the complete range of values. Axes are labelled according to the metadata listed in Table 2.

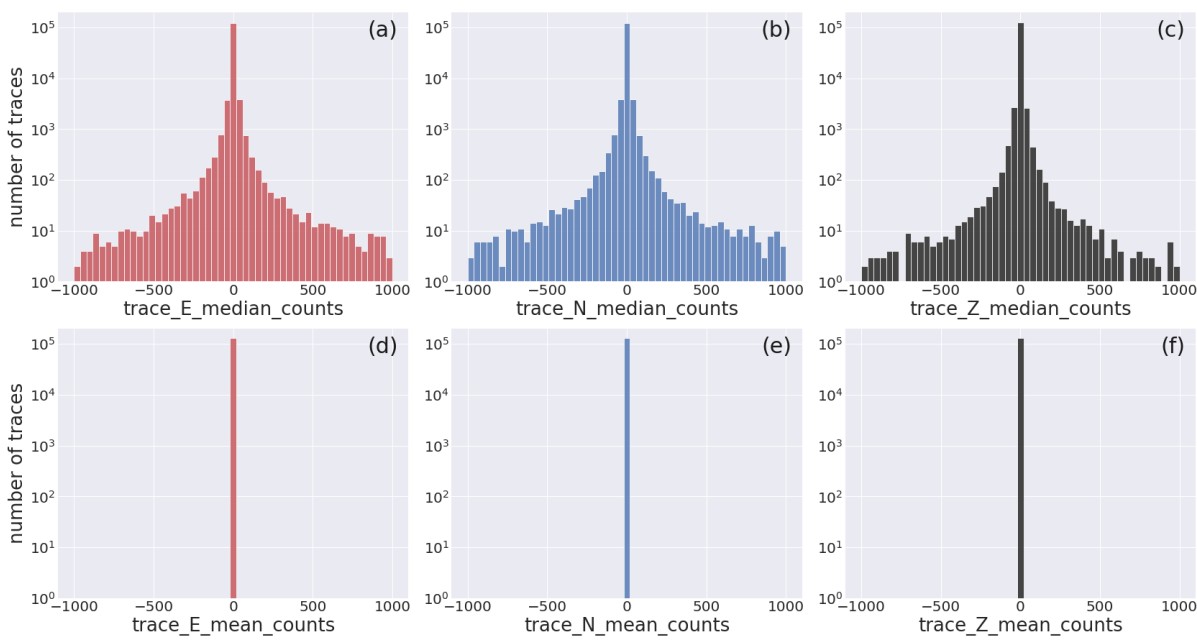

**Figure (D5).** Histogram of the distribution of the noise quality control metadata including the full range of values attained by the median values: median (a-c) and mean (d-f). The width of the bins is $2 \times 10^3$; axes are labelled according to the metadata listed in Table 2.

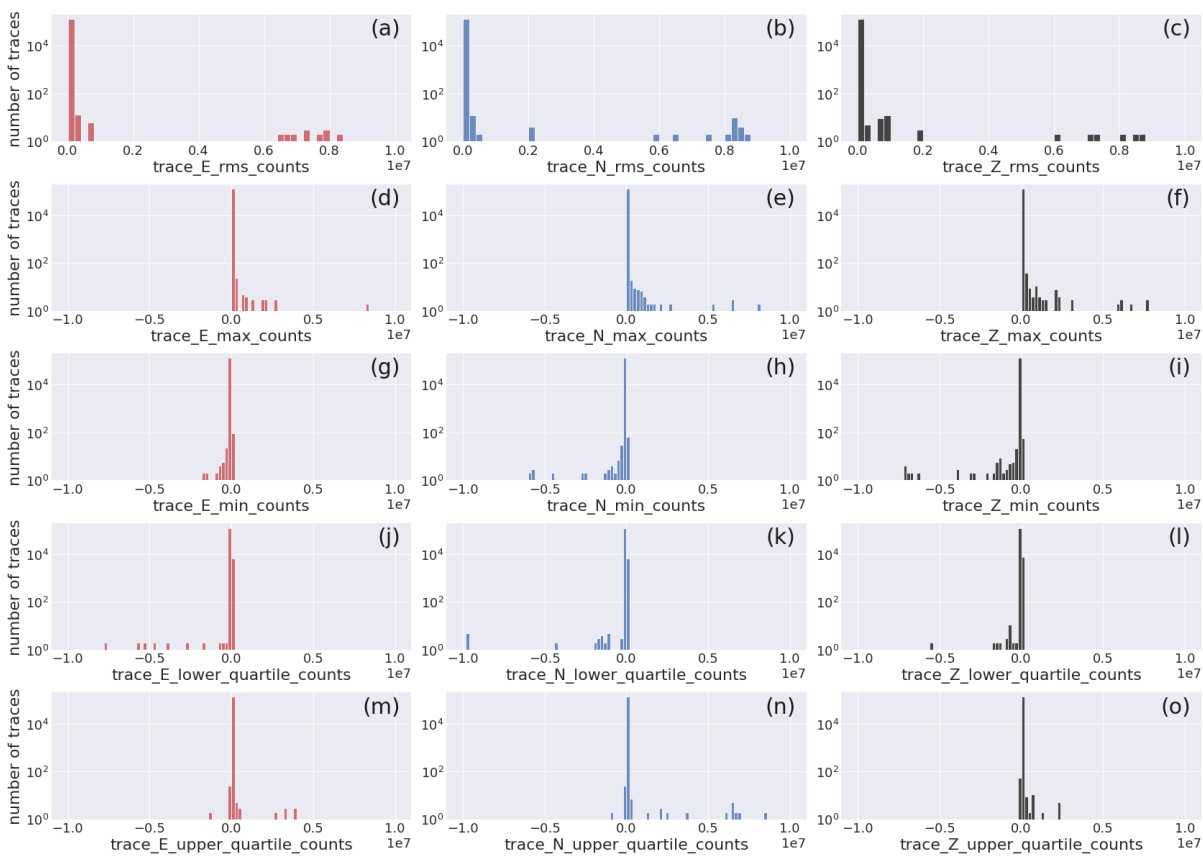

**Figure (D6).** Histogram of the distribution of noise quality control metadata: rms, min, max, first and third quartile. The width of the bins is $2 \times 10^5$; axes are labelled according to the metadata listed in Table 2.

*Author contributions.* AM prepared the initial data and metadata selection, drafted the manuscript and initial version of the figures. SC and SG provided ML estimation of the waveforms data, contributed to final versions of the figures and manuscript revision. CG contributed to data preparation and manuscript revision. DJ compiled the HDF5 formatted volumes and contributed to the data and manuscript revision. VL set up the virtual machines and the scripts to perform the massive data download.

*Competing interests.* No competing interests are present.

*Disclaimer.* The authors decline any responsibility for possible errors present in the INSTANCE dataset which can lead erroneous evaluations and to physical and economic damages.

*Acknowledgements.* This work would not have been possible without the effort and dedication of the people that install and maintain the stations of the networks used here, and the skilled IT people that are in charge of archiving, curating and providing access to the data. We
also greatly acknowledge the earthquake analysts that routinely perform the data analysis for the compilation of the earthquake bulletins. The metadata nomenclature benefited from the interaction and discussions within the recently formed *SeisBench* group. We would like to thank Jannes Münchmeyer, Elisa Tinti and Anthony Lomax for reading the initial manuscript and providing suggestions. This work has been partially supported by the project INGV Pianeta Dinamico 2021 Tema 8 SOME (CUP D53J1900017001) funded by Italian Ministry of University and Research "Fondo finalizzato al rilancio degli investimenti delle amministrazioni centrali dello Stato e allo sviluppo del Paese,
legge 145/2018" and by the European Union's Horizon 2020 research and innovation program under Grant Agreement Number 821115, real-time earthquake risk reduction for a resilient Europe (RISE). The reviewers Martijn van den Ende and John Clinton are kindly acknowledged for the insightful comments, questions and suggestions that helped to clarify the text and improve the manuscript overall.

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
