# Peer review of "INSTANCE - the Italian seismic dataset for machine learning"

_Earth System Science Data, 2021_

## Author Comment (AC2)

**Below in dark red the comments by the referee (John Clinton) and in black the reply.**

Note: The replies are provided only for the important comments which required additional work and/or clarification.

This is an excellent article introducing a new dataset targeting machine learning applications using a near-complete set of earthquake records from Italy. It follows the example of STEAD described in Mousavi et al, 2019. The manuscript provides an overview of how the dataset was generated and organised, and then provides an overview of the general features of the dataset.  The manuscript is well written, particularly the introduction that gives a strong motivation for why this type of product is sorely needed, and provides an overview of similar datasets. It is timely and important that high quality local earthquake records datasets outside the US W. Coast are highlighted and made easily available for researchers to use. I hope this sort of documentation of datasets and their preparation in research-ready format becomes the standard, and I expect that publication of this manuscript will lead to numerous publications on ML that use this dataset.

We are very pleased for the appreciation.

I know some information on the uniqueness of the dataset is dispersed through the article, but I suggest, in a single place,  in the discussion, the authors extend and accentuate what is different with this collection compared to others, besides from the obvious that this is solely an Italian dataset based on the Italian earthquake catalogue. Are the metadata fields better? How do they differ? Are formats modified ?  Is it unique to provide both raw and corrected waveform data? Data volumes similar to other datasets? Maybe a comparison table to STEAD and the Caltech datasets would be helpful. The authors could also provide stronger comments on the benefits of standardisation of formats / metadata for these datasets.

Yes, we will include a list of those components of the dataset that differ or are unique in the revised manuscript. Below some quick answers on the questions raised.

The metadata are many more than those included in previously published ML datasets. We will provide a summary table  comparing our metadata with those presented in other similar datasets cited in the ms (i.e., STEAD and the Caltech datasets, LEN-DB and the dataset used by Lomax et al., 2019, when developing ConvNetQuake_INGV).

In summary,

-  we are not aware of other datasets that include both raw and corrected waveforms. The only exception appears to be the article by Meier et al.  (2019) that uses the SCSN data (https://scedc.caltech.edu/data/deeplearning.htm)  and seems to include data in SI units but this is only stated in the README file linked above and not in the paper.

- The data volumes have been assembled following the indications of SeisBench -  a novel initiative to standardize the seismic data for machine learning (https://meetingorganizer.copernicus.org/EGU21/EGU21-12218.html).

- we will reword and add stronger comments on the benefits of format standardization.

A comment on the metadata on earthquake parameters summarised in Table 2: numerous fields are provided, including location uncertainty, but in the text, there is no comment on what location algorithm or velocity model is used. Since the dataset spans 15 years and a very wide geographic range, its likely that despite efforts to ensure a continuous approach to manual catalogue review, the velocity model and location algorithms have changed across the catalogue. If they have not changed, this point should be made. If they have changed, this should be indicated in the document, and consideration should be given to add this information in future updates of the dataset.

The location of all the earthquakes was made using the same 1D standard model adopted by INGV  for the locations reported in the 2008 Bollettino Sismico Italiano  (Mele et al, 2010). We have added a small section in the appendix to provide the model. The earthquake locations  are performed using the software IPOP developed by Alberto Basili (Bono, 2007). We can then assert that the source and the path/derived parameters are pretty much consistent through time.

The DOI of each of the network codes used in the dataset / publication must be made available in the references or in the data availability section

Yes, this is definitely important and they have been added in the data availability section.

In general, the figures are not optimal, often use strange axis labelling (that may be a direct metadata field from Table 2 - if so mention it!), and often have captions that are too terse / insufficiently descriptive. I suggest the authors look through these carefully. Also font sizes on  Figs 14,15,16 are too small- in particular the exponents are completely illegible.

We will add a note specifying that the labels of the figures follow from the naming of the metadata and we will make more exhaustive and self-contained descriptive captions for the figures. We are also fixing the font size issue in Figs 14, 15, 16 and 20.

...

P4 l25: some >M4.0 are rejected. A bit more info on  significant events that have been removed is needed. Are these only those that include multiple events in the same time window in the catalogue? I hope no very significant events are rejected simply because a very small foreshock or aftershock is also catalogued…

We attempted to keep all the earthquakes with M ≥ 4 but in some cases this was not possible because of missing data. As a consequence  30 earthquakes with M ≥ 4 were removed and they are shown in the picture below. Almost all of them occur outside the Italian country borders. The four earthquakes in Italy (near Catania and in southern Tyrrhenian sea) were removed accidentally because of a download technical problem and will be reintroduced in updated versions of the dataset.

[Figure]

P4 station selection: mention in this section that only stations on Italian territory are used. Are the Civil Defence stations not added? If not, mention why this very significant dataset missing - is it technical or political?

Also stations belonging to the MedNet network outside Italy and some stations of the Albanian and the Greek networks have been used.  Regarding the Italian Civil Protection (RAN network), these  stations  are not inserted in the dataset because they are not available in EIDA and they are not used for the compilation of the BSI upon which the data selection has been based. They may be included in future releases of the dataset although this would also involve the seismogram manual picking  (P and S phases) which may be quite heavy to be completed with the available human resources.

P5 l1-5: Be more specific on what picks are made available. I assume the INGV catalogue makes first arriving P and S picks only. No additional phase type is indicated (Pg vs Pn), and secondary phases are also not identified (eg PmP)

In the BSI there is no distinction between Pg and Pn or secondary phases like PmP and in the dataset they are just referred to as P phases.

P5 l6-9 its should be accentuated either here or later that since 1/ not all stations used in the catalogue generation are included, eg foreign-operated stations;

Although very few, there are foreign stations (i.e., AC and HL neworks) in the dataset. There are no stations from the countries bordering northern Italy  because they are not included in the EIDA Italian node that was used and our efforts were focused on Italian stations waveform data. Nevertheless, we consider that a dataset like INSTANCE can be easily integrated with additional data by following the same standards adopted here. In principle, it would be possible that other similar datasets be assembled for other regions following the same standards adopted here and then even merge them together in a single dataset.

2/  phases with large residuals or low weight are removed that it is not possible to use this dataset to relocate the catalogue.

The main point is that INSTANCE targets ML applications so that the results of these analyses could be then used to re-process the entire data and to rebuild the earthquake catalog. Since  the average number of stations per earthquake is about 21, in many cases these would be more than sufficient to attain good and stable earthquake relocation.

P5 waveform data selection: in the case of multiple available sampling rates, I infer that the same sampling rate used to make the manual pick is selected for inclusion here. Or is it the highest available sampling rate for each channel?

The understanding is correct. We provide the trace data resampled at 100 Hz and the arrival times obtained from the original trace data used for the picking. In most cases there was no need for resampling and the trace data coincide with those used to pick the arrival times. P6l11: I don't understand what is 'arrival time samples'. Why not simply use time in seconds?

The 'arrival time samples' represents the sample number of the phase arrival time into the array available in the hdf5. The use of  'arrival time samples' serves to  simplify the use of these quantities especially by non-seismologists. The arrival times are also provided.

P6 2.1.5: the authors should mention are all traces rotated into ZNE, or in the entire Italian catalogue in ZNE by default. If so, I am amazed!

They are by default all along ZNE. We do not have any waveform included that resulted from sensors oriented differently.

P7 l.29 2.2 Metadata: in source, the location method or velocity model are not included. They should be if either of these have changed over 15 years of the catalogue.

The velocity model used for location in the compilation of the BSI has not been changed in 15 years. Thus the earthquake  locations in this sense are all consistent. Details on the BSI procedure are provided in Mele et al. (2010) and the velocity model is also provided in an additional appendix.

P8 l25 'missing data' - please expand

In Jozinovic et al. (2020), the dataset used for ML consists of a fixed number of stations and when data from one or more stations are missing (either the whole trace or parts of it), the signal trace is set to be an array of zeros. The ML model used there was found to detect and learn the problematic values, and compensate for it, having a similar prediction accuracy on those stations as the accuracy on the stations which had the input data available .

P10 Table 2: location code is not part of the International Registry?

Thanks for noting it. Yes, the location code is not part of the International Registry. "Changed in  table 2.

P13 l 5 I'm surprised to see selection criteria was for even number of traces for each channels? Seems in contradiction to 2.1.2, where all reasonable phases according to seismicity were selected. Was seismicity for smaller events actually selected according to numbers of station pick?

These are selection criteria for the *noise recordings* and there is no relation with the number of picks available for each station. We made an attempt to select all the station channels with a more or less even number of recordings.

P14 l4 'great majority' seems an exaggeration.

The histogram in Figure 3d adopts a log scale. This issue was also reported by the other referee and the histogram scale will  be changed to linear to better evidence the assertion.

P15 Fig 5 / l9 onwards: the number of up first motion polarities is double that of down. This is surprising, and possible concerning unless there is a reasonable explanation I do not see. The authors should explain this. Is it possible eventype=earthquake is not selected, and blasts are also included here?

We thank the referee for raising this issue which we did not address thoroughly in our manuscript.

As described in the manuscript, we have adopted the "event" FDSN web service implemented at INGV which adopts the standard FDSN parameters which at the moment do not include the "event_type" field for selection. This has been also noted recently by Gulia and Gasperini (2021). However, it is still possible to download the quakeML (a xml formatted file standard for seismology) for each event which includes the "event_type" parameter.

We have therefore proceeded to obtain the event_type value and we have included it as additional parameter (`source_type`) in the metadata file. Thus, the new metadata file now includes 115 parameters total. It appears, however, that the addition of the new parameter extracted from the INGV archive captures only a fraction of the non-earthquake sources. That is, many artificial sources are still catalogued as earthquakes in the INGV archive. The

table below provides a snapshot of the event_type included in the proposed dataset.

| type_event | |
|---|---|
| anthropogenic event | 1 |
| controlled explosion | 44 |
| earthquake | 53753 |
| experimental explosion | 8 |
| explosion | 5 |
| landslide | 1 |
| quarry blast | 194 |
| volcanic eruption | 2 |

In addition, the BSI distinguishes between earthquakes and other sources like quarry blasts only since 2012 (Gulia and Gasperini, 2021).

Given that the inclusion of the event_type above still misses several artificial sources, we have addressed the asymmetry noted by the referee between the number of positive and negative polarities by other means. To this end, we performed two different analyses to verify i.) how the inclusion of blasts can affect the reported asymmetry and ii.) how the region with its dominant tectonic style can condition the number of positive and negative polarities in INSTANCE.

Following Mele et al. (2010) who found that the 99.6% of the blasts have local magnitude ML ≤ 2.2 (Fig. 23 of their study), we have progressively increased the lower magnitude threshold to verify whether the nearly 2:1 ratio between positive and negative polarities persists as the magnitude is increased. The expectation is that as the magnitude increases, the ratio progressively levels out since the blasts (or other artificial sources) do not produce magnitudes greater than M=3 in Europe (Giardini et al., 2004).

Secondly, we have subdivided the Italian area into two zones: earthquakes inside the Apennines area [vertices (lat,lon) (41N,9E and 44N,15E)], and earthquakes elsewhere outside this area. This data selection is aimed to verify if the observed asymmetry of positive and negative polarities can result from the dominant extensional stress field characterizing the Apennines when compared to the other areas in Italy.

To address the variation of the proportion between positive and negative polarities with magnitude, the table below shows that the fraction (per cent values) of negative polarities increases progressively from 36% to ~41% when including earthquakes with M>0.25 and M>3, respectively. For larger minimum magnitudes (M>3), the percentage stabilizes around 42-43%. This would indicate that inclusion of the polarities of unrecognized blasts (i.e., with M<3) has a moderate but still significant impact on the observed asymmetry between the reported positive and negative polarities. This asymmetry, although somewhat surprising, seems to occur also elsewhere. For example, Ross et al. (2019) report in their analysis of the southern California earthquake dataset (before data augmentation) that their data contains 67% and 33% up and down polarities, respectively. We also note that the regional tectonic setting in Southern California is quite different from that in Italy.

| min_magnitude | total | positive | positive_percent | negative | negative_percent |
|---|---|---|---|---|---|
| 0.25 | 236345 | 151544 | 64.12 | 84801 | 35.88 |
| 0.5 | 235806 | 151204 | 64.12 | 84602 | 35.88 |
| 0.75 | 234400 | 150335 | 64.14 | 84065 | 35.86 |
| 1 | 227810 | 146213 | 64.18 | 81597 | 35.82 |
| 1.25 | 219688 | 141096 | 64.23 | 78592 | 35.77 |
| 1.5 | 204277 | 131159 | 64.21 | 73118 | 35.79 |
| 1.75 | 194880 | 125020 | 64.15 | 69860 | 35.85 |
| 2 | 160464 | 102359 | 63.79 | 58105 | 36.21 |
| 2.25 | 118581 | 75072 | 63.31 | 43509 | 36.69 |
| 2.5 | 75907 | 46810 | 61.67 | 29097 | 38.33 |
| 2.75 | 57366 | 34740 | 60.56 | 22626 | 39.44 |
| 3 | 37183 | 21821 | 58.69 | 15362 | 41.31 |
| 3.25 | 27333 | 15748 | 57.62 | 11585 | 42.38 |
| 3.5 | 16749 | 9447 | 56.4 | 7302 | 43.6 |
| 3.75 | 12328 | 6979 | 56.61 | 5349 | 43.39 |
| 4 | 7200 | 4151 | 57.65 | 3049 | 42.35 |
| 4.25 | 4935 | 2810 | 56.94 | 2125 | 43.06 |
| 4.5 | 2232 | 1312 | 58.78 | 920 | 41.22 |
| 4.75 | 1369 | 833 | 60.85 | 536 | 39.15 |
| 5 | 814 | 468 | 57.49 | 346 | 42.51 |

For our second analysis (proportion between positive and negative polarities depending on the area), we have considered that in Europe the maximum magnitude of quarry blasts is usually assumed to be 2.5–3.0 (Giardini ed al., 2004) and, following the findings of Mele et al (2010), we focus only on earthquakes with M> 2.5.  We have extracted the polarities for the target Apennine region and compared to those reported for earthquakes elsewhere in Italy.  In the target area, the largest majority of the earthquakes  are characterized by normal faulting mechanisms with the lobes of the seismic radiation pattern having negative polarities at short epicentral distances. Given these conditions, the observed asymmetry could result from the complex interplay between the source receiver geometry, the width of 200-300 km coast to coast from the Tirrhenian to the Adriatic seas of peninsular Italy and the dominant extensional faulting with faults striking NW-SE characterizing the Apennines and dominated by normal faulting.

In this setting, the radiation pattern predicts negative polarities in the near source  and positive polarities farther away. Also, the negative polarity source radiation lobes map into a smaller  extension region near the epicenter that, in general, will also have a smaller

number of stations when compared to the positive lobes of larger extension and, consequently, a larger number of stations.

The figure below shows the histograms of the distribution of the positive and negative polarities with distance. The panel to the left shows the distribution of the polarities for the chosen target area in peninsular Italy, in the rightmost the polarities in the same area but only along the approximate NE-SW direction of the backazimuth (i.e., the ranges 15-105 and 195-285 degrees) and, in the middle, the area outside this target area. We note that within the target area the polarities are overwhelmingly positive in gross agreement with what described above and, for further confirmation, we see that if we restrict to the NE-SW propagation direction perpendicular to the Italian peninsula (rightmost panel), the ratio between positive and negative polarities (%pos,%neg) increases from (68%,32%) to (81%,19%), respectively. Conversely, the number of polarities for the earthquakes outside the target area are pretty much well balanced (49%, 51%).

[Figure]

In conclusion, i.) the INSTANCE dataset does contain positive polarities resulting from the inclusion of quarry blasts misidentified as earthquakes for magnitudes less than ~2.5-3.0. This follows from what reported by Mele et al. (2010) (and very recently by Gulia and Gasperini, 2021) and the change in positive and negative polarities percentages reported in our table above confirms it; ii.) the current modalities of earthquake revision at INGV do not include identification of all the manmade sources and, the web service used does not include the *eventtype* identification but it was still possible to retrieve the event_type and, accordingly, add a new source parameter (`source_type`) to the dataset metadata; iii.) the target area in the selected Apennine region includes ~76% of the total number of polarities of the dataset; iv) In the Apennine region there is dominance of positive polarities which is likely the result of the dominant normal type of earthquake faulting in the area; v) the asymmetry observed in the target area disappears for M>2.5 elsewhere in Italy.

P17 l1 1/2: is it possible this can also be explained by systematically mis-identified first arrivals, rather than complications in the velocity structure?

It has been verified that these very long traveltimes belong to earthquakes that occurred during the 2012 Emilia earthquake sequence. The stations recording these events were located on the soft and thick alluvium characterizing the Po plain which features very low seismic velocities.

P27 l10: earthquake in INGV catalogue - so its very possible that noise traces include energy from regional and teleseismic events.

Yes the referee is correct. Will be pointed out in the manuscript

P27 l14: any effort to include the same spread of stations as found in the event dataset?

No, if for spread it is meant the same group of stations detecting earthquakes in a given area for the same time window. Anyhow, the stations are exactly the same as those of the event dataset as evidenced in Figure 2.

P39 FigA4 - over 100 records have PGA >2g, and many even over 4g. Which is rather unphysical.  Is this understood?

The units are cm/s^2 and we do not see any value above 1g for PGA.

**References**

Basili A., 2005. Locator: Il manuale, Documentazione disponibile in forma digitale.

Bono, Andea. "SisPick! 2.0 Sistema interattivo per l'interpretazione di segnali sismici, Manuale utente," 2008. https://istituto.ingv.it/images/collane-editoriali/rapporti%20tecnici/rapporti-tecnici-2008/rapporto59.pdf.

Gulia, Laura, and Paolo Gasperini. "Contamination of Frequency–Magnitude Slope (B‑Value) by Quarry Blasts: An Example for Italy." Seismological Research Letters, June 30, 2021. https://doi.org/10.1785/0220210080.

Meier, Men-Andrin, Zachary E. Ross, Anshul Ramachandran, Ashwin Balakrishna, Suraj Nair, Peter Kundzicz, Zefeng Li, Jennifer Andrews, Egill Hauksson, and Yisong Yue. "Reliable Real-Time Seismic Signal/Noise Discrimination With Machine Learning." Journal of Geophysical Research: Solid Earth 124, no. 1 (2019): 788–800. https://doi.org/10.1029/2018JB016661.

Mele, F., Luca Arcoraci, Patrizia Battelli, Michele Berardi, Corrado Castellano, Giulio Lozzi, Alessandro Marchetti, Anna Nardi, Mario Pirro, and Antonio Rossi. "Bollettino Sismico Italiano 2008 (Italian Seismic Bulletin 2008)." *Quaderni di geofisica*, no. 85 (2010): 45.

Ross, Zachary E., Men-Andrin Meier, and Egill Hauksson. "P Wave Arrival Picking and First-Motion Polarity Determination With Deep Learning." Journal of Geophysical Research: Solid Earth 123, no. 6 (2018): 5120–29. https://doi.org/10.1029/2017JB015251.

---

## Author Response (AR1)

Rome, 30 August, 2021

Dear Editor,

Please find below the point-by-point response to the reviews including a list of all the relevant changes made in the manuscript.

With regard to the annotated version (INSTANCE_annotated_ESSD-2021-164.pdf), because of a glitch of the `latexdiff` package used for tracing the manuscript changes, it includes modifications of the text but not of the figures. However, all the figure modifications are listed in the replies to the referees below.

With regard to the annotated version, you will also see some small changes to improve the English which had not been noted before.

Overall, we have replied to all the comments and suggestions made by the referees following the guidelines provided by ESSD.

I am looking forward to hearing from you.

Best Regards,

Alberto Michelini

**Below in dark red the comments by the referees, in black the reply and in blue the modification in the text**

In the following we adopt:

Dark red **text** for (1) comments from Referees,
Black **text** for (2) author's response,
Blue **text** for (3) author's changes in manuscript

**Review by Martijn van den Ende (ref1)**

This manuscript describes a newly created dataset comprising broadband seismic recordings of earthquakes and ambient noise in Italy, called INSTANCE. Public datasets are of utmost importance to advance Machine Learning in seismology, and ensure their reproducibility. I therefore applaud the authors for their efforts to create a dataset with Machine Learning applications in mind. The authors perform various quality checks and provide metadata based on which the user can determine which data to include in their investigations. Moreover, the INSTANCE dataset could be suitable for analyses outside of the domain of Machine Learning, and the authors facilitate these by including detailed metadata of both the earthquake and noise recordings.

The manuscript that describes the dataset is well written and provides an extensive analysis of various summary statistics of the dataset, as well as a useful review of other datasets

that are currently available. I have a few minor suggestions for clarifications in the text (see below), but other than that I think that the manuscript is in good shape.

The project webpage and GitHub repository are clearly structured, and include Python notebooks to reproduce the figures presented in the manuscript. The example data represent only a subset of the data analysed in the manuscript, and so the figures look different, but I understand that it could be too computationally expensive to re-run the analyses on the full dataset. In principle the users could apply the same code to the full dataset if they wish to exactly reproduce this study.

This is correct, it would not have been possible to provide the complete dataset on GitHub due to size constraints. The purpose was to provide the python notebooks that we used. These can be easily modified to access the entire dataset and replicate the manuscript's figures or to make any other kind of data selection

One suggestion I would like to make here is to consider some form of versioning of the INSTANCE dataset. It is not unimaginable that the dataset be modified or expanded in the future, for instance to re-classify misclassified earthquake or noise traces, or to include future seismic events of great significance. By including a version tag (and changelog), users can specify which version of INSTANCE they used in their analyses, which would improve the reproducibility of future studies.

We fully agree with the referee and we have already included this feature in our DOI scheme. We now make it explicit in the manuscript. This will be clarified in the revised version. In the "Data availability" section the following sentence has been added.

A versioning schema has been also included since the dataset is expected to undergo modifications or expansions in the future. For instance, it is possible that some earthquakes or noise traces have been misclassified, or future significant seismic events be included.
* * *
As for the core data files (I reviewed the sample dataset): the waveforms are stored in datasets labelled in correspondence with their trace name, which makes it convenient to extract only the waveforms that match a metadata query. I did notice that in many cases (about 1 in 3 traces), the evaluation of EQTransformer has yielded only NaNs. I'm not sure how to interpret this, because I would expect an integer number (zero if no detection was made) instead of NaN. Perhaps the authors could check this and mention how NaN should be interpreted, or replace them with zeroes where applicable. Nonetheless, the fact that EQTransformer fails in 1 out of 3 cases is a bit startling, and underscores the need for dedicated datasets for specific regions to (re)train Machine Learning models.

The NaN is used to indicate the missing values for the low gain (accelerometric) channels where both EQTransformer and GPD have not been run. In the case of high gain channels, the lack of detection is indicated by '0'. High gain channels are (e.g. Figure 5a) ~71% of the data.

To avoid misunderstandings we have added the following sentence in the "Metadata description" section

Both GPD and EQTransformer have been run only on the high gain channels (i.e., HH, EH).

For the EQTransformer, we do agree that an "ad hoc" training could improve its performances, but this is beyond the aim of our manuscript.
* * *
Lastly, I noticed that after bzip decompression the data files take up a lot more disk space, so it could be helpful for users to indicate on the INSTANCE webpage what the data size after decompression is.

We agree that expliciting the disk space in the landing page is important and it has been fixed. Please see the snapshot below taken from the landing page (http://www.pi.ingv.it/instance/)

- Events metadata (csv file, 236 MB bz2 file, 1.1 GB after decompression) − doi:10.13127/instance/eventsmetadata.1
- Events data in digital units as single hdf5 file (39 GB bz2 file, 156 GB after decompression) or 10 GB parts (part-a, part-b, part-c, part-d) − doi:10.13127/instance/events.1
- Events data in ground motion units as single hdf5 file (151 GB bz2 file, 156 GB after decompression) or 20 GB parts (part-a, part-b, part-c, part-d, part-e, part-f, part-g, part-h). Ground motion units are m/s for HH and EH channels and $m/s^2$ for HN channel − doi:10.13127/instance/groundmotion.1
- Noise metadata (csv file, 6.7 MB bz2 file, 53 MB after decompression) − doi:10.13127/instance/noisemetadata.1
- Noise data in digital units (h5 file, 3.9 GB bz2 file, 18 GB after decompression) − doi:10.13127/instance/noise.1
- Stations inventory (StationXML, 15 MB)

Overall, I think that INSTANCE is an important contribution to the seismic community, and I recommend acceptance of the manuscript after some (very) minor revisions.

Minor comments on the manuscript (P = page; L = line):

1. P1, L19-20: in addition to the URLs, I would add the names of the ML platforms to make these more recognisable (and in case the URLs are changed in the future).

We have followed the indication and added the names

...software platforms like TensorFlow (https://www.tensorflow.org), Py-Torch (https://pytorch.org), Keras (https://keras.io), Caffe (https://caffe.berkeleyvision.org) (see Abadi et al., 2016; Paszke et al., 2019; Chollet and others, 2015; Jia et al., 2014, respectively), the availability...
* * *
2.    P6, L1-2: I don't quite understand that is meant here. Could the authors clarify this?

This is a technical detail that describes how the waveform trace windowing has been performed. In practice, we did prefer to select the data  trace windows to preserve a time buffer of 15-20 s of data before the P-wave arrival. The actual value of this buffer was selected randomly to avoid, for instance, that a ML model learns that exactly at a given time sample corresponds the P-wave arrival time. This procedure was also adopted by Mousavi et al. (2019) in the preparation of STEAD. The selection of the 125 s long window is arbitrary since during processing all the windows are cut to 120 s.

More technically, the time windows set for data download were defined by inserting a randomly selected buffer time ranging between 15 and 20 s before the P-wave onset arrival phase and enlarging the time window to 125 s. The adoption of 125 s long windows at the data download stage is arbitrary since after data processing the time windows have been all set to 120 s.
* * *
3.    Figure 3, panel d: I would plot these data on a linear scale to better see the trends. It doesn't seem very logical to me to plot azimuth on a log scale.

Yes, the referee is correct and we did replace the log scale with the linear scale.

[Figure]

4.    Figure 1, panel a: the colour scale does not really provide any insights into the depth of the vast majority of earthquakes on this map (practically all the events are coloured deep red). To make this panel more informative, I would tailor the colour scale to the crustal earthquakes and accept clipping for the deep earthquakes. Also, the jet colour map is hard to interpret in grayscale (when printing) and by readers with colour vision deficiencies. I would recommend re-rendering this figure with a perceptually uniform colour map like viridis or cividis.

We do agree with the referee and we have changed the map according to the suggestions provided. The new map adopts the inferno color scheme and the colors have been chosen to appreciate the earthquake depth.

[Figure]

5.  P16, L8, Fig. 6: maybe I missed a mention of this earlier in the manuscript, but which velocity model is used to calculate the theoretical arrivals / residuals? Is the model consistent for the entire dataset?

The location of all the earthquakes was made using the same 1D standard model adopted by INGV  for the locations reported in the 2008 Bollettino Sismico Italiano  (Mele et al, 2010). We have added a small section in the appendix to provide the model. The earthquake locations  are performed using the software IPOP developed by Alberto Basili (Bono, 2007).

**Appendix B: Velocity model used by the Italian Seismic Bulletin for the earthquake locations**

The earthquake locations provided in the Italian Seismic Bulletin (http://terremoti.ingv.it/en/help#BSI) are fully described by Mele et al. (2010). The model consists of two layers over a half space assuming a ratio $V_P/V_S = 1.732$.

Table (B1). P and S velocity model used in the location procedures for the Italian Seismic Bulletin. Two crustal layers are superimposed to and half-space.

|  | Thickness (km) | $V_P$ (km/s) | $V_S$ (km/s) |
|---|---|---|---|
| Upper Crust | 11.1 | 5.0 | 2.89 |
| Lower Crust | 26.9 | 6.5 | 3.75 |
| Mantle | half-space | 8.05 | 4.65 |

The software IPOP developed by Alberto Basili is used for the earthquake locations (Bono, 2008).

6.  P17, L11-12: since the mean is removed from each trace, I would be surprised if the mean would be anything else than zero. The figures that show histograms of the mean are therefore not very informative and could be omitted, in my opinion.

We are well aware that the figures showing the mean are concentrated at zero since the mean has been removed. The reason for showing these panels is to evidence the difference with the median values that are also plotted in the same figures and we prefer to keep the panels with the mean values as presented.

7.  P27, L11: could the authors confirm that the 120s time window is different for each station? In the way it is written now, it is not fully clear to me. I think it's unlikely that one would find a 120s window common to all stations that matches all of the criteria simultaneously, but this is not explicitly mentioned in the text. This would be an important consideration when making any assumptions regarding the spatial coherence of the noise.

The referee is correct. The 120 s is different for each station. The following text has been inserted

It follows that the adopted procedure does not entail the selection of the same time window for multiple stations.

8.  P32, L4: personally I would also have created a dataset with low SNR waveforms to complete the "spectrum" from noise to clear earthquakes (useful for microseismicity studies), but I understand that this may be beyond the scope of the project. Perhaps this is something that the authors could consider including in the future (see my comment about versioning).

The objective of the dataset is to provide a data set that encompasses a broad range of SNR values. Considering that 10% (see Table 3) of the trace data of the HH channels have SNR values less than 2.3 (1.2 for the vertical component) that corresponds to roughly to 59,000 waveform traces out of the ~592,000 traces of the HH channels included in the dataset, we believe that the dataset still has a more than reasonable representation of low SNR traces which could be used for detecting e.g., small earthquakes. We agree that in future versions of the dataset even more low SNR traces could be added.

We have added the following sentence in the article in "Dataset description" section

In contrast and at the lower end of the SNR distribution, we find that 10\% (see Table~\ref{tab:criteria_selection}) of the trace data of the HH channels have SNR values less than 2.3 (1.2 for the vertical component) that corresponds to roughly to 59,000 waveform traces out of the ~592,000 traces of the HH channels included in the dataset. This number of low SNR traces could be used, for example, to train machine learning models aimed to the detection of very small magnitude earthquakes little above the background noise level.

**Review by John Clinton (ref2)**

This is an excellent article introducing a new dataset targeting machine learning applications using a near-complete set of earthquake records from Italy. It follows the example of STEAD described in Mousavi et al, 2019. The manuscript provides an overview of how the dataset was generated and organised, and then provides an overview of the general features of the dataset. The manuscript is well written, particularly the introduction that gives a strong motivation for why this type of product is sorely needed, and provides an overview of similar datasets. It is timely and important that high quality local earthquake records datasets outside the US W. Coast are highlighted and made easily available for researchers to use. I hope this sort of documentation of datasets and their preparation in research-ready format becomes the standard, and I expect that publication of this manuscript will lead to numerous publications on ML that use this dataset.

We are very pleased for the appreciation.

I know some information on the uniqueness of the dataset is dispersed through the article, but I suggest, in a single place, in the discussion, the authors extend and accentuate what is different with this collection compared to others, besides from the obvious that this is solely an Italian dataset based on the Italian earthquake catalogue. Are the metadata fields better? How do they differ? Are formats modified ? Is it unique to provide both raw and corrected waveform data? Data volumes similar to other datasets? Maybe a comparison table to STEAD and the Caltech datasets would be helpful. The authors could also provide stronger comments on the benefits of standardisation of formats / metadata for these datasets.

We would like to thank the ref2 for the very good suggestion. We have followed his advice and included a summary table comparing our metadata with those presented in other similar datasets cited in the ms (i.e., STEAD, the SCEDC Caltech datasets, LEN-DB, the dataset used by Lomax et al., 2019, when developing ConvNetQuake_INGV and the global NEIC dataset developed by Yeck and Patton, 2020). In this new table we have included several parameters that can help the reader to better understand the difference between the datasets. These include the number of traces, the average number of traces per earthquake, the total number of hours the provided traces amount to, the number of receivers, …

In doing this additional work, we have realized that in some cases it was not possible to gather some of the metadata provided in the dataset univocally. For example, the SCEDC data (https://scedc.caltech.edu/data/deeplearning.htm), has been used for different studies (Ross et al., 2018a, 2018b; Meier et al., 2019) each one with different sets of overlapping metadata. The same dataset seems to include data both in SI units and in digital counts but the former is only stated in the README file linked above and not in the paper by Meier et al. (2019). The NEIC dataset which provides data at global level, includes only the relevant part of the trace signal of the phases (e.g., P, Pg, Pn, S, Sg, and Sn) and not the entire waveform.

In the text, we have therefore added the new Table 6

**Table (6).** Comparison between INSTANCE and other published seismic waveform datasets. It was not possible to retrieve some attributes of the original SCEDC dataset since it is available as different subsets extracted for specific application (list available at https://scedc.caltech.edu/data/deeplearing.html). [1] INSTANCE, doi:10.13127/instance. [2] STEAD, doi:10.1109/ACCESS.2019.2947848. [3] SCEDC, https://scedc.caltech.edu/data/deeplearning.html. [4] LenDB, doi:10.5281/zenodo.3648232. [5] ConvNetQuake_INGV (CNQ_INGV), doi:10.5281/zenodo.5040865. [6] NEIC, doi:10.5066/P9OHF4WL. [7] D: digital; P: physical. [8] L: local; R: regional; G: global. [9] BB: broadband; SM: strong motion; SP: short period.

| | INSTANCE[1] | STEAD[2] | SCEDC[3] | LEN-DB[4] | CNQ_INGV[5] | NEIC[6] |
|---|---|---|---|---|---|---|
| Metadata (events) | 115 | 35 | – | 14 | 6 | 8 |
| Metadata (noise) | 46 | 8 | – | 7 | 2 | – |
| Trace length (s) | 120 | 60 | 4,6 | 27 | 50 | 60 |
| Units[7] | D, P | D | D | P | P | D |
| Events | 54,008 | $\sim 450,000$ | 273,882 | 304,874 | 6,213 | 136,716 |
| Traces (events) | 1,159,249 | 1,050,000 | – | 629,095 | 22,046 | – |
| Traces (noise) | 132,288 | $\sim 100,000$ | – | 615,847 | 12,543 | – |
| Receivers | 620 | 2,613 | – | 1,487 | 26 | 2361 |
| Average receivers per event | 21 | 2 | – | 2 | 4 | – |
| Duration in hours (events) | 38,641 | $\sim 17,500$ | – | 4,718 | 306 | – |
| Duration in hours (noise) | 4,409 | $\sim 1,700$ | – | 4,618 | 174 | – |
| Epicentral distance range (km) | $< 620$ | $< 350$ | $< 360$ | $< 189$ | $< 19,310$ | $< 10,000$ |
| Magnitude range | $0 - 6.5$ | $0 - 7.9$ | $-0.81 - 7.3$ | $0.4 - 7.1$ | $3 - 9.1$ | $1 - 8.3$ |
| Sampling rate (Hz) | 100 | 100 | 100 | 20 | 20 | 40 |
| Storage size (GB) | 331.2 | 91.4 | – | 18.4 | 0.9 | $\sim 51$ |
| Focal mechanism | 527 | 6,200 | – | – | – | – |
| Event type[8] | L, R | L | L, G | L | L, R, G | L, R, G |
| Data type[9] | BB, SM, SP | BB, SM, SP | BB, SM | – | BB | BB, SP? |

And the following text regarding the strengthening proposed by the ref2 has been added at the end of the "Discussion" section.

To the purpose of comparison, in Table 6, we summarize the main features of the currently available seismological datasets assembled for ML analysis. As noted above, the main features that distinguish INSTANCE from the other datasets are the number of metadata for both earthquakes and noise traces and the average number of traces per event. In addition, the dataset provides a generally large number of traces for each recording site making the dataset suitable for quite diversified target studies. The dataset is also unique since it is the only one (yet) to provide the waveform traces in both digital counts and physical units. In this context, the set of parameters provided by INSTANCE spans both specific seismological parameters like P and S arrival times, fault plane and moment tensor solutions, and also peak ground motion parameters in physical units (e.g., PGA, PGV) which can be used for studies that target the estimation of the ground shaking (e.g., shakemaps).

For what concerns strengthening the importance of format standardization remarked by the ref2, we have emphasized more the undergoing SeisBench initiative

(https://meetingorganizer.copernicus.org/EGU21/EGU21-12218.html) that was followed throughout our work. Reference to this initiative had been already made at the end of the Applications section (P34 L33 to P35 L3) in our submission and we have reworded the original text to emphasize the benefits of format standardization.

Needless to emphasize that widespread adoption of the same metadata schema and data volume formats can foster the compilation of similar datasets also for other regions with the possibility to merge them all together giving the opportunity to perform ML analysis exploiting the potentials of the resulting huge large datasets. Perhaps more importantly, standardization of data and metadata formats will make it easier to test different datasets using the same ML model or, alternatively, benchmarking different models on the same dataset and in both cases the benefits appear clear.
* * *
A comment on the metadata on earthquake parameters summarised in Table 2: numerous fields are provided, including location uncertainty, but in the text, there is no comment on what location algorithm or velocity model is used. Since the dataset spans 15 years and a very wide geographic range, its likely that despite efforts to ensure a continuous approach to manual catalogue review, the velocity model and location algorithms have changed across the catalogue. If they have not changed, this point should be made. If they have changed, this should be indicated in the document, and consideration should be given to add this information in future updates of the dataset.

A very similar question was asked by ref1. The location of all the earthquakes was made using the same 1D standard model adopted by INGV for the locations reported in the 2008 Bollettino Sismico Italiano (Mele et al, 2010). We have added a small section in the appendix to provide the model. The earthquake locations are performed using the software IPOP developed by Alberto Basili (Bono, 2007). The following text has been added to refer to the appendix B where the velocity model is given.

The event data belong to the Italian Seismic Bulletin (or INGV bulletin hereinafter) which has been adopting the same velocity model and earthquake location software in the time period included in this study (see Appendix B for detail).
* * *
The DOI of each of the network codes used in the dataset / publication must be made available in the references or in the data availability section

Yes, this is definitely important and they have been added in the "Code and data availability" section.

The data used in this work were downloaded using the web services provided by INGV (http://terremoti.ingv.it/en/webservices_and_software). The following networks were used: OX: North-East Italy Seismic Network, https://doi.org/10.7914/SN/OX.ST: Trentino Seismic

Network, https://doi.org/10.7914/SN/ST. SI: Province Sudtirol, No DOI is registered for this network.XO: EMERSITO Working Group. (2018). https://doi.org/10.13127/SD/7TXeGdo5X8. NI: North-East Italy Broadband Net-work https://doi.org/10.7914/SN/NI. IX: Irpinia Seismic Network, http://isnet.fisica.unina.it/, No DOI is registered for this network. OT: OTRIONS, https://doi.org/10.7914/SN/OT. RF: Friuli Venezia Giulia Accelerometric Network, https://doi.org/10.7914/SN/RF. YD (2018-2018): INGV SISMIKO Emergency Seismic Network for Molise-Italy. TV: INGV experiments network, No DOI is registered for this network. AC: Albanian Seismological Network, https://doi.org/10.7914/SN/AC. HL:National Observatory of Athens Seismic Network, https://doi.org/10.7914/SN/HL. ZM (2017-2021): Seismic Emergency for Ischia by Sismiko, No DOI is registered for this network. 3A (2016-2016): Centro di microzonazione sismica Network, 2016 Central Italy seismic sequence, https://doi.org/10.13127/SD/ku7Xm12Yy9.
* * *
In general, the figures are not optimal, often use strange axis labelling (that may be a direct metadata field from Table 2 - if so mention it!), and often have captions that are too terse / insufficiently descriptive. I suggest the authors look through these carefully. Also font sizes on Figs 14,15,16 are too small- in particular the exponents are completely illegible.

We have added a note specifying that the labels of the figures follow from the naming of the metadata and we have made more exhaustive and self-contained descriptive captions for the figures. We e have  also updated the figures 14, 15, 16 and 20 enlarging the fonts

[revised manuscript text omitted]

Generally, the standard of English is very high. In the following are direct places in the text where I suggest technical clarification or language improvements:

p1 l17 dispose of-> make available

Corrected to

...it can be important to make available well organized representative subsets….

P2 l6 …Ml into seismology has shown the… -> …Ml in the field of seismology has highlighted the…

Corrected to

Specifically, the advent of ML in the field of seismology has highlighted the importance ...

P2 l28-30 rephrase sentence

Rephrased as

It follows that in order to attract a broader audience of users and developers there is a strong need to assemble and publish benchmark datasets that can be readily used with the existing software platforms(Mousavi et al., 2019). In practical terms, the matter consists of assembling quality checked data and metadata according to volume and formats ready to be used in ML applications.

P3 l 14 good -> impressive ; are -> have been

Done

In general, the impressive performances  of ML applications have been strongly related to the availability of large amounts of data with associated properly labeled metadata.

P4 l18 I know it is clarified later - much later - but maybe add here that the preferred INGV catalogue magnitude is used here, mainly Ml but sometimes also Mw and MdThe following sentence has been added

...depending on the area. To this regard, the preferred INGV catalogue magnitude is the local magnitude, Ml, (Richter, 1935) but sometimes also Mw and Md (see below for additional detail).

P4 l21 taking -> using

Done

...In seismology, when using earthquake magnitude...

P4 l25: some >M4.0 are rejected. A bit more info on significant events that have been removed is needed. Are these only those that include multiple events in the same time window in the catalogue? I hope no very significant events are rejected simply because a very small foreshock or aftershock is also catalogued…

We attempted to keep all the earthquakes with M>=4 but in some cases this was not possible because data were missing. In all, 30 earthquakes with M>=4 were removed and they are shown in the picture below. Almost all of them occur outside the Italian country borders.

[Figure]

The following text has been added:

the great majority of the earthquakes with M≥4.0. The earthquakes that have been discarded (30) occurred all but five outside the Italian country borders and mainly in the Balkan area. The earthquakes that have been discarded (30) occurred all but five outside the Italian country borders and mainly in the Balkan area. (The earthquakes in Italy, all with M < 5, will be included in a future update of the dataset.)

P4 station selection: mention in this section that only stations on Italian territory are used. Are the Civil Defence stations not added? If not, mention why this very significant dataset missing - is it technical or political?

Also stations belonging to the MedNet network outside Italy and some stations of the Albanian and the Greek networks have been used.  Regarding the Italian Civil Protection (RAN network), these  stations  are not inserted in the dataset because they are not available in EIDA and they are not used for the compilation of the BSI upon which the data selection has been based upon. They may be included in future releases of the dataset although this would involve also the seismogram reading (P and S phases) which may be quite heavy to be completed with the available human resources.

The following text has been added:

We note that the strong motion data provided by the national strong motion network (``Rete Accelerometrica Nazionale'') operated by the Italian Department of Civil Protection do not enter in the earthquake picking and location performed by the INGV staff and the same data are not available through EIDA. They may be included, however, in future releases of the dataset.

P5 l1-5: Be more specific on what picks are made available. I assume the INGV catalogue makes first arriving P and S picks only. No additional phase type is indicated (Pg vs Pn), and secondary phases are also not identified (eg PmP)

In the BSI there is no distinction between Pg and Pn or secondary phases like PmP and in the dataset they are just referred to as P phases.  The following text has been added.

...earthquake location (no distinction is made between Pg and Pn and no secondary phases like PmP are picked);

P5 l6-9 its should be accentuated either here or later that since 1/ not all stations used in the catalogue generation are included, eg foreign-operated stations;

Although very few, there are foreign stations (i.e., AC and HL neworks) in the dataset. There are no stations from the countries bordering northern Italy  because they are not included in the EIDA Italian node that was used. The following sentence has been added to the relevant bullet

• all stations that feature P-wave (and S-wave when available) onset phases used for the preferred earthquake location (no distinction is made between Pg and Pn and no secondary phases like PmP are picked);•all stations with waveform data available through the INGV EIDA node (see the dataset contributing networks in the piediagram of Fig. 5b);

2/  phases with large residuals or low weight are removed that it is not possible to use this dataset to relocate the catalogue.

The main point is that INSTANCE targets ML analysis and the results of the analysis could be then used to re-process the entire data in the archive. Nevertheless, the average number

of stations per earthquake is about 21 and in many cases they would be more than sufficient for relocation.  This issue had been already addressed in the originally submitted manuscript at the bottom of P31 referring to the inconsistency of the resulting power law distribution. To make the point clear we have added the following (in bold below)

Thus, given the criteria adopted it is pleonastic to remark that this dataset is not designed for studies addressing the earthquake magnitude power-law distribution (e.g., the b-value). **Similarly,  although the dataset contains an average of 21 traces per earthquake, it may not be optimal for dedicated earthquake relocation studies.**

P5 waveform data selection: in the case of multiple available sampling rates, I infer that the same sampling rate used to make the manual pick is selected for inclusion here. Or is it the highest available sampling rate for each channel?

The phase arrival  times come from the BSI (i.e., on the original traces before resampling) and no attempt is made to re-pick them after resampling.

P6 l 7 starting -> start

Done

...it can occur that the **start** time of the trace is earlier….

P6l11: I don't understand what is 'arrival time samples'. Why not simply use time in seconds?

The use of  'arrival time samples' serves to  simplify the use of these quantities especially by non-seismologists. The arrival times are also provided.

P6 l 19 Title does not read well. 'Data counts waveforms' is not clear. I suggest rephrase: 'preparation of processed waveforms in digital units'

Changed

P6 2.1.5: the authors should mention are all traces rotated into ZNE, or in the entire Italian catalogue in ZNE by default. If so, I am amazed!

They are by default all along ZNE. We do not have any waveform included that resulted from sensors oriented differently. The following sentence has been added below the processing steps in the section 2.1.5 "Preparation of processed waveforms in digital units"

No rotation of the horizontal component along the N-S and E-W directions was required since all sensors used are oriented accordingly.

P7 l16 … generated the associated ground motion units dataset after… -> … generated a dataset in units of physical ground motion after …

Done

...we have also **generated a dataset in units of physical ground motion after** deconvolving...

P7 l 23 result -> be

Changed

...are not included since they **be** from single or double...

P7 l.29 2.2 Metadata: in source, the location method or velocity model are not included. They should be if either of these have changed over 15 years of the catalogue.

The velocity model used for location in the compilation of the BSI has not been changed in 15 years. Thus the earthquake  locations in this sense are all consistent. Details on the BSI procedure are provided in Mele et al. (2010) and the velocity model is also provided in the additional appendix B.

P8 l25 'missing data' - please expand

In Jozinovic et al. (2020), the dataset used for ML consists of a fixed number of stations and when data from one or more stations are missing (either the whole trace or parts of it), the signal trace is set to be an array of zeros. The ML model used there was found to detect and learn the problematic values, and compensate for it, having a similar prediction accuracy on those stations as the accuracy on the stations which had the input data available. The following sentence has been added for clarification

An approach of this kind has been used by (Jozinovic et al., 2020) for missing data. In Jozinovic et al. (2020), the dataset used for ML consists of a fixed number of stations and when data from one or more stations are missing (either the whole trace or parts of it), the signal trace is set to be an array of zeros. The ML model used there was found to detect and learn the problematic values, and compensate for it, having a similar prediction accuracy on those stations as the accuracy on the stations which had the input data available

P8 l 28 onset -> onsets

Done

Our metadata includes P- and S-wave onsets manually picked by INGV analysts as provided in the INGV bulletin.

P8 l 31 upon -> using

Done

...facilitating the training of ML models **using** traces containing ...

P9 l 6 besides -> in addition to the fact

Done

**In addition to the fact** that not all the metadata...

P9 l11 analogously -> similarly

Done

**Similarly**, it may have also...

P9 l12 remove 'found'

Done

...instrument transfer functions were incorrect producing...

P10 Table 2: location code is not part of the International Registry?

Thanks for noting it. Yes, the location code is not part of the International Registry. Changed accordingly in Table 2.

P12 l 4: outside -> in the near vicinity to

Done

...those earthquakes occurring **in the near vicinity** to the Italian national borders.

P12 l9/10 check red / blue for normal / inverse - seems these are switched in the figure / text. Use either thrust or inverse in both text / figure.

Thanks for noting the inconsistency. Corrected.

The size of the moment tensors symbol is proportional to source_magnitude while the colors are defined according to the prevalent strain regime: black, blue and red for strike slip, normal and thrust faults, respectively. The prevalent strain regime is determined according to the fault's rake as derived from source_mechanisms_strike_dip_rake: strike slip for -45°<rake<45° and 135°<rake<225°; normal for 225°≤rake≤315°; thrust for 45°≤rake≤135°

P12 l14 evidences that quite different…-> demonstrates that quite a different….

Done

... **demonstrates** that quite a different number of...

P13 l3: …it difficult the phase picking -> the phase picking difficult

Done

...is high making the phase picking difficult.

P13 l 5 I'm surprised to see selection criteria was for even number of traces for each channels? Seems in contradiction to 2.1.2, where all reasonable phases according to seismicity were selected. Was seismicity for smaller events actually selected according to numbers of station pick?

These are selection criteria for the *noise recordings* and there is no relation with the number of picks available for each station. We made an attempt to select all the station channels with a more or less even number of recordings.

P13 l6,7 - revise sentence, clumsy.

We did not find the sentence particularly clumsy but we did correct for the adoption of the linear scale for backazimuth. The whole sentence now reads as follows

In Fig. 3, we show the distribution according to magnitude, earthquake to station epicentral distance, earthquake depth and backazimuth of the 3C record traces composing the dataset.

P14 fig 4 caption: diagrams -> diagram

Done

Diagram of the earthquake magnitude...

P14 l3 remove 'included'

Done

although a few thousand  occur in the depth

P14 l4 'great majority' seems an exaggeration.

The histogram in Figure 3d adopts a log scale. This issue was also reported by the other referee and the histogram scale has been changed to linear to better evidence the assertion. The text has been changed as follows

The panels show the histograms using the log10 scale to provide a complete representation of the distribution of the dataset. We adopt the linear scale, however, to emphasize the distribution of the backazimuth in Fig. 3d.

P15 Fig 5 / l9 onwards: the number of up first motion polarities is double that of down. This is surprising, and possible concerning unless there is a reasonable explanation I do not see. The authors should explain this. Is it possible eventype=earthquake is not selected, and blasts are also included here?

We thank the referee for raising this issue which we did not address in our manuscript.

As described in the manuscript, we have adopted the "event" FDSN web service implemented at INGV which adopts the standard FDSN parameters and does not include the "event_type" field for selection. However, it is still possible to download the quakeML (a xml formatted file standard for seismology) for each event which includes the "event_type" parameter. We have therefore proceeded to obtain the event_type value and we have included it as additional parameter (`source_type`) in the metadata file. Thus, the new metadata file now includes 115 parameters total. Nevertheless, it appears that the addition of the new parameter captures only a fraction of the non-earthquake sources. The table below provides a snapshot of the event_type included in the proposed dataset. In addition,

the BSI distinguish between earthquakes and other sources like quarry blasts only since 2012 (Gulia and Gasperini, 2021).

| type_event | |
|---|---|
| anthropogenic event | 1 |
| controlled explosion | 44 |
| earthquake | 53753 |
| experimental explosion | 8 |
| explosion | 5 |
| landslide | 1 |
| quarry blast | 194 |
| volcanic eruption | 2 |

Given that the inclusion of the event_type above still misses several artificial sources, we have addressed the asymmetry between the number of positive and negative polarities by other means. We performed two different analysis to verify i.) how the inclusion of blasts can affect the reported asymmetry and ii.) how the region with its dominant tectonic style can condition  the number of positive and negative polarities in INSTANCE.

Following Mele et al. (2010) who found that the 99.6% of the blasts have local magnitude ML ≤ 2.2 (Fig. 23 of their study), we have progressively increased the lower magnitude threshold to verify whether the nearly 2:1 ratio between positive and negative polarities persists as the magnitude is increased. The expectation is that as the magnitude increases, the ratio progressively levels out since the blasts (or other artificial sources) do not produce magnitudes greater than M=3 in Europe (Giardini et al., 2004).

Secondly, we have subdivided the Italian area into two zones: earthquakes inside the Apennines area [vertices (lat,lon) (41N,9E and 44N,15E)], and earthquakes elsewhere outside this area. This data selection seeks to verify if the observed asymmetry of positive and negative polarities can result from the dominant extensional stress field characterizing the Apennines when compared to the other areas in Italy.

To address the variation of the proportion between positive and negative polarities with magnitude, the table below shows  that the fraction (per cent values) of negative polarities increases progressively from 36% to ~41% when including earthquakes with magnitudes larger that 0.25 and up to M>3. For larger minimum magnitudes, the percentage stabilizes around 42-43%. This would indicate that inclusion of the polarities of unrecognized blasts (i.e., with M<3)  has a moderate impact on the observed asymmetry between the reported positive and negative polarities. This issue, although somewhat surprising, is not new when compared to the number of up and down polarities reported by Ross et al. (2019).  In their analysis of the southern California earthquake dataset, they have (before data augmentation) 67% and 33% for up and down polarities, respectively. We also note that the regional tectonic setting in Southern California is quite different from that in Italy.

| min_magnitude | total | positive | positive_percent | negative | negative_percent |
|---|---|---|---|---|---|
| 0.25 | 236345 | 151544 | 64.12 | 84801 | 35.88 |
| 0.5 | 235806 | 151204 | 64.12 | 84602 | 35.88 |
| 0.75 | 234400 | 150335 | 64.14 | 84065 | 35.86 |
| 1 | 227810 | 146213 | 64.18 | 81597 | 35.82 |
| 1.25 | 219688 | 141096 | 64.23 | 78592 | 35.77 |
| 1.5 | 204277 | 131159 | 64.21 | 73118 | 35.79 |
| 1.75 | 194880 | 125020 | 64.15 | 69860 | 35.85 |
| 2 | 160464 | 102359 | 63.79 | 58105 | 36.21 |
| 2.25 | 118581 | 75072 | 63.31 | 43509 | 36.69 |
| 2.5 | 75907 | 46810 | 61.67 | 29097 | 38.33 |
| 2.75 | 57366 | 34740 | 60.56 | 22626 | 39.44 |
| 3 | 37183 | 21821 | 58.69 | 15362 | 41.31 |
| 3.25 | 27333 | 15748 | 57.62 | 11585 | 42.38 |
| 3.5 | 16749 | 9447 | 56.4 | 7302 | 43.6 |
| 3.75 | 12328 | 6979 | 56.61 | 5349 | 43.39 |
| 4 | 7200 | 4151 | 57.65 | 3049 | 42.35 |
| 4.25 | 4935 | 2810 | 56.94 | 2125 | 43.06 |
| 4.5 | 2232 | 1312 | 58.78 | 920 | 41.22 |
| 4.75 | 1369 | 833 | 60.85 | 536 | 39.15 |
| 5 | 814 | 468 | 57.49 | 346 | 42.51 |

For our second analysis (proportion between positive and negative polarities depending on the area), we have considered that in Europe the maximum magnitude of quarry blasts is usually assumed to be 2.5–3.0 (Giardini ed al., 2004) and following the findings of Mele et al (2010), we focus only on earthquakes with M> 2.5.  We have extracted the polarities for the target Apennine region and compared to those reported for earthquakes elsewhere in Italy. In the target area, the largest majority of the earthquakes  feature normal faulting mechanism which feature the lobes of the seismic radiation pattern having negative polarities at short epicentral distances. That is, the observed asymmetry could result from the complex interplay between the source receiver geometry, the width of 200-300 km coast to coast from the Tirrhenian to the Adriatic seas of peninsular Italy and the dominant extensional faulting with faults striking NW-SE characterizing the Apennines and dominated by normal faulting. In this setting, the radiation pattern predicts negative polarities in the near source  and positive polarities farther away. The negative polarity source radiation lobe maps, however, into a smaller  extension region near the epicenter with a smaller number of stations when compared to the other lobe of larger extension and a larger number of stations reporting positive polarities.

The figure below shows the histograms of the distribution of the positive and negative polarities with distance. The panel to the left shows the distribution of the polarities for the chosen target area in peninsular Italy, in the middle the polarities in the same area but only along the NE-SW directions of the backazimuth (i.e., 45-135 and 225-315 degrees)  and to the right in the area outside this target area. We note that within the target area the polarities are overwhelmingly positive in gross agreement  with what described above and, for further confirmation, we see that if we restrict to the NE-SW propagation  direction perpendicular to the Italian peninsula (rightmost panel), the ratio between positive and negative polarities  (%pos,%neg) increases from (68%,32%) to (72%,28%), respectively. Conversely, the number of polarities for the earthquakes outside the target area are pretty much well balanced (49%,51%).

[Figure]

In conclusion, i.) the INSTANCE dataset does contain positive polarities resulting from the inclusion of  quarry blasts misidentified as earthquakes for magnitudes less than ~2.5-3.0. This follows from what reported by Mele et al. (2010) and published recently by Gulia and Gasperini (2021)  and the change in positive and negative polarities percentages  reported in the table above appears to confirm it; ii.)  the current modalities of earthquake revision at INGV do not include accurate identification of quarry blasts, the web service used does not include the *eventtype* identification yet (it is not yet a FDSN standard) but it was still possible to retrieve the event_type and, accordingly, add a new source parameter (source_type) to the dataset metadata;  iii.) the target area in the selected Apennine region includes ~76% of the total number of polarities of the dataset; iv) In the Apennine region there is dominance of positive polarities which is likely the result of the dominant normal type of earthquake faulting in the area; v) the asymmetry observed in the target area disappears for M>2.5 elsewhere in Italy.

The new Appendix C provides a more concise description of the explanation provided above. We report below the appendix for completeness.

**Appendix C: Positive and negative polarities**

15 In this appendix, we examine the origin for the observed asymmetry (almost 2:1 ratio) in the number of reported positive (up) and negative (down) polarities of the INSTANCE dataset. We evaluate whether *i)* the inclusion of anthropic, unidentified sources like quarry blasts mistaken for earthquakes, can affect the reported asymmetry and *ii)* the tectonics of the region can condition the number of positive and negative polarities in INSTANCE.

For the first investigation, we have followed Mele et al. (2010) (see also the recent work by Gulia and Gasperini, 2021)

20 who found that in the 2008 bulletin the 99.6 % of the blasts have local magnitude $M_L \leq 2.2$ (Fig. 23 of their study). We have progressively increased the lower magnitude threshold to verify whether the nearly 2:1 ratio between positive and negative

polarities persists as the magnitude is increased. The expectation is that, as the magnitude increases, the ratio progressively levels out since the blasts (or other artificial sources) do not produce magnitudes greater than $M=3$ in Europe (cf. Giardini et al., 2004). To address the variation of the proportion between positive and negative polarities with magnitude, we have selected from the INSTANCE dataset earthquakes with progressively larger minimum threshold magnitudes and found that the fraction (percent values) of negative polarities increases progressively from 36 % to 41 % when including earthquakes with $M > 0.25$ and $M > 3$, respectively. For larger minimum magnitudes, the percentage stabilizes around 42-43 %. This indicates that inclusion of the polarities of e.g., unrecognized blasts (i.e., with $M < 3$) has a moderate but still significant impact on the observed asymmetry between the reported positive and negative polarities. This asymmetry, although somewhat surprising, seems to occur also elsewhere. For example, Ross et al. (2018b) report, in their analysis of the southern California earthquake dataset (before data augmentation), a content of 67 % and 33 % for up and down polarities, respectively. We also note that the regional tectonic setting in Southern California is quite different from that in Italy.

Secondly, we have subdivided the Italian area into two zones selecting earthquakes with magnitude $M > 2.5$ occurring in the Apennines (defined as a rectangular area from 41°N to 44°N latitude and 9°E to 15°E longitude), and elsewhere outside this area. This data selection is aimed to verify if the observed asymmetry of positive and negative polarities can result from the dominant extensional stress field characterizing the Apennines when compared to the other areas in Italy. In the Apennines target area, the largest majority of the earthquakes are characterized by normal faulting mechanisms: in this case the lobes of the seismic radiation pattern show negative polarities at short epicentral distances.

[Figure]

**Figure (C1).** Distribution of the positive and negative P-wave polarities for earthquakes with $M > 2.5$ in the Apennines region (41°N – 44°N and 9°E – 15°E) (a); (b) as in (a) but for earthquakes outside the Apennines; (c) as in (a) but filtered by backazimuth along the NE-SW directions, corresponding to the intervals $15° – 105°$ and $195° – 285°$.

Given these conditions, the observed asymmetry could result from the complex interplay between the source receiver geometry, the 200-300 km width of peninsular Italy and the dominant NW-SE extensional faulting characterizing the tectonic regime in the Apennines. In this setting, the radiation pattern predicts negative polarities in the near field and positive polarities farther away from the source. Also the negative lobes are characterized by a smaller extension with respect to the positive lobes. Given that the seismic receivers are more or less evenly distributed, this implies that the chance to record a negative pulse is smaller with respect to the positive one.

40

The Fig. C1 shows the histograms of the distribution of the positive and negative polarities with distance. The Fig. C1(a) shows the distribution of the polarities for the chosen target area in the Apennines, while Fig. C1(c) exhibits the polarities in the same area but only along backazimuth approximately NE-SW (i.e., the ranges $15°–105°$ and $195°–285°$ degrees). Finally, Fig. C1(b), displays the polarities from earthquakes outside the target area. We note that within the target area (Fig. C1(a))

5    the polarities are overwhelmingly positive in gross agreement with the explanation above. For further confirmation, we note in Fig. C1(c) that, if we restrict to the NE-SW propagation direction perpendicular to the Italian peninsula, the ratio between positive and negative polarities (pos %, neg %) increases further from (68 %, 32 %) to (81 %, 19 %), respectively. Conversely, the number of polarities for the earthquakes outside the target area are pretty much well balanced (49 %, 51 %) as shown in Fig. C1(b) .

10   In conclusion, *i)* the INSTANCE dataset does contain positive polarities resulting from the inclusion of quarry blasts misidentified as earthquakes for magnitudes less than $\sim2.5 - 3.0$. This follows from what reported by Mele et al. (2010) (see also the recent work by Gulia and Gasperini, 2021) and the change in positive and negative polarities proportions in INSTANCE at varying minimum magnitude thresholds appears to confirm it; *ii)* the current modalities of earthquake revision at INGV do not allow for the identification of all the anthropic sources so that the (`source_type`) parameter of the metadata can be

15   misleading; *iii)* the target area in the selected Apennines region includes 76 % of the total number of polarities of the dataset and features a dominance of positive polarities: a possible explanation is the dominant normal type of earthquake faulting in the selected Apennines region; *iv)* the asymmetry observed in the target Apennines region disappears for $M > 2.5$ elsewhere in Italy.

P16 l10 wide range 'of' waveform…

Done

...showing the wide range **of** waveform paths...

P17 l1 1/2: is it possible this can also be explained by systematically mis-identified first arrivals, rather than complications in the velocity structure?

It has been verified that these very long traveltimes belong to earthquakes that occurred during the 2012 Emilia earthquake sequence. The stations recording these events were located on the soft and thick alluvium characterizing the Po plain which features very low seismic velocities.

P17 l 12 evidence -> display or feature?

Done

...do **display** a broader distribution….

P18 l9/10 rephrase sentence!

The sentence is "This is expected because the S-wave motion is polarized perpendicular to the nearly vertical propagation direction at the surface, implying that the ground motion occur mainly along the horizontal components" has been reworded as follows

This is expected because the S-wave motion in the shallow, near surface low velocity layers is polarized on a plane perpendicular to the nearly vertical propagation direction of the wavefront, implying that the ground motion occurs mainly along the horizontal components.

P18 l11 rather satisfying -> sensible?

Done

...SNR values of our dataset can be considered **sensible** given that values larger than...

P19 l1 at higher values -> at higher and lower values

Done

...some horizontal stripes at higher **and lower** values of ground...

P19 l4 …IMs can be assimilated to an -> IMs represent an

Done

...concentration of **IMs represent** an average...

P20 Fig 10: add to caption that top shows linear y-axes, bottom shows logarithmic y-axes. See caption of Fig. 10 above.

P21 l7: 'exhaustive' is quite an exaggeration - its more to show how metadata can be used to isolate end members.

This whole sentence has been changed to

To show how metadata can be used to isolate end members of the dataset, we focus next on examples of problematic traces.

P24 Fig 14: would be good to mention here and elsewhere that the fields in italics can be defined in Table 2. (Same for Fig 15, 16)

Reference in the caption to the metadata listed in Table 2 has been provided for all relevant figures and tables (see captions listed above).

 earthquake in INGV catalogue - so its very possible that noise traces include energy from regional and telesiesmic events.

Yes the referee is correct. We pointed this out in the manuscript. The following text has been added in the "Data preparation" section

We note also that this procedure does not preclude the presence of noise traces that include energy from regional and teleseismic events.

P27 l14: any effort to include the same spread of stations as found in the event dataset?

No, if for spread it is meant the same group of stations detecting earthquakes in a given area for the same time window. Anyhow, the stations are exactly the same as those of the event dataset as evidenced in Figure 2.

P27 l17 46 metadata -> 46 metadata elements

Done

The 46 metadata **elements** ….

P33 l 16: reveal -> prove

Done

...benchmark datasets can **prove** very effective…

P34 l 35 Sesbench -> SeisBench

Done

….have adopted the schema proposed by the **SeisBench** initiative...

P39 FigA4 - over 100 records have PGA >2g, and many even over 4g. Which is rather unphysical.  Is this understood?

The units are cm/s^2 and we do not see any value above 1g for PGA.

---

## Editor Decision (ED1)

Dear Alberto Michelini and co-authors,

Many thanks for the very comprehensive revision of the manuscript as suggested by the referees.

Before finally accepting the paper for publication, I would like to suggest s slight modification to the data availability section, where you newly added the seismic networks that you have used to derive your data. These sources for the data should also be properly cited and included in the "References" section of the article. Please use the following version of the paragraph (I am using track change mode) or organise the data sources as a small table.

Furthermore, I would like to encourage you to add all seismic networks that you are identifying as source for your data to the DOI landing page. Interestingly, all DOIs and URLs are already in the DataCite metadata, but unfortunately not visible on the DOI landing page.

**Please update this paragraph in the article (in the Data availablity section):**

The data used in this work were downloaded using the web services provided by INGV (http://terremoti.ingv.it/en/webservices_and_software). The following networks were used: OX: North-East Italy Seismic Network, https://doi.org/10.7914/SN/OX (OSG, 2016). ST: Trentino Seismic 10 Network, https://doi.org/10.7914/SN/ST (Geological Survey-Provincia Autonoma di Trento, 1981). SI: Province Sudtirol, No DOI is registered for this network (URL: https://www.fdsn.org/networks/detail/SI/). XO: EMERSITO Working Group. (2018). https://doi.org/10.13127/SD/7TXeGdo5X8 (EMERSITO Working Group, 2018). NI: North-East Italy Broadband Net-work https://doi.org/10.7914/SN/NI (OGS and University of Trieste, 2002). IX: Irpinia Seismic Network, http://isnet.fisica.unina.it/, No DOI is registered for this network. OT: OTRIONS, https://doi.org/10.7914/SN/OT (University of Bari "Aldo Moro", 2013). RF: Friuli Venezia Giulia Accelerometric Network, https://doi.org/10.7914/SN/RF (University of Trieste, 1993). YD (2018-2018): INGV SISMIKO Emergency Seismic Network for Molise-Italy. TV: INGV experiments network, No DOI is registered for this network (INGV, 2008). AC: Albanian Seismological Network, https://doi.org/10.7914/SN/AC (Institute of Geosciences, Energy, Water and Environment, 2002). HL:National Observatory of Athens Seismic Network, https://doi.org/10.7914/SN/HL (National Observatory of Athens and Institute of Geodynamics, Athens, 1997). ZM (2017-2021): Seismic Emergency for Ischia by Sismiko, https://www.fdsn.org/networks/detail/ZM_2017/ (INGV, 2017) No DOI is registered for this network. 3A (2016-2016): Centro di microzonazione sismica Network, 2016 Central Italy seismic sequence, https://doi.org/10.13127/SD/ku7Xm12Yy9 (INGV et al., 2018)

**These are the new references (to be included in the "References" section):**

(please note that I also suggest to cite seismic networks (with URL) that are not assigned with a DOI)

OGS - Istituto Nazionale di Oceanografia e di Geofisica Sperimentale -: North-East Italy Seismic Network, https://doi.org/10.7914/SN/OX, 2016.

Geological Survey-Provincia Autonoma di Trento: Trentino Seismic Network, https://doi.org/10.7914/SN/ST, 1981.

ZAMG - Central Institute for Meteorology and Geodynamics. SI Province Südtirol Seismic Network. URL: https://www.fdsn.org/networks/detail/SI/, 2006

EMERSITO Working Group: Rete sismica del gruppo EMERSITO, sequenza sismica del 2016 in Italia Centrale, https://doi.org/10.13127/SD/7TXEGDO5X8, 2018.

OGS (Istituto Nazionale di Oceanografia e di Geofisica Sperimentale) and University of Trieste: North-East Italy Broadband Network, https://doi.org/10.7914/SN/NI, 2002.

University of Bari "Aldo Moro": OTRIONS, https://doi.org/10.7914/SN/OT, 2013.

University of Trieste: Friuli Venezia Giulia Accelerometric Network, https://doi.org/10.7914/SN/RF, 1993.

Istituto Nazionale di Geofisica e Vulcanologia (INGV). TV INGV experiments network. URL: https://www.fdsn.org/networks/detail/TV/, 2008

Institute of Geosciences, Energy, Water and Environment: Albanian Seismological Network, https://doi.org/10.7914/SN/AC, 2002.

National Observatory of Athens, Institute of Geodynamics, Athens: National Observatory of Athens Seismic Network, https://doi.org/10.7914/SN/HL, 1997.

Istituto Nazionale di Geofisica e Vulcanologia (INGV). ZM (2017-2021) Seismic Emergency for Ischia by Sismiko (SEISk). URL: https://www.fdsn.org/networks/detail/ZM_2017/, 2017.

Istituto Nazionale di Geofisica e Vulcanologia (INGV), Istituto di Geologia Ambientale e Geoingegneria (CNR-IGAG), Istituto per la Dinamica dei Processi Ambientali (CNR-IDPA), Istituto di Metodologie per l'Analisi Ambientale (CNR-IMAA), and Agenzia Nazionale per le nuove tecnologie, l'energia e lo sviluppo economico sostenibile (ENEA): Rete del Centro di Microzonazione Sismica (CentroMZ), sequenza sismica del 2016 in Italia Centrale, https://doi.org/10.13127/SD/KU7XM12YY9, 2018.

---

## Author Response (AR2)

Dear Editor,

we are very pleased that you found our revision comprehensive. In the reply below, we have used black fonts for the comments received and we adopt blue fonts for our reply.

In the companion pdf file (*[Compare Report] INSTANCE_Overleaf_revised_final.pdf*) that compares the previous version and the current one you will find all the modifications obtained using an advanced feature of the acrobat reader software. We adopted this software since we experienced problems with "latexdiff".  In this file you will also find some other minor changes (e.g., in Table 6 we changed the number of metadata for the NEIC dataset after contacting one of the authors of this dataset) and the citation of the panda software that was recently changed.

We are looking forward to hearing from you on the final decision.

Best regards,

Alberto Michelini
* * *
Dear Alberto Michelini and co-authors,

Many thanks for the very comprehensive revision of the manuscript as suggested by the referees.

Thank you

Before finally accepting the paper for publication, I would like to suggest a slight modification to the data availability section, where you newly added the seismic networks that you have used to derive your data. These sources for the data should also be properly cited and included in the "References" section of the article. Please use the following version of the paragraph (I am using track change mode) or organise the data sources as a small table.

We have made the modifications you suggested and, following your advice, we inserted a table in the data availability section that you can see below.

**Table (7).** Seismic networks used in the compilation of INSTANCE dataset

| Code | Name | Identifier | Citation |
|---|---|---|---|
| 3A | Seismic Microzonation Network 2016 Central Italy (2016-2016) | https://doi.org/10.13127/SD/ku7Xm12Yy9 | Istituto Nazionale di Geofisica e Vulcanologia (INGV) et al. (2018) |
| AC | Albanian Seismological Network | https://doi.org/10.7914/SN/AC | Institute of Geosciences, Energy, Water and Environment (2002) |
| BA | UniBAS | https://www.fdsn.org/networks/detail/BA/ | Universita della Basilicata (2005) |
| GU | Regional Seismic Network of North Western Italy | https://doi.org/10.7914/SN/GU | University of Genoa (1967) |
| HL | National Observatory of Athens Seismic Network | https://doi.org/10.7914/SN/HL | National Observatory of Athens, Institute of Geodynamics, Athens (1997) |
| IV | Italian National Seismic Network | https://doi.org/10.13127/SD/X0FXNH7QFY | INGV Seismological Data Centre (1997) |
| IX | Irpinia Seismic Network | http://isnet.unina.it/ | Universita Federico II (2005) |
| MN | MedNet | https://doi.org/10.13127/SD/FBBBTDTD6Q | MedNet Project Partner Institutions (1988) |
| NI | North-East Italy Broadband Network | https://doi.org/10.7914/SN/NI | OGS (Istituto Nazionale di Oceanografia e di Geofisica Sperimentale) and University of Trieste (2002) |
| OT | OTRIONS | https://doi.org/10.7914/SN/OT | University of Bari "Aldo Moro" (2013) |
| OX | North-East Italy Seismic Network | https://doi.org/10.7914/SN/OX | Istituto Nazionale di Oceanografia e di Geofisica Sperimentale - OGS (2016) |
| RF | Friuli Venezia Giulia Accelerometric Network | https://doi.org/10.7914/SN/RF | University of Trieste (1993) |
| SI | Province Sudtirol | https://www.fdsn.org/networks/detail/SI/ | ZAMG - Zentralanstalt fur Meterologie und Geodynamik (2006) |
| ST | Trentino Seismic Network | https://doi.org/10.7914/SN/ST | Geological Survey-Provincia Autonoma di Trento (1981) |
| TV | INGV experiments network | https://www.fdsn.org/networks/detail/TV/ | Istituto Nazionale di Geofisica e Vulcanologia (INGV) (2008) |
| XO | EMERSITO Working Group. (2018) | https://doi.org/10.13127/SD/7TXeGdo5X8 | EMERSITO Working Group (2018) |
| YD | INGV SISMIKO Emergency Seismic Network for Molise-Italy (2018-2018) | https://doi.org/10.13127/SD/FIR72CHYWU | Moretti et al. (2018) |
| ZM | Seismic Emergency for Ischia by Sismiko (2017-2021) | https://www.fdsn.org/networks/detail/ZM_2017/ | Istituto Nazionale di Geofisica e Vulcanologia (INGV) (2017) |

We think that the table format is probably optimal to acknowledge the contributing networks. All the column(s) information is (are) taken from the FDSN web site (https://www.fdsn.org/networks/) which reports either the DOI or the network URL information. In addition, all the Identifier and Citations are clickable. When performing this revision we realized that four networks (IV, MN, GU, BA) were missing and we added them and now the total number of 18 networks corresponds to those contained in the dataset.

All the networks are now included in the "References".

In addition, since we have been using the latex template for the manuscript, we adopted the .bib files for the bibliography and these files are provided only for the networks with an assigned DOI. For the networks that have not been assigned a DOI, we have taken the information provided on the FDSN web site and created the corresponding .bib file.

Furthermore, I would like to encourage you to add all seismic networks that you are identifying as source for your data to the DOI landing page. Interestingly, all DOIs and URLs are already in the DataCite metadata, but unfortunately not visible on the DOI landing page.

The INSTANCE landing page (http://www.pi.ingv.it/instance/) has been adjourned following your encouragement and you find the link "INSTANCE dataset" within the "Data Sources" section (https://data.ingv.it/en/dataset/471#related-doc).

Please update this paragraph in the article (in the Data availability section):

The data used in this work were downloaded using the web services provided by INGV (http://terremoti.ingv.it/en/webservices_and_software). The following networks were used: OX: North-East Italy Seismic Network, https://doi.org/10.7914/SN/OX (OSG, 2016). ST: Trentino Seismic 10 Network, https://doi.org/10.7914/SN/ST (Geological Survey-Provincia Autonoma di Trento, 1981). SI: Province Sudtirol, No DOI is registered for this network (URL:

https://www.fdsn.org/networks/detail/SI/). XO: EMERSITO Working Group. (2018). https://doi.org/10.13127/SD/7TXeGdo5X8 (EMERSITO Working Group, 2018). NI: North-East Italy Broadband Net-work https://doi.org/10.7914/SN/NI (OGS and University of Trieste, 2002). IX: Irpinia Seismic Network, http://isnet.fisica.unina.it/, No DOI is registered for this network. OT: OTRIONS, https://doi.org/10.7914/SN/OT (University of Bari "Aldo Moro", 2013). RF: Friuli Venezia Giulia Accelerometric Network, https://doi.org/10.7914/SN/RF (University of Trieste, 1993). YD (2018- 2018): INGV SISMIKO Emergency Seismic Network for Molise-Italy. TV: INGV experiments network, No DOI is registered for this network (INGV, 2008). AC: Albanian Seismological Network, https://doi.org/10.7914/SN/AC (Institute of Geosciences, Energy, Water and Environment, 2002). HL:National Observatory of Athens Seismic Network, https://doi.org/10.7914/SN/HL (National Observatory of Athens and Institute of Geodynamics, Athens, 1997). ZM (2017-2021): Seismic Emergency for Ischia by Sismiko, https://www.fdsn.org/networks/detail/ZM_2017/ (INGV, 2017) No DOI is registered for this network. 3A (2016-2016): Centro di microzonazione sismica Network, 2016 Central Italy seismic sequence, https://doi.org/10.13127/SD/ku7Xm12Yy9 (INGV et al., 2018)

Please see the table inserted above.

These are the new references (to be included in the "References" section):

==(please note that I also suggest to cite seismic networks (with URL) that are not assigned with a DOI)==

OGS - Istituto Nazionale di Oceanografia e di Geofisica Sperimentale -: North-East Italy Seismic Network, https://doi.org/10.7914/SN/OX, 2016.

Geological Survey-Provincia Autonoma di Trento: Trentino Seismic Network, https://doi.org/10.7914/SN/ST, 1981.

ZAMG - Central Institute for Meteorology and Geodynamics. SI Province Südtirol Seismic Network. URL: https://www.fdsn.org/networks/detail/SI/, 2006

[revised manuscript text omitted]